# Isotopic evidence of high reliance on plant food among Later Stone Age hunter-gatherers at Taforalt, Morocco

Zineb Moubtahij [1,2] ✉, Jeremy McCormack [1,3], Nicolas Bourgon[1,4], Manuel Trost[1], Virginie Sinet-Mathiot [1,5,6], Benjamin T. Fuller[2], Geoff M. Smith [1,7], Heiko Temming[1], Sven Steinbrenner[1], Jean-Jacques Hublin [1,8], Abdeljalil Bouzouggar[9,10], Elaine Turner[11] & Klervia Jaouen[1,2]

The transition from hunting-gathering to agriculture stands as one of the most important dietary revolutions in human history. Yet, due to a scarcity of well-preserved human remains from Pleistocene sites, little is known about the dietary practices of pre-agricultural human groups. Here we present the isotopic evidence of pronounced plant reliance among Late Stone Age hunter-gatherers from North Africa (15,000–13,000 cal BP), predating the advent of agriculture by several millennia. Employing a comprehensive multi-isotopic approach, we conducted zinc ($\delta^{66}Zn$) and strontium ($^{87}Sr/^{86}Sr$) analysis on dental enamel, bulk carbon ($\delta^{13}C$) and nitrogen ($\delta^{15}N$) and sulfur ($\delta^{34}S$) isotope analysis on dentin and bone collagen, and single amino acid analysis on human and faunal remains from Taforalt (Morocco). Our results unequivocally demonstrate a substantial plant-based component in the diets of these hunter-gatherers. This distinct dietary pattern challenges the prevailing notion of high reliance on animal proteins among pre-agricultural human groups. It also raises intriguing questions surrounding the absence of agricultural development in North Africa during the early Holocene. This study underscores the importance of investigating dietary practices during the transition to agriculture and provides insights into the complexities of human subsistence strategies across different regions.

While the term 'Neolithic' remains ambiguous and this period occurred at different times worldwide, it generally implies the domestication of wild animals and plants, as well as the adoption of sedentary settlements[1,2]. The transition from hunting-gathering economies to agriculture-based ones, also known as Neolithization, is one of the most important dietary revolutions in human history[3,4]. Beyond being a revolution, a progressive intensification of plant consumption is believed to have begun long before domestication in the Neolithic[1,5]. Evidence of an early shift to grain-based resources is demonstrated

by the discovery of a substantial archaeobotanical assemblage in the Upper Palaeolithic site of Ohalo II, in the Near East, dated to approximately 23,000 cal BP[5] (Fig. 1). This transformation intensified with the Natufians, a hunter-gatherer group that inhabited the Near East during the Late Pleistocene and the beginning of the Holocene (14,600–11,500 cal BP)[6]. A shift towards an increased reliance on plant foods occurred during this period[6–8], probably driven by several factors, including the depletion of large game species and the availability of a wider range of edible plants in the environment, which led to the

**Fig. 1 | Location of the Taforalt site in Morocco and the other sites mentioned in the text.** The circles indicate Iberomaurusian sites, the squares indicate European Upper Palaeolithic sites, the triangle indicates the Natufian site and the star indicates the Neolithic site in the Levant.

adoption of a broad-spectrum diet[9]. Natufian hunter-gatherers also engaged in early forms of plant cultivation, such as the intentional planting and harvesting of wild cereals. This practice probably paved the way for the development of agriculture in the region[10,11].

The preconditions of the transition to food production in the Levant are deeply rooted in the Natufian hunter-gatherers, but this transition is still a poorly understood and complex phenomenon in northwest Africa[12]. In this region, a shift towards a reliance on plant resources in the diet was thought to be a relatively late phenomenon, which started with the spread of domesticated species from the Near East into this region during the Neolithic (~7,600 BP)[12–14]. In recent years, scholars have become increasingly interested in whether the Iberomaurusians, a population with some genetic connections with the Natufians[15], exhibited changes that preceded the transition to farming in North Africa[16,17]. Recent investigations at the site of Taforalt (Fig. 1), Morocco have suggested early consumption of carbohydrate-rich plants associated with the Iberomaurusian culture. This has been attested by the high number of wild plant taxa along with the prevalence of tooth caries among the human burials[17].

The Iberomaurusian hunter-gatherers, characterized by bladelet-based technology, inhabited North Africa during the Late Pleistocene. The first evidence of this culture, found in Tamar Hat (Fig. 1), dates back to 25,000 cal BP[18,19]. The timing of its end remains uncertain, with some evidence suggesting the possibility of its persistence into the Holocene after 11,000 cal BP[20,21].

Two key areas of interest are the domestication of plant and animal species, a crucial step in agricultural development, and the adoption of a sedentary lifestyle, often associated with plant cultivation. While there is no evidence of local domestication during the Iberomaurusian period[22–24], some behaviours suggestive of a shift towards sedentism in the subsistence economy were present among these hunter-gatherers. For example, at the Iberomaurusian site of Taforalt (Fig. 1), evidence points to the selective harvesting and possible storage of some edible plant species[17]. This is documented by the presence of fragments of alfa grass (*Stipa tenacissima*), which would have been used to make baskets. Wild plants have been recovered from other Iberomaurusian sites and could have been collected for the purpose of consumption, such as at Tamar Hat, Algeria[25], and Ifri el Baroud, Morocco (Fig. 1)[26].

Currently, our knowledge of the Iberomaurusian diets is mostly derived from zooarchaeological evidence. Studies have revealed that the Iberomaurusians relied primarily on ungulates, mainly represented by the Barbary sheep (*Ammotragus lervia*), in addition to snails[24,27]. These conclusions find further support in an isotopic study conducted on bulk collagen, which identified a predominance of meat in the diet of the Taforalt humans[28]. Studies on the exploitation of marine resources for food are scarce despite both the proximity of Iberomaurusian sites to the coast[29] and the recovery of marine mollusc shells from various Iberomaurusian sites, where these shells appear to have been used for ornamental purposes[29].

However, it is worth noting that the faunal remains may not fully represent the entire spectrum of the foods consumed. This limitation arises because plant remains are less likely to preserve well in the archaeological record, and their recovery and identification may not be as frequent as that of animal bones[30–32]. Furthermore, the detection of plant consumption can be easily overprinted by the presence of meat consumption when assessed using nitrogen isotopes on bulk collagen[33]. In terms of settlement patterns, while no stone-built structures similar to those in Natufian settlements are evident[5], the presence of large Iberomaurusian cemeteries (such as Taforalt and Afalou; Fig. 1) in frequently reused sheltered sites—from 15,000 to 13,000 cal BP (Fig. 1)[17,34]—is interpreted as evidence of sedentarism[35].

Taforalt is one of the two largest known Iberomaurusian cemeteries. This site has yielded substantial amounts of recovered plant remains. In addition, it contains the longest and best-dated occupation sequence for the Iberomaurusian period[35–37]. To date, it is one of the oldest cemeteries in North Africa, with the largest number of human burials (including adults, adolescents and infants). The human remains were directly dated to 15,077 to 13,892 cal BP[17], which coincides with a rapid warming period following the Last Glacial Maximum[26]. It is a key site for studying human dietary behaviour during the Late Pleistocene in North Africa and offers an exceptional opportunity to investigate human dietary behaviours at the end of the Late Pleistocene and before the spread of farming practices in the region. In addition, we have at this site contradictory evidence of dietary reliance on meat (faunal remains[24], C and N isotopes[28]) and plant foods (plant remains, tooth caries[17]). A plant-based diet combined with economic intensification could indicate a transitional subsistence strategy towards

sedentism. By combining previously used isotope tracers and new ones that are more sensitive to plant consumption, we aimed here to investigate the dietary habits and the mobility patterns of pre-Neolithic hunter-gatherers in North Africa at Taforalt. In particular, we investigated the proportion of plants in their diet and whether this population was relying on local foods.

To accomplish this, we evaluated the bulk stable isotope compositions of carbon ($\delta^{13}C_{collagen}$) and nitrogen ($\delta^{15}N_{collagen}$) in bone and dentine collagen to reconstruct dietary patterns of both human individuals and coexisting fauna rather than to determine the presence/absence of food products in the diet of a population (Supplementary Information Section 2 and Supplementary Fig. 1). However, these bulk isotopic results can be impacted by baseline variations related to environmental parameters such as aridity, essential element availability or the nature of local mycorrhizae[38–40]. To overcome this issue, compound-specific isotope analysis of single amino acids (CSIA-AA) is used to determine more precisely the trophic position (TP) of an organism independent of environmental factors using the $\delta^{15}N$ results for two amino acids: Phe and Glu[38]. In addition, $\delta^{13}C$ analysis of amino acids such as Phe and Val can effectively distinguish between four main dietary groups ($C_3$, $C_4$, marine and freshwater) (Supplementary Information Section 2)[41].

While organic isotopic proxies are powerful for dietary reconstruction, their application in Africa often faces challenges due to limited collagen preservation in fossil remains from arid environments[42]. Hence, we enhanced our analysis by investigating zinc isotope ratios ($\delta^{66}Zn$) in tooth enamel, a method that has been proved to reliably document trophic levels[43–45] even in the absence of collagen preservation[43,46,47]. Zinc and nitrogen isotope ratios have an inverse relationship, wherein lower $\delta^{66}Zn$ values reflect an elevation in the TP. Given that baseline effects related to geological and environmental parameters can influence $\delta^{66}Zn$ values[43,46,48], we ensured valid comparisons by also conducting analyses of commonly used mobility indicators, including strontium ($^{87}Sr/^{86}Sr$) and sulfur ($\delta^{34}S$) isotope ratios[49,50]. All details on these isotopic proxies are provided in Supplementary Information Section 2.

We analysed the human remains from the Iberomaurusian burials recovered from sector 10 and associated fauna at Taforalt (Supplementary Information Section 1 and Supplementary Tables 1–3). Human samples consisted of 25 teeth (permanent and deciduous) and seven bone samples belonging to seven identified and ten unassigned individuals (Supplementary Tables 3, 14 and 23). The tissues sampled record different periods of the lives of the individuals, including the breastfeeding period (Supplementary Table 15). Special attention was therefore paid to the potential impact of breastmilk consumption on the isotope ratios throughout the text and figures (Figs. 2 and 3 and Supplementary Information Section 5). To preserve morphometric information, the human teeth samples were CT-scanned, and we took this opportunity to document the presence or absence of hypoplasia and caries (Supplementary Information Sections 3 and 7 and Supplementary Table 4). We selected several teeth and bones from various species of associated faunal taxa (sectors 8 and 10) that were exploited by humans to reconstruct the isotopic baseline for Taforalt[24] ($n_{samples} = 20$; Supplementary Fig. 1 and Supplementary Table 13): Barbary sheep (*Ammotragus lervia*), Equidae (*Equus* sp.), hare (*Lepus* sp.), hartebeest (*Alcelaphus buselaphus*), gazelle (*Gazella* sp.) and Rhinocerotidae. We also analysed two canid specimens (*Canis* sp. and *Vulpes vulpes*) to evaluate isotope values associated with a meat-based diet[51]. The faunal taxa were identified using traditional zooarchaeological methods[24] and zooarchaeology by mass spectrometry (ZooMS)[52,53] (Supplementary Information Section 3 and Supplementary Table 2). Through the use of these isotopic proxies, the focus of this work is to quantify this population's reliance on plants and determine whether their transition to a more plant-based diet mirrors that of the Levantine Natufian.

## Results and discussion

The measured $\delta^{66}Zn_{enamel}$, $\delta^{13}C$, $\delta^{15}N$ (bulk collagen and amino acids), $^{87}Sr/^{86}Sr_{enamel}$ and $\delta^{34}S_{collagen}$ for the humans and fauna from Taforalt are presented in Figs. 2–4, Extended Data Figs. 1–3, Supplementary Information Section 4 (Supplementary Figs. 3–12 and Supplementary Tables 5–12), Supplementary Tables 16–24 and Supplementary Fig. 16. The diets of the humans are discussed in Supplementary Information Section 5 (Supplementary Figs. 13–15 and Supplementary Table 13).

All the faunal remains from Taforalt exhibit similar $^{87}Sr/^{86}Sr_{enamel}$ values to the humans, which are close to the modern seawater value (~0.7092, Fig. 2b)[49]. Since the herbivores also exhibit this seawater value and given the $\delta^{13}C_{collagen}$ of human individuals, it is unlikely that the similar values in the humans indicate marine food consumption; rather, they probably reflect the values of the local geology, which is dominated by calcareous bedrocks[35] (expected to be between 0.707 and 0.709 (refs. 49,54); Fig. 2b). All of the other proxies used in this study ($\delta^{15}N_{collagen}$, $\delta^{13}C_{collagen}$, $\delta^{34}S_{collagen}$, $\delta^{15}N_{AA}$ and $\delta^{13}C_{AA}$) suggest the absence of regular aquatic food consumption (Supplementary Information Section 4).

Trophic level information was determined using three isotopic tracers: $\delta^{66}Zn_{enamel}$, $\delta^{15}N_{collagen}$ and the TP (C3) equation based on $\delta^{15}N_{Phe}$ and $\delta^{15}N_{Glu}$ values[38,55] (Fig. 3 and Supplementary Information Section 4). Given that the canids from Taforalt primarily have a meat-based diet[51] and that the plant portion of their diet consists of fruits, a resource showing exceptionally low zinc concentrations[56], their $\delta^{66}Zn_{enamel}$ values should be indicative of a carnivorous diet. For $\delta^{66}Zn_{enamel}$, the trophic level spacing ($TLS_{herbivores–canids}$) for all individuals is +0.62‰ and +0.70‰ if we consider only teeth formed post-weaning. This is close to that of Late Pleistocene sites in Laos (+0.63‰)[46,48] and higher than in a modern food web in Kenya (+0.40‰)[57]. However, the $TLS_{herbivores–canids}$ should be considered with precaution, given the small number of canid samples and the fact that the teeth might have been impacted by consumption of their mother's milk. When only considering human teeth formed after weaning, we found elevated adult human $\delta^{66}Zn_{enamel}$ values, which indicates a low trophic level (0.78 ± 0.07‰, $n_{samples} = 28$, $n_{individuals} = 12$), and these values are close to those of Taforalt herbivores ($n_{herbivores} = 7$, $n_{samples} = 20$, $TLS_{humans–herbivores} = +0.34‰$). The offset is 0.32‰ between the humans and the Barbary sheep, the primary source of game at Taforalt[24]. In contrast, this isotopic spacing is much higher at other Late Pleistocene sites such as Tam Pà Ling ($TLS_{humans–herbivores} = +0.48‰$) in Laos[46,48], Gabasa ($TLS_{humans–herbivores} = +0.85‰$) in Spain[43] and the medieval site of Rennes ($TLS_{humans–herbivores} = +0.63‰$) in France[44]. Furthermore, the $\delta^{66}Zn_{enamel}$ results from the Taforalt humans overlap with those from populations with historically documented cereal-based diets and not with those from populations that regularly consumed meat (Supplementary Fig. 17)[44], although this comparison does not consider baseline effects (Supplementary Information Section 4). As dietary zinc is likely to be primarily absorbed from animal sources[58,59], the minimal isotopic differences between Taforalt Iberomaurusians and herbivores at low trophic levels ($TLS_{humans–herbivores}$) and the elevated $\delta^{66}Zn_{enamel}$ values provide compelling evidence of substantial plant consumption. This, in turn, affirms their meat intake as well.

This interpretation of $\delta^{66}Zn_{enamel}$ data is also supported by the trophic level estimations obtained from the isotopic analyses of amino acids. The TP of adult humans, in tissues formed post-weaning, was found to vary between 2.2 and 2.6 with an average of 2.4 ± 0.2 ($n_{samples} = 9$). Thus, for the majority of individuals, plant resources were the primary source of dietary proteins. This finding highlights a substantial consumption of plant protein in a pre-agriculturist human population[60,61]. In particular, these TP values at Taforalt are similar to the TP values of Neolithic farmers from the Levant (Tell El Kerkh) (Fig. 1)[62]. Evidence for substantial plant consumption has also been found for two early modern humans (TP values of 2.5 and 2.6) at the Palaeolithic site of Buran Kaya in Crimea (Fig. 1), and this was similar to most of the associated canids at this site[60]. While the canids at Taforalt

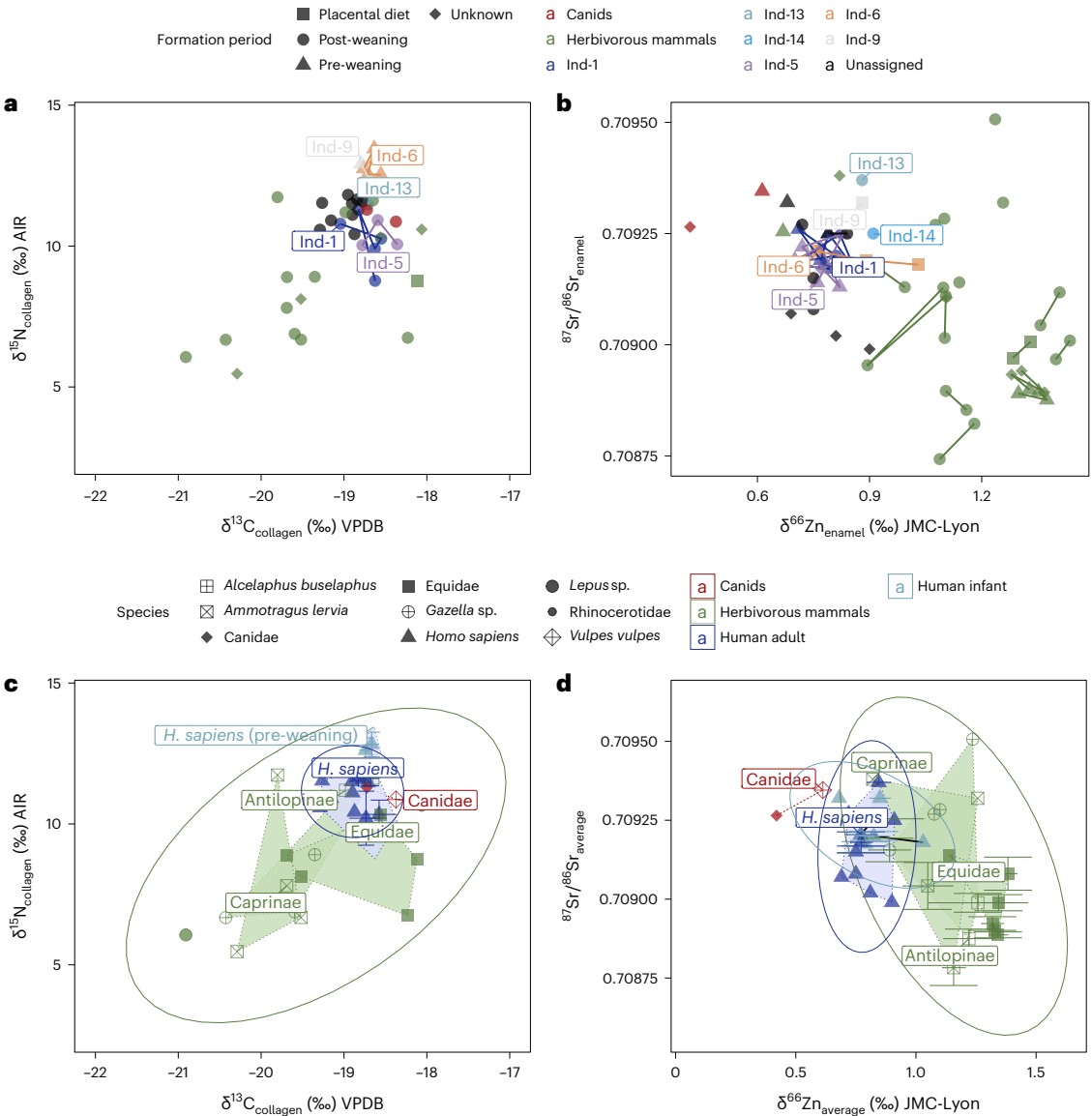

**Fig. 2 | Isotopic ratios of various elements from the human and faunal teeth/bone of Taforalt. a**, Carbon ($\delta^{13}$C) and nitrogen ($\delta^{15}$N) isotopic ratios from bulk collagen of dentine and bone samples. Each point corresponds to a sample; samples from the same individual are connected with a line. The typical analytical error is 0.1‰ for the two isotope systems. VPDB, Vienna PeeDee Belemnite; AIR, atmospheric N₂; Ind, individual. **b**, Zinc ($\delta^{66}$Zn) and strontium ($^{87}$Sr/$^{86}$Sr) isotopic ratios from enamel bioapatite. Each point corresponds to a sample; samples from the same individual are connected with a line. The typical analytical error is 0.05‰ for $\delta^{66}$Zn and $7 \times 10^{-6}$ for $^{87}$Sr/$^{86}$Sr. **c**, Carbon ($\delta^{13}$C) and nitrogen ($\delta^{15}$N)

isotopic ratios from bulk collagen of dentine and bone samples with associated 95% confidence ellipses. Each point corresponds to the average value of all samples coming from a single individual ($n_{individual}$ = 33; 44 samples in total); the error bars give the standard deviation for all the values from the same individual. **d**, Zinc ($\delta^{66}$Zn) and strontium ($^{87}$Sr/$^{86}$Sr) isotope ratios from enamel bioapatite with associated 95% confidence ellipses. Each point corresponds to the average value of all samples coming from a single individual ($n_{individual}$ = 33; 41 samples in total); the error bars give the standard deviation for all the values from the same individual.

may not be categorized as pure carnivores, their TP values remain notably high, especially in the case of the red fox (*Vulpes vulpes*) with a TP of 2.9. It is interesting to note that the canids' TP values surpass that of the humans, further supporting the idea that humans had a low reliance on animal protein. In particular, the TP results for individual 1 and an unassigned tooth closely resemble those of the herbivores (Fig. 4).

In addition to the small TLS$_{humans-herbivores}$ values for $\delta^{66}$Zn (0.34‰) and the low TP calculated by CSIA-AA, the $\delta^{15}$N$_{collagen}$ values between the humans and herbivores ($\Delta^{15}$N = 2.5‰) are smaller than those from other Upper Palaeolithic sites in Europe and Asia (Fig. 1) where animal proteins were the main dietary component (for example, Buran Kaya ($\Delta^{15}$N = 6.2‰), Oase ($\Delta^{15}$N = 10.8‰), Brno-Francouzska ($\Delta^{15}$ = 7.1‰) and Tianyuan ($\Delta^{15}$N = 6.4‰))[60,63,64]. Our results on the TLS between humans and herbivores are different from the $\Delta^{15}$N (TLS) observed by

Lee-Thorp et al. (+4.2‰) for Taforalt (Supplementary Information Section 4 and Supplementary Table 10)[28]. While their study focused on the Barbary sheep, the primary hunted faunal species at the site[24], it is important to consider that this species has a flexible diet[65], which might have influenced the accuracy of the TLS value due to potential differences in isotopic baselines. Our study demonstrates that this species had variable $\delta^{15}$N$_{collagen}$ values while having a stable trophic level of 2.1 ± 0.0 based on $\delta^{15}$N$_{AA}$ values (Supplementary Information Section 4). However, Lee-Thorp et al.[28] observed an absence of aquatic food consumption, aligning with our conclusions. In addition, Hedges and Reynard[33] found that $\delta^{15}$N$_{collagen}$ bulk-based diet reconstructions tend to overestimate animal protein intake by 60–80% when a nitrogen isotopic ratio enrichment of 3‰ or more is applied using the standard model for $\delta^{15}$N interpretation. This conclusion is supported by the association of a

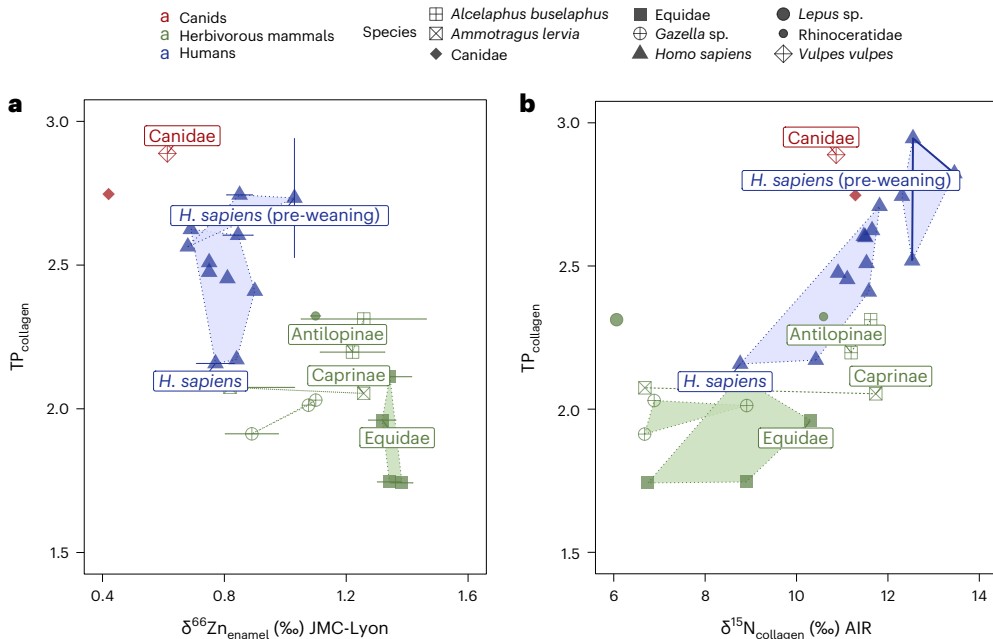

**Fig. 3 | Zinc and nitrogen isotope values versus TP. a,** Zinc ($\delta^{66}Zn$) isotope values versus the TP obtained from single amino acids (Supplementary Information Section 4). **b,** Nitrogen ($\delta^{15}N$) isotope values from bulk collagen versus the TP obtained from single amino acids. The TP was estimated from $\delta^{15}N_{Phe}$ and $\delta^{15}N_{Glu}$ values (Supplementary Information Section 2). Samples from the same human individual are connected with a line.

TLS of +3‰ with a plant intake of 50% (ref. 66) and a TLS of +4‰ among European Neolithic farmers with a meat intake of 40% (ref. 67). We should take into consideration that the plants eaten by humans could be more enriched in $\delta^{15}N$ than plants consumed by herbivores due to the effect of charring, which can increase their $\delta^{15}N_{collagen}$ by up to +1‰ (ref. 68). The $\delta^{15}N_{collagen}$ values observed in humans are probably affected by their consumption of these processed plants[69,70], compared with unprocessed forage plants consumed by herbivores. Our TLS estimations for Taforalt based on $\delta^{15}N_{bulk}$ of +4.2‰ and +2.5‰ could therefore suggest a plant food intake of about 50% in the Taforalt human diets. This is in agreement with our conclusions based on Zn isotope ratios and CSIA-AA, the presence of a variety of wild plants at the site[17] and the high prevalence of tooth caries and other periodontal diseases, which frequently exceeds those observed for hunter-gatherers, all suggesting a high consumption of fermentable starchy plants[4,17,71]. However, it must be stressed that the Taforalt humans studied here were not strict vegetalians, as isotopic offsets between the $\delta^{15}N$ and $\delta^{66}Zn$ herbivore and human values are documented and because zooarchaeological data indicate that animal protein was consumed. In particular, cut marks were observed on the faunal assemblage, mostly on Barbary sheep but also on gazelle, equid, large bovines and hartebeest[24]. These cut marks provide evidence of butchery and processing of animal remains, which directly supports the notion that animal protein was an integral part of the Taforalt human diet.

On the basis of multiple isotope proxies, we can also document an early weaning age for an infant (Ind-6) at Taforalt (Fig. 4 and Supplementary Information Section 5). The noticeable decline in TP, as calculated from the $\delta^{15}N$ values of single amino acids, among tissues with varying formation periods is particularly evident. The tissues, such as the deciduous second molar (TP = 2.8, formed over a span of approximately −0.34 to 1 year[72,73]) and the long bone (TP = 2.9, formed over the first year[72,73]), which have slower remodelling rates, exhibit higher TP values. In contrast, the rib bone, with its faster turnover rate[68] and TP of 2.5, is likely to have recorded dietary information much closer to the time of the individual's death (6–12 months[73]) (Supplementary Information Sections 2–4). This pattern of decreasing TP values strongly suggests a rapid transition in the individual's diet, with the introduction of adult foods playing a substantial role in this dietary shift[74] (Fig. 4).

This is evidence that weaning was initiated before 1 year of age and possibly with plant-based foods, since we observed a clear decrease in this individual's TP (2.8 to 2.5). Unlike at other sites[75], we do not see a clear weaning pattern in the $\delta^{66}Zn$ results when comparing different teeth of a single individual or at the population level (Supplementary information Section 5 and Supplementary Fig. 13). This observation may be due to a sample bias, as the limited sample size per individual prevented the tracing of potential weaning patterns. Alternatively, this could be attributed to the early introduction of solid foods in infant diets. The adoption of a starchy diet in Taforalt may have facilitated early weaning, a pattern commonly associated with the transition to agriculture due to the availability of soft and digestible foods such as cereals. However, early weaning can result in increased stress and mortality for infants[76]. This contrasts with hunter-gatherer societies, where extended breastfeeding periods are the norm due to the limited availability of suitable weaning foods[4,77]. These observations suggest that changes in diet and lifestyle in the Iberomaurusians from Taforalt might have had important impacts on infant feeding practices. However, it is clear that additional detailed analyses are needed to fully understand this weaning pattern on a larger scale.

According to the broad-spectrum and dietary breadth models, a reduction in the availability of large to medium-sized game animals often leads to increased foraging for previously overlooked resources such as lagomorphs and small birds and an increased exploitation of wild plants[10,78]. This hypothesis has been commonly applied to explain the emergence of farming in Southwest Asia, where the Natufian hunter-gatherers, initially reliant on small to medium-sized ungulates, adapted their subsistence strategy due to ecological pressure on these animals[79]. As a result, they gradually diversified their diet by incorporating a broader range of food resources, including wild plants. This may have been the case for the Taforalt population, as evidenced by the high incidence and diversity of charred macrobotanical plant remains found in the Grey Series level[17]. The prevalence of caries in the human teeth in burials also suggests a substantial reliance on highly cariogenic wild plant foods such as sweet acorns, pine nuts and some legumes. Furthermore, the presence of grinding stones in the same deposits suggests plant processing, which is possible evidence that the nuts and acorns

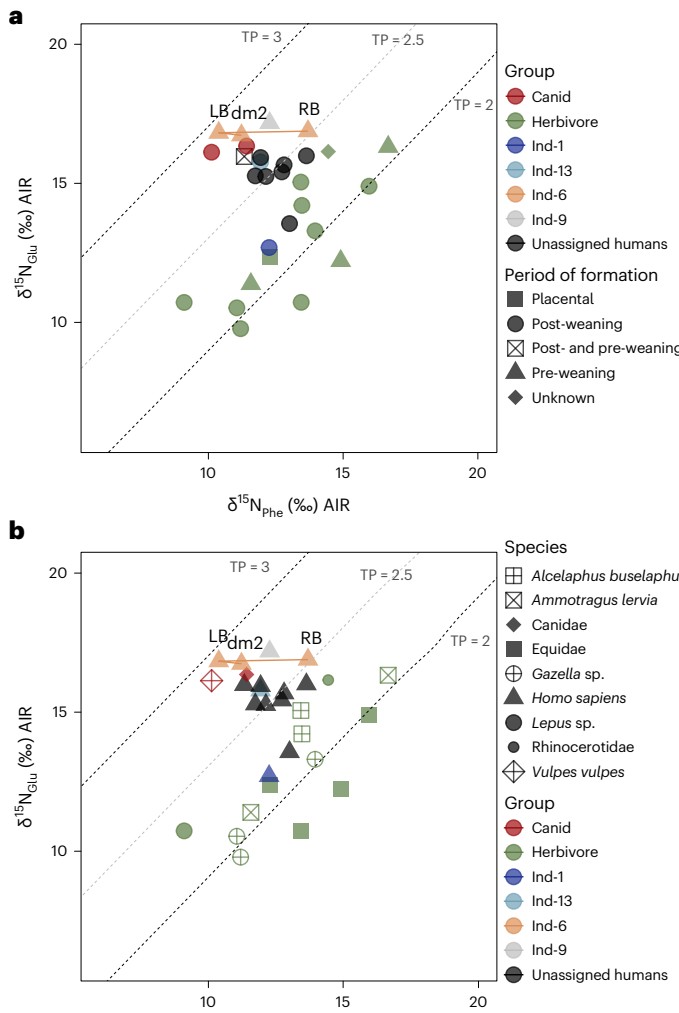

**Fig. 4 | Measured $\delta^{15}N_{Phe}$ and $\delta^{15}N_{Glu}$ values on human and faunal collagen from Taforalt. a**, Values according to the formation time of the sample. **b**, Values according to the species. The dashed black lines indicate approximately the theoretical TP of herbivores (TP = 2) and carnivores (TP = 3). The dashed grey line is the intermediate (TP = 2.5). Samples from the same human individual are connected with a line. RB, rib bone; LB, long bone; dm2, deciduous second molar.

were ground into flour or meal[17]. While the removal of the central upper incisors was a prevalent practice among 90% of the Taforalt population and is common among the Iberomaurusians[17,79], it is important to note that this practice is not linked to oral pathology. Instead, it may have impacted the functional use of teeth for mastication[79].

The $\delta^{13}C$ amino acid results presented here indicate that most of the humans and herbivores consumed $C_3$ plants (Supplementary Information Section 4), which is the photosynthetic pathway of all the edible plant species found at Taforalt. It is likely that most of the wild plants were collected during autumn, such as acorns, while pulses were harvested from late spring to summer[17]. The inhabitants probably stored plants, which would have ensured consistent food staples throughout the year[17].

These findings suggest a notable increase in the reliance on plant resources by the Taforalt population. While there is no evidence of a decline in Barbary sheep (the main hunted species during the Iberomaurusian period[24]) at the site, it is plausible that the seasonal availability of these species and other ungulates at the site influenced the access to meat proteins through the year. The mortality age of Barbary sheep and gazelle points to hunting activities occurring between spring

and early summer[24]. Simultaneously, the increased abundance of wild plant resources in the inhabitants' environment may have played a role in their subsistence strategy behaviour. Land snails might have been consumed seasonally too[27]. The consumption of wild plant resources (such as acorns) may explain why most of the Iberomaurusian sites were located in the coastal Mediterranean forest regions of Northwest Africa. However, more Iberomaurusian sites need to be studied to confirm this hypothesis.

## Conclusion

Our study highlights the importance of the Taforalt population's dietary reliance on plants, while animal resources were consumed in a lower proportion than at other Upper Palaeolithic sites with available isotopic data. The potential early weaning of infants at Taforalt reinforces the notion of a plant-based food focus for the population, potentially extending to the primary source of nutrition for infants. However, it is crucial to acknowledge that further comprehensive investigations are necessary to fully understand these findings and their implications. Evidence of intensive exploitation of wild plants at the end of the Late Pleistocene is also documented in the Near East with the Natufian hunter-gatherers, who developed cultivation and became some of the earliest agriculturists. In that region, it is believed that the Younger Dryas climatic deterioration in the early Holocene (11,000–10,300 uncal BP) was the major trigger for systematic cultivation in response to the reduction of the vegetal cover and, consequently, the availability of exploited wild plants[8,80]. Although the Natufian and Iberomaurusian populations had broad similarities regarding the preconditions for the emergence of food production (intensive plant consumption and increased sedentism) and genetic connections (63% of shared genes between Natufian and Iberomaurusian individuals)[15], these factors did not lead in North Africa to a similar local development of agriculture and farming despite the high reliance on plants as a food staple during the Later Stone Age. While the origin of this dissimilarity is still open to debate, the Younger Dryas cooling phase might have reduced the abundance of plant resources, which could explain why Iberomaurusian sites became less occupied during the period[81].

## Methods
### Zinc and strontium isotope analyses
Zinc and strontium were extracted from tooth enamel. The sampling strategy details are given in Supplementary Information Section 7 and Supplementary Tables 16 and 17. To explore potential dietary variations during tooth development in humans, we used a specific sampling approach involving the collection of enamel samples from both the upper (top) and lower (bottom) portions of the tooth crown (Supplementary Table 16 and Supplementary Information Section 7). The extended formation period of enamel means that it captures dietary information from various stages of an individual's life. By sampling both the top and bottom parts of the crown, we aimed to examine potential dietary changes and isotopic variations that could provide insights into the individual's nutritional history over time. It is important to acknowledge that this approach was not uniformly applied to all teeth due to the varying condition of the teeth themselves. Some teeth presented a challenge due to insufficient enamel (the material we also used for Sr analyses) extending along the height of the crown. This deficiency in preserved enamel was primarily attributed to wear and abrasion, affecting teeth such as SEVA 35976, 35968, 35959 and 35973. Enamel samples were collected using a precise and minimally destructive method. We used a diamond-tipped burr to mechanically extract a powder from multiple parts along the height of the tooth crown. The amount of enamel collected ranged from 5 to 20 mg, ensuring that we captured sufficient material for isotopic analyses while minimizing the destructive sampling as much as possible.

For faunal teeth, particularly those of herbivores, we implemented a multisampling method to account for potential isotopic variations

(Supplementary Fig. 2 and Supplementary Table 17). Faunal taxa such as equids, Barbary sheep, hartebeest and rhinoceros with teeth of sufficient height provided an opportunity for multisampling. We collected samples from the tooth crown at multiple points, approximately three to five times along its height (5 to 20 mg), to track isotopic changes from the estimated initiation of enamel formation to the completion of the crown (Supplementary Table 17). To account for the influence of nursing signals on isotopic signatures, we also estimated the age of enamel formation for the faunal species (more details are provided in Supplementary Information Section 3).

The Zn and Sr samples were prepared from separate enamel powder aliquots. For each column chromatography batch, we included a matrix-matched powder standard with known isotopic composition (SRM 1486 for $^{87}Sr/^{86}Sr$ and SRM 1400 for $\delta^{66}Zn$) and a procedural blank to monitor contamination. The purification step of zinc was achieved using the ion exchange method. The samples were dissolved in 1 ml of HCl, evaporated and then dissolved in 1.5 M HBr. Zinc was purified following the protocol adapted from Moynier et al.[82] and described in Jaouen et al.[57]. The purification of Zn was achieved on $HNO_3$-cleaned AG1X8 resin conditioned with 1.5 M HBr, which was also used for matrix elution. For the Zn elution, we used 5 ml of 3% $HNO_3$. The Zn isotope ratios were measured using a Thermo Neptune Multicollector inductively coupled plasma mass spectrometer at Max Planck Institute for Evolutionary Anthropology (Leipzig, Germany). Each enamel sample was analysed in duplicate. The average values were calculated from the duplicate measurements with a standard deviation of 0.0‰ to 0.05‰. Standard sample bracketing and Cu doping procedures were implemented to enhance measurement accuracy[83]. All the zinc data are reported using the δ notation in per mille relative to the standard JMC-Lyon. The typical analytical standard deviation of the bracketing standard for $\delta^{66}Zn$ is 0.00‰ to 0.03‰. The sample Zn concentrations were estimated by measuring the $^{64}Zn$ signal intensity of three solutions with known concentrations (150 ppb, 300 ppb and 600 ppb) relative to the sample concentrations.

For strontium, the purification was achieved using the protocol adapted from Copeland et al.[84]. The enamel was dissolved in 2 ml of $HNO_3$, evaporated and subsequently re-dissolved in 3 M $HNO_3$. Strontium was purified using Sr.spec.TM resin and eluted from the resin with 1.5 ml of ultrapure deionized Milli Q water before evaporation. The dry samples were dissolved in 2 ml of 3% $HNO_3$ for the isotopic measurements. The Sr isotopic ratios were measured on the Thermo Neptune Multicollector inductively coupled plasma mass spectrometer. The interference from Rb with $^{87}Sr/^{86}Sr$ measurements was corrected on the basis of the repeated analysis of the standard (SRM 987). The Sr concentrations were calculated by measuring the $^{88}Sr$ signal intensity ($V$) of three diluted solutions with known concentrations (100 ppb, 400 ppb and 700 ppb).

### Carbon and nitrogen isotope analyses

For collagen extraction, both human and faunal tooth dentine and bone samples, weighing between ~200 and 500 mg, were carefully removed using a diamond-tipped burr. Collagen was extracted following the protocol described in Talamo and Richards[85]. Dentine and bone were cleaned by abrasion and then demineralized in HCl for several weeks at 4 °C. The samples were immersed in NaOH for 30 minutes. The purified solution containing collagen was rinsed and soaked again in HCl. The insoluble collagen was solubilized in HCl (pH 3) at 75 °C for 20 h. The supernatant containing the collagen was filtered with Ezee filters and then ultra-filtrated to collect the >30 kDa collagen molecules, frozen for 48 h and lyophilized.

The C and N isotopic ratios were obtained using a Thermo Finnigan Flash EA coupled to a Delta V isotope ratio mass spectrometer. Carbon and nitrogen stable isotope ratios were measured relative to Vienna PeeDee Belemnite and atmospheric $N_2$, respectively. The analytical

errors of 0.1‰ and 0.2‰ for $\delta^{13}C$ and $\delta^{15}N$ were determined by the repeated analysis of internal and international standards (IAEA-N1, IAEA-N2, MET and MRG).

### Sulfur isotope analyses

The sulfur isotopic analyses were conducted by Isoanalytical. Eight milligrams of collagen were loaded into tin capsules in addition to the NBS-1577B (bovine liver powder) standard used as a control. The samples and the standards were measured using an EA-IRMS (ANCA-GSL/20-20, Europa Scientific), and the isotope ratios are reported to the international reference standard Vienna-Canyon Diablo Troilite.

### CSIA-AA

CSIA-AA was conducted in the commercial Stable Isotope Facility of the University of Davis, California. The samples were analysed following the protocol developed by Yarnes and Herszage[86]. The detailed procedure is described at https://stableisotopefacility.ucdavis.edu/compound-specific-13c-15n-analysis-amino-acids-gc-c-irms.

### ZooMS

Faunal remains were taxonomically identified using both traditional zooarchaeology[24] and ZooMS[52] (Supplementary Tables 2 and 17). ZooMS is a minimally destructive proteomic method that focuses on unidentifiable bone fragments. Through the analysis of bone collagen protein type I, it provides a taxonomic identification based on protein amino acid sequence variation. ZooMS analysis followed extraction protocols detailed elsewhere[52,53,87]. Selected faunal specimens from Taforalt were analysed through ZooMS and sampled using pliers (10–20 mg). Soluble collagen was first extracted through incubation in 100 µl of 50 mM ammonium bicarbonate ($NH_4HCO_3$, pH 8.0) buffer at 65 °C for 1 h. To improve the taxonomic identity obtained from soluble collagen, each bone sample was demineralized in 250 µl of 0.6 M hydrochloric acid (HCl) at 4 °C for 20 h. The samples were then centrifuged for 1 min at 9,520 $g$, and the supernatant was removed. The demineralized collagen was rinsed three times in 200 µl of 50 mM ammonium bicarbonate to be neutralized to pH 8, and 100 µl of 50 mM ammonium bicarbonate was added to each sample. Next, the samples were incubated at 65 °C for 1 h. Then, 50 µl of the resulting supernatant was digested with trypsin (0.5 µg µl$^{-1}$, Promega) at 37 °C overnight, acidified using 1 µl of trifluoroacetic acid (20% TFA) and cleaned on C18 ZipTips (Thermo Scientific). Digested peptides were spotted in triplicate on a MALDI Bruker plate with the addition of α-cyano-4-hydroxycinnamic acid (Sigma) matrix. MALDI-TOF-MS analysis was conducted at the Fraunhofer IZI in Leipzig, Germany, using an autoflex speed LRF MALDI-TOF (Bruker) in reflector mode, with positive polarity, with matrix suppression up to 590 Da and collected in the mass-to-charge range 700–3,500 $m/z$. Triplicates were merged for each sample, and taxonomic identifications were made manually through peptide marker mass identification in comparison to a database of peptide marker series for medium- to larger-sized Pleistocene mammalian species[53,88]. We performed laboratory blanks alongside the samples to assess any potential contamination by non-endogenous peptides. These remained empty of collagenous peptides, excluding the possibility of modern laboratory or storage contamination.

### Statistical analyses

Statistical analyses and data visualization were performed using R software (R version 4.3.1). The function stat_ellipse in the R package ggplot2 was used to produce ellipses with 95% confidence (https://cran.r-project.org/web/packages/ggplot2/index.html).

### Reporting summary

Further information on research design is available in the Nature Portfolio Reporting Summary linked to this article.

## Data availability

All data are available in the manuscript and supplementary materials.

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

## Acknowledgements

This study was funded by the Max Planck Society and the ERC starting grant ARCHEIS (no. 803676). J.M. and N.B. thank the DFG for salary support (project nos 505905610 and 378496604, respectively). V.S.-M. was supported by a Fyssen Foundation postdoctoral fellowship (2023–2025). We thank the National Institute for Archaeological Sciences and Heritage (INSAP, Morocco). We thank the IZI Fraunhofer (Leipzig, Germany), S. Kalkhof and J. Schmidt for providing access to the MALDI-TOF MS instrument for ZooMS analysis of the faunal remains. We thank Z. Tsegai for helping in scanning the teeth. We thank the multimedia department at MPI-EVA.

## Author contributions

Z.M. and K.J. designed the research. Z.M., A.B. and K.J. selected the sample material. A.B. and E.T. provided contextual information. H.T. scanned the human teeth. E.T. and G.M.S. identified the faunal specimens. V.S.-M. performed the ZooMS analyses. J.-J.H. provided technical support. Z.M., K.J., J.M., N.B., M.T. and S.S. performed the isotopic analyses. B.T.F. contributed to interpreting the isotopic data. Z.M., K.J., J.M. and N.B. analysed and interpreted the data and wrote the manuscript with input from all the authors.

## FundingInformation

## Competing interests

The authors declare no competing interests.

## Additional information

**Extended data** is available for this paper at https://doi.org/10.1038/s41559-024-02382-z.

**Correspondence and requests for materials** should be addressed to Zineb Moubtahij.

[1]Max Planck Institute for Evolutionary Anthropology, Leipzig, Germany. [2]Géosciences Environnement Toulouse, UMR 5563, CNRS, Observatoire Midi Pyrénées, Toulouse, France. [3]Goethe University Frankfurt, Institute of Geosciences, Frankfurt am Main, Germany. [4]IsoTROPIC Research Group, Max Planck Institute for Geoanthropology, Jena, Germany. [5]PACEA, UMR 5199, CNRS, Université de Bordeaux, Ministère de la Culture, Pessac, France. [6]CBMN, UMR 5248 and Bordeaux Proteome Platform, Bordeaux INP, CNRS, Université de Bordeaux, Bordeaux, France. [7]School of Anthropology and Conservation, University of Kent, Canterbury, UK. [8]Chaire de Paléoanthropologie, CIRB (UMR 7241–U1050), Collège de France, Paris, France. [9]Institut National des Sciences de l'Archéologie et du Patrimoine, Origin and Evolution of Homo Sapiens Cultures, Rabat, Morocco. [10]Department of Archaeogenetics, Max Planck Institute for Evolutionary Anthropology, Leipzig, Germany. [11]Monrepos Archaeological Research Centre and Museum for Human Behavioural Evolution, LEIZA, Neuwied, Germany. ✉e-mail: zinebmoubtahij3@gmail.com

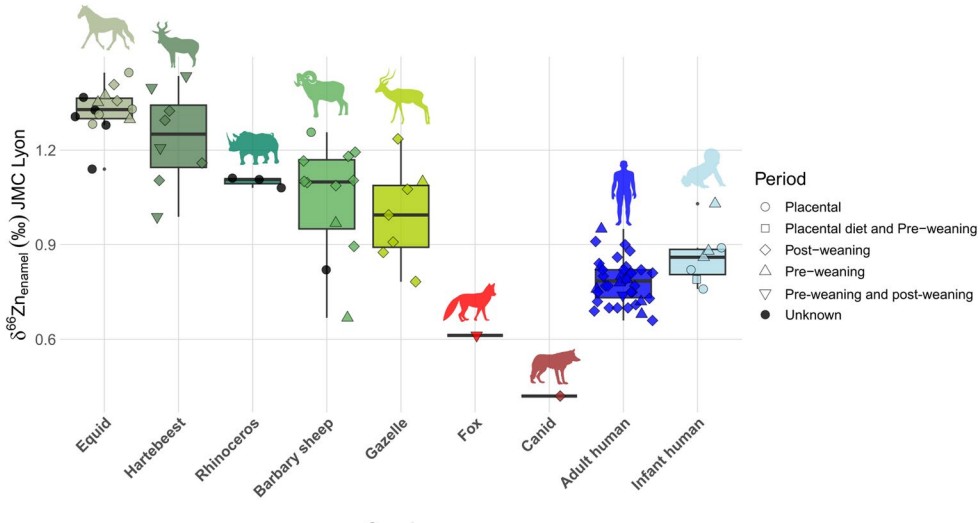

**Extended Data Fig. 1 | Zinc isotope values in tooth enamel of various faunal species and the humans analyzed in this study ($n_{individual}$ = 33, 41 samples in total).** Each point represents the mean value of a tooth. Boxes correspond to the median (centre line) and the first and third quartiles, while whiskers indicate the minimum and maximum values.

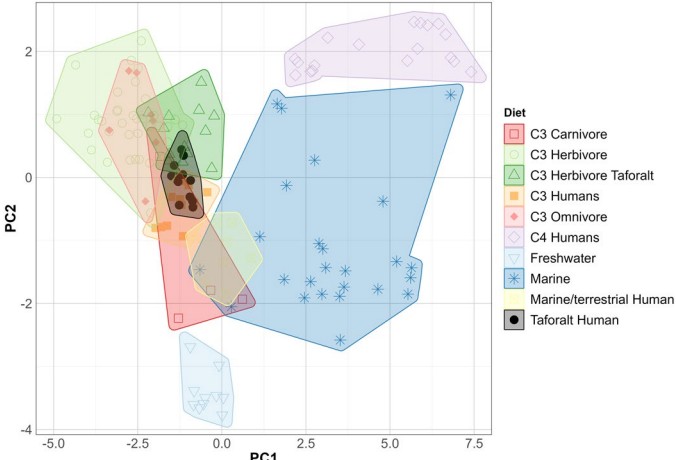

**Extended Data Fig. 2 | Carbon isotope values in amino acids of the humans and fauna from Taforalt.** PCA plot of the $\delta^{13}C_{Ala}$, $\delta^{13}C_{Asp}$, $\delta^{13}C_{Glx}$, $\delta^{13}C_{Gly}$, $\delta^{13}C_{Gly\text{-}Phe}$, $\delta^{13}C_{Lys}$, $\delta^{13}C_{Phe}$, $\delta^{13}C_{Pro}$, $\delta^{13}C_{val}$, $\delta^{13}C_{Val\text{-}Phe}$ measurement with bulk $\delta^{13}C$ and $\delta^{15}N$ values from Taforalt and published data[41,89–95].

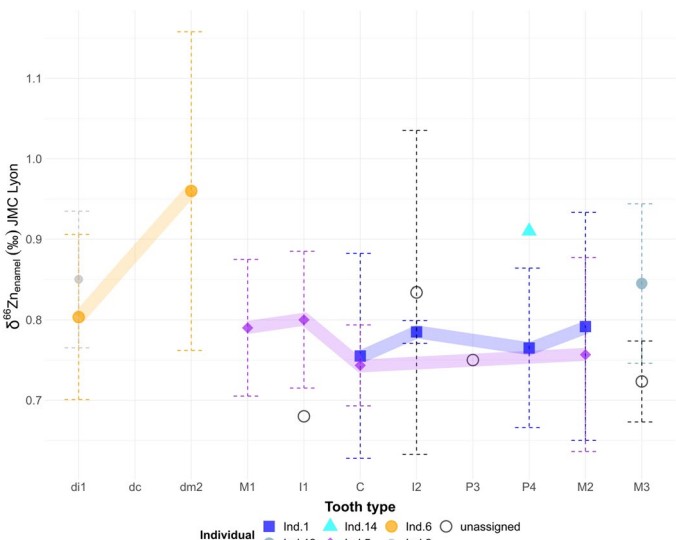

**Extended Data Fig. 3 | Average $\delta^{66}$Zn values per dental specimen of different individuals analyzed in this study.** Different parts of the tooth crowns were sampled so each point represents an average zinc isotope value of different samplings taken from the same tooth. Error bars are the standard deviation of different samples from the same dental specimen. ($n_{individual}$= 17, 45 samples in total).

# Reporting Summary

## Statistics

For all statistical analyses, confirm that the following items are present in the figure legend, table legend, main text, or Methods section.

| n/a | Confirmed | |
|---|---|---|
| ☐ | ☒ | The exact sample size (*n*) for each experimental group/condition, given as a discrete number and unit of measurement |
| ☐ | ☒ | A statement on whether measurements were taken from distinct samples or whether the same sample was measured repeatedly |
| ☐ | ☒ | The statistical test(s) used AND whether they are one- or two-sided<br>*Only common tests should be described solely by name; describe more complex techniques in the Methods section.* |
| ☒ | ☐ | A description of all covariates tested |
| ☒ | ☐ | A description of any assumptions or corrections, such as tests of normality and adjustment for multiple comparisons |
| ☐ | ☒ | A full description of the statistical parameters including central tendency (e.g. means) or other basic estimates (e.g. regression coefficient) AND variation (e.g. standard deviation) or associated estimates of uncertainty (e.g. confidence intervals) |
| ☐ | ☒ | For null hypothesis testing, the test statistic (e.g. *F*, *t*, *r*) with confidence intervals, effect sizes, degrees of freedom and *P* value noted<br>*Give P values as exact values whenever suitable.* |
| ☒ | ☐ | For Bayesian analysis, information on the choice of priors and Markov chain Monte Carlo settings |
| ☒ | ☐ | For hierarchical and complex designs, identification of the appropriate level for tests and full reporting of outcomes |
| ☒ | ☐ | Estimates of effect sizes (e.g. Cohen's *d*, Pearson's *r*), indicating how they were calculated |

*Our web collection on statistics for biologists contains articles on many of the points above.*

## Software and code

Policy information about availability of computer code

| Data collection | Thermo Scientific Mass Spectrometry Software |
|---|---|
| Data analysis | Results visualization and plot generation: RStudio (R version 4.2.2), ggplot2. Statistical analyses were conducted using R software. Isotopic niches were estimated using SIBER − Stable Isotope Bayesian ellipses in R . |

For manuscripts utilizing custom algorithms or software that are central to the research but not yet described in published literature, software must be made available to editors and reviewers. We strongly encourage code deposition in a community repository (e.g. GitHub). See the Nature Portfolio guidelines for submitting code & software for further information.

## Data

Policy information about availability of data

All manuscripts must include a data availability statement. This statement should provide the following information, where applicable:
- Accession codes, unique identifiers, or web links for publicly available datasets
- A description of any restrictions on data availability
- For clinical datasets or third party data, please ensure that the statement adheres to our policy

All the data generated in this study are included in this article and in the supplementary information.

# Research involving human participants, their data, or biological material

Policy information about studies with human participants or human data. See also policy information about sex, gender (identity/presentation), and sexual orientation and race, ethnicity and racism.

| Reporting on sex and gender | N/A |
|---|---|
| Reporting on race, ethnicity, or other socially relevant groupings | N/A |
| Population characteristics | N/A |
| Recruitment | N/A |
| Ethics oversight | N/A |

Note that full information on the approval of the study protocol must also be provided in the manuscript.

# Field-specific reporting

Please select the one below that is the best fit for your research. If you are not sure, read the appropriate sections before making your selection.

☐ Life sciences  ☐ Behavioural & social sciences  ☒ Ecological, evolutionary & environmental sciences

For a reference copy of the document with all sections, see nature.com/documents/nr-reporting-summary-flat.pdf

# Life sciences study design

All studies must disclose on these points even when the disclosure is negative.

| Sample size | N/A |
|---|---|
| Data exclusions | N/A |
| Replication | N/A |
| Randomization | N/A |
| Blinding | N/A |

# Behavioural & social sciences study design

All studies must disclose on these points even when the disclosure is negative.

| Study description | N/A |
|---|---|
| Research sample | N/A |
| Sampling strategy | N/A |
| Data collection | N/A |
| Timing | N/A |
| Data exclusions | N/A |
| Non-participation | N/A |
| Randomization | N/A |

# Ecological, evolutionary & environmental sciences study design

All studies must disclose on these points even when the disclosure is negative.

| | |
|---|---|
| Study description | We investigated the diet of Later Stone Age human populations in Taforalt, Morocco, using a comprehensive multi-isotopic analysis. Dental enamel underwent zinc (δ66Zn) and strontium (87Sr/86Sr) analyses, while dentin and bone collagen were subjected to carbon (δ13C), nitrogen (δ15N), sulfur (δ34S), and single amino acid isotope analyses for both archaeological humans and fauna. |
| Research sample | We carefully sampled bone and tooth samples from human remains excavated at the Taforalt archaeological site's Sector 10.We analyzed faunal remains from both Sector 10 and Sector 8 of the Taforalt site. |
| Sampling strategy | We sampled tooth enamel for analyzing zinc and strontium isotopes. Additionally, we extracted collagen for stable isotopic analyses of carbon and nitrogen, conducted on both bulk collagen and amino acids. This sampling strategy was conducted on both human and faunal samples. |
| Data collection | Mass Spectrometry |
| Timing and spatial scale | The analyses performed were not time dependent. The samples were collected during October 2019 from the Institut National des Sciences de l'Archéologie et du Patrimoine (INSAP) in Rabat, Morocco |
| Data exclusions | Samples that did not meet the quality qriteria were excluded from the data interpretation. |
| Reproducibility | All isotopic measurements included the analyses of internationally recognised standards and sample replicate analyses. |
| Randomization | Does not apply to this type of study. |
| Blinding | Does not apply to this type of study. |

Did the study involve field work?  ☐ Yes  ☒ No

## Field work, collection and transport

| | |
|---|---|
| Field conditions | N/A |
| Location | Institut National des sciences de l'archéologie et du patrimoine, Rabat, Morocco |
| Access & import/export | The permit for exporting and sampling the human and faunal samples were granted by Institut National des Sciences de l'Archéologie et du Patrimoine (INSAP) in Rabat, Morocco. The samples were analyzed at the Max Planck Institute for Evolutionary Anthropology in Leipzig. The samples were transported in accordance with the permit (N°02/2019-2020-232) obtained for their exportation. |
| Disturbance | N/A |

# Reporting for specific materials, systems and methods

We require information from authors about some types of materials, experimental systems and methods used in many studies. Here, indicate whether each material, system or method listed is relevant to your study. If you are not sure if a list item applies to your research, read the appropriate section before selecting a response.

## Materials & experimental systems

| n/a | Involved in the study |
|---|---|
| ☒ | Antibodies |
| ☒ | Eukaryotic cell lines |
| ☐ | ☒ Palaeontology and archaeology |
| ☒ | Animals and other organisms |
| ☒ | Clinical data |
| ☒ | Dual use research of concern |
| ☒ | Plants |

## Methods

| n/a | Involved in the study |
|---|---|
| ☒ | ChIP-seq |
| ☒ | Flow cytometry |
| ☒ | MRI-based neuroimaging |

# Antibodies

| | |
|---|---|
| Antibodies used | N/A |
| Validation | *Describe the validation of each primary antibody for the species and application, noting any validation statements on the manufacturer's website, relevant citations, antibody profiles in online databases, or data provided in the manuscript.* |

# Eukaryotic cell lines

Policy information about cell lines and Sex and Gender in Research

| | |
|---|---|
| Cell line source(s) | N/A |
| Authentication | N/A |
| Mycoplasma contamination | N/A |
| Commonly misidentified lines (See ICLAC register) | N/A |

# Palaeontology and Archaeology

| | |
|---|---|
| Specimen provenance | Archaeological samples were excavated from the archaeological sites of Taforalt, Morocco. Additional information on the provenance of the samples is provided in the supplementary information. |
| Specimen deposition | Institut National des Sciences de l'Archéologie et du Patromoine (INSAP), Rabat, Morocco |
| Dating methods | No new dates are provided. Reference for the previously published dates are provided in the supplementary information. |

☒ Tick this box to confirm that the raw and calibrated dates are available in the paper or in Supplementary Information.

| | |
|---|---|
| Ethics oversight | Permits for sampling and analyses of the archaeological material were obtained from the  Institut National des Sciences de l'Archèologie et du Patrimoine (Rabat, Morocco). Ethical approval was not required for this study. |

Note that full information on the approval of the study protocol must also be provided in the manuscript.

# Animals and other research organisms

Policy information about studies involving animals; ARRIVE guidelines recommended for reporting animal research, and Sex and Gender in Research

| | |
|---|---|
| Laboratory animals | N/A |
| Wild animals | N/A |
| Reporting on sex | N/A |
| Field-collected samples | N/A |
| Ethics oversight | N/A |

Note that full information on the approval of the study protocol must also be provided in the manuscript.

# Clinical data

Policy information about clinical studies
All manuscripts should comply with the ICMJE guidelines for publication of clinical research and a completed CONSORT checklist must be included with all submissions.

| | |
|---|---|
| Clinical trial registration | N/A |
| Study protocol | N/A |
| Data collection | N/A |
| Outcomes | N/A |

# Dual use research of concern

Policy information about dual use research of concern

## Hazards

Could the accidental, deliberate or reckless misuse of agents or technologies generated in the work, or the application of information presented in the manuscript, pose a threat to:

| No | Yes | |
|----|-----|--|
| ☒ | ☐ | Public health |
| ☒ | ☐ | National security |
| ☒ | ☐ | Crops and/or livestock |
| ☒ | ☐ | Ecosystems |
| ☒ | ☐ | Any other significant area |

## Experiments of concern

Does the work involve any of these experiments of concern:

| No | Yes | |
|----|-----|--|
| ☒ | ☐ | Demonstrate how to render a vaccine ineffective |
| ☒ | ☐ | Confer resistance to therapeutically useful antibiotics or antiviral agents |
| ☒ | ☐ | Enhance the virulence of a pathogen or render a nonpathogen virulent |
| ☒ | ☐ | Increase transmissibility of a pathogen |
| ☒ | ☐ | Alter the host range of a pathogen |
| ☒ | ☐ | Enable evasion of diagnostic/detection modalities |
| ☒ | ☐ | Enable the weaponization of a biological agent or toxin |
| ☒ | ☐ | Any other potentially harmful combination of experiments and agents |

# Plants

| Seed stocks | N/A |
|-------------|-----|
| Novel plant genotypes | N/A |
| Authentication | N/A |

# ChIP-seq

## Data deposition

☐ Confirm that both raw and final processed data have been deposited in a public database such as GEO.

☐ Confirm that you have deposited or provided access to graph files (e.g. BED files) for the called peaks.

| Data access links | *For "Initial submission" or "Revised version" documents, provide reviewer access links. For your "Final submission" document, provide a link to the deposited data.* |
|-------------------|--------|
| May remain private before publication. | |
| Files in database submission | *Provide a list of all files available in the database submission.* |
| Genome browser session | *Provide a link to an anonymized genome browser session for "Initial submission" and "Revised version" documents only, to enable peer review. Write "no longer applicable" for "Final submission" documents.* |
| (e.g. UCSC) | |

## Methodology

| | |
|---|---|
| Replicates | *Describe the experimental replicates, specifying number, type and replicate agreement.* |
| Sequencing depth | *Describe the sequencing depth for each experiment, providing the total number of reads, uniquely mapped reads, length of reads and whether they were paired- or single-end.* |
| Antibodies | *Describe the antibodies used for the ChIP-seq experiments; as applicable, provide supplier name, catalog number, clone name, and lot number.* |
| Peak calling parameters | *Specify the command line program and parameters used for read mapping and peak calling, including the ChIP, control and index files used.* |
| Data quality | *Describe the methods used to ensure data quality in full detail, including how many peaks are at FDR 5% and above 5-fold enrichment.* |
| Software | *Describe the software used to collect and analyze the ChIP-seq data. For custom code that has been deposited into a community repository, provide accession details.* |

## Flow Cytometry

### Plots

Confirm that:

☐ The axis labels state the marker and fluorochrome used (e.g. CD4-FITC).

☐ The axis scales are clearly visible. Include numbers along axes only for bottom left plot of group (a 'group' is an analysis of identical markers).

☐ All plots are contour plots with outliers or pseudocolor plots.

☐ A numerical value for number of cells or percentage (with statistics) is provided.

### Methodology

| | |
|---|---|
| Sample preparation | *Describe the sample preparation, detailing the biological source of the cells and any tissue processing steps used.* |
| Instrument | *Identify the instrument used for data collection, specifying make and model number.* |
| Software | *Describe the software used to collect and analyze the flow cytometry data. For custom code that has been deposited into a community repository, provide accession details.* |
| Cell population abundance | *Describe the abundance of the relevant cell populations within post-sort fractions, providing details on the purity of the samples and how it was determined.* |
| Gating strategy | *Describe the gating strategy used for all relevant experiments, specifying the preliminary FSC/SSC gates of the starting cell population, indicating where boundaries between "positive" and "negative" staining cell populations are defined.* |

☐ Tick this box to confirm that a figure exemplifying the gating strategy is provided in the Supplementary Information.

## Magnetic resonance imaging

### Experimental design

| | |
|---|---|
| Design type | *Indicate task or resting state; event-related or block design.* |
| Design specifications | *Specify the number of blocks, trials or experimental units per session and/or subject, and specify the length of each trial or block (if trials are blocked) and interval between trials.* |
| Behavioral performance measures | *State number and/or type of variables recorded (e.g. correct button press, response time) and what statistics were used to establish that the subjects were performing the task as expected (e.g. mean, range, and/or standard deviation across subjects).* |

## Acquisition

| | |
|---|---|
| Imaging type(s) | *Specify: functional, structural, diffusion, perfusion.* |
| Field strength | *Specify in Tesla* |
| Sequence & imaging parameters | *Specify the pulse sequence type (gradient echo, spin echo, etc.), imaging type (EPI, spiral, etc.), field of view, matrix size, slice thickness, orientation and TE/TR/flip angle.* |
| Area of acquisition | *State whether a whole brain scan was used OR define the area of acquisition, describing how the region was determined.* |

Diffusion MRI ☐ Used ☐ Not used

## Preprocessing

| | |
|---|---|
| Preprocessing software | *Provide detail on software version and revision number and on specific parameters (model/functions, brain extraction, segmentation, smoothing kernel size, etc.).* |
| Normalization | *If data were normalized/standardized, describe the approach(es): specify linear or non-linear and define image types used for transformation OR indicate that data were not normalized and explain rationale for lack of normalization.* |
| Normalization template | *Describe the template used for normalization/transformation, specifying subject space or group standardized space (e.g. original Talairach, MNI305, ICBM152) OR indicate that the data were not normalized.* |
| Noise and artifact removal | *Describe your procedure(s) for artifact and structured noise removal, specifying motion parameters, tissue signals and physiological signals (heart rate, respiration).* |
| Volume censoring | *Define your software and/or method and criteria for volume censoring, and state the extent of such censoring.* |

## Statistical modeling & inference

| | |
|---|---|
| Model type and settings | *Specify type (mass univariate, multivariate, RSA, predictive, etc.) and describe essential details of the model at the first and second levels (e.g. fixed, random or mixed effects; drift or auto-correlation).* |
| Effect(s) tested | *Define precise effect in terms of the task or stimulus conditions instead of psychological concepts and indicate whether ANOVA or factorial designs were used.* |

Specify type of analysis: ☐ Whole brain ☐ ROI-based ☐ Both

| | |
|---|---|
| Statistic type for inference<br>(See Eklund et al. 2016) | *Specify voxel-wise or cluster-wise and report all relevant parameters for cluster-wise methods.* |
| Correction | *Describe the type of correction and how it is obtained for multiple comparisons (e.g. FWE, FDR, permutation or Monte Carlo).* |

## Models & analysis

| n/a | Involved in the study | |
|---|---|---|
| ☐ | ☐ | Functional and/or effective connectivity |
| ☐ | ☐ | Graph analysis |
| ☐ | ☐ | Multivariate modeling or predictive analysis |

| | |
|---|---|
| Functional and/or effective connectivity | *Report the measures of dependence used and the model details (e.g. Pearson correlation, partial correlation, mutual information).* |
| Graph analysis | *Report the dependent variable and connectivity measure, specifying weighted graph or binarized graph, subject- or group-level, and the global and/or node summaries used (e.g. clustering coefficient, efficiency, etc.).* |
| Multivariate modeling and predictive analysis | *Specify independent variables, features extraction and dimension reduction, model, training and evaluation metrics.* |

