## [Peer Review File · Nature Ecology & Evolution]

Peer Review Information

Journal: Nature Ecology & Evolution

Manuscript Title: Isotopic evidence of high reliance on plant food among Later Stone Age hunter-gatherers at Taforalt, Morocco

Corresponding author name(s): Zineb Moubtahij

Editorial Notes:

Reviewer Comments & Decisions:

Decision Letter, initial version:

25th September 2023

Dear Ms Moubtahij,

First, I'd like to apologise for the delay in getting the decision on your manuscript entitled "Earliest isotopic evidence of high reliance on plant food in the Late Pleistocene hunter-gatherer population (Taforalt, Morocco)" to you. This delay was caused by the absence of a third reviewer originally assigned to the manuscript, but we have decided that we have sufficient expertise from the other two reviewers with which to proceed to a decision in the absence of reviewer 3. The reviewers are overall positive about the manuscript's contribution but identify a number of technical and presentation issues which will need to be addressed before we can offer publication in Nature Ecology & Evolution.

We therefore invite you to revise your manuscript taking into account all reviewer and editor comments. Please highlight all changes in the manuscript text file.

* If you have not done so already please begin to revise your manuscript so that it conforms to our Article format instructions at <http://www.nature.com/natecolevol/info/final-submission>. Refer also to any guidelines provided in this letter.

2[REDACTED]

Nature Ecology & Evolution is committed to improving transparency in authorship. As part of our efforts in this direction, we are now requesting that all authors identified as 'corresponding author' on published papers create and link their Open Researcher and Contributor Identifier (ORCID) with their account on the Manuscript Tracking System (MTS), prior to acceptance. ORCID helps the scientific community achieve unambiguous attribution of all scholarly contributions. You can create and link your ORCID from the home page of the MTS by clicking on 'Modify my Springer Nature account'. For more information please visit www.springernature.com/orcid.

[REDACTED]

Reviewer expertise:

Reviewer #1: stable isotopes for dietary and environmental reconstruction

Reviewer #2: Moroccan hunter-gatherer archaeology

Reviewers' comments:

Reviewer #2 (Remarks to the Author):

In their manuscript, Moubtahij et al. use a multiproxy geochemical approach to reconstruct the diet of a Late Pleistocene hunter-gatherer population in Morocco. They focus on the onset of agriculture and if the population relied on local foods, the proportion of plant foods in their diet, predating the advent of agriculture by several millennia. While I overall agree with their findings and these data could potentially

3be of great interest for a broad readership, the presented manuscript is lacking some important aspects in the results and discussion. Moreover, especially the SI has a large number of grammatical, structural errors, typos, structural issues and other rather careless mistakes.

Major comments:

In your sample set (human and non-human) are some permanent ("adult") teeth that typically mineralize pre-weaning (e.g., canines, some incisors, see your Table SI3...). These materials (enamel and dentin) are therefore affected by the nursing effect, which is especially pronounced in $\delta^{15}\text{N}$. There is a rather small section dedicated to weaning towards the end of the SI, but as I understand this, these permanent teeth are still part of the "adult diet" reconstructions. The authors should elaborate on this, should clearly mark tooth-type/nursing etc. in the figures and exclude this data from general reconstructions.

The text, especially the SI would benefit from some restructuring, reorganizing and improving the language. For example, line 162ff, mention first what you analyzed (humans, equids, hare...), which material (dentin, enamel, bone) and then which samples were analyzed for what (CT-scans of human teeth only (?)...). Moreover, the SI is very hard to read, the titles are misleading and there are a lot of errors. The authors should work on this carefully to meet publication standards. I indicated some issues in this review, but it was just too many to keep track.

Some very important points in this study seem to be just mentioned as a side note despite the fact that many of the interpretations rely on this. Here I'm missing some info/discussion to elaborate important aspects. For example, the authors state that cereals have generally higher $\delta^{15}\text{N}$ values compared to forage plant, and use only one reference for this statement. Is this systematically true in different habitats (i.e., including Northern Africa)? How do we know this was true already for paleolithic cereal, when agriculture was potentially very young (i.e., is this an effect of fertilizer today? Did this isotopic enrichment increase over time with evolving cereal?). What causes these differences in wild plants vs. cereal? Did the sampled herbivore have access to these cereals? In the same paragraph they casually mention cut marks on the sheep, again an important factor. Not clear in the text if there is evidence for domestication of these sheep or if they were probably wild. These are two examples where the manuscript would benefit from some clear information/discussion.

I am confused about the sulphur analyzes. They state in the SI that only three samples had enough collagen for measurement, then replot "a sheep", "an equid" and "the humans". Is this only one human or many (all) of them? In Table SI3 I can only find one number for $d_{34}\text{S}$ (unassigned human #34561). I cannot find the S data anywhere else, there is also no plot etc., please explain and make clear what the results/interpretations are (i.e., sea spray effect). And show the data in a plot or clear table.

Minor comments:

Abstract:

Information of the geological age of these hunter-gatherer population is missing. "Late Pleistocene" is stated in the title and abstract, but a detailed age is only given much later, in the intro (Line 131 pp.). Sulphur was also measured, but not mentioned in the abstract.

Line 38: many Nature journals ask authors to refrain from making priority or novelty claims. Check if you can use "first" here.

Introduction:

Figure 1: This is a large, almost empty map and pretty useless. It could profit from some more information which you mention in the text (e.g., indicating the region of the Natufian in the Near East, and/or showing the Upper Paleolithic sites in Europe or Asia you compare your AA data with, etc.).

Lines 107, 116, 157, 165, 219 and elsewhere: the word "also" is often redundant. Check and delete

4whereover appropriate.

Line 120: Refer to Fig. 1 when you mention the sites you show in the figure.

Line 145: you mention "specific individuals", but should make clear that you not only analyzed humans, but also coexisting fauna that is art of the local food web.

Line 144 and elsewhere: The "see" in See Supp. Info. should start with a small s or should be removed.

Line 145f: The manuscript would benefit from a short explanation what these baseline variations could be and note that you analyzed non-human faunal remains as well (see my comment on line 146, too).

Line 158: this is the first time you mention Sulphur and is thrown in a bit as a surprise in this section. This should be in abstract etc as well.

Line 162: Make clear if you really scanned all samples. The bones, too? And did you scan the faunal tooth (and bone) remains as well, or just human teeth?

Line 164f: The manuscript could benefit from an overview table with number of individuals and which geochemical analyzes were performed on which element, e.g., a compiled table S3.1 and 3.2.

Line 168: Are there any indicators that these animals were used by the humans and part of the same local food web?

Line 174f and elsewhere: Be careful with dividing your samples into "human" and "animal" as humans are also animals. Maybe use other wording like non-human fauna or something

Results and Discussion:

Figure 2: This figure needs some work to present the data in a way that's easy to understand for the reader.

- Have you considered performing statistical analyzes of this data? I know your samples size are small in some cases, but if you group accordingly, (e.g., all (adult) humans vs. all herbivores), your sample size might be big enough for some statistical sound implications of your findings using e.g., multivariate isotopes comparisons corrected for small sample size using e.g., R SIBER after Jackson et al., 2011 (doi:10.1111/j.1365-2656.2011.01806).

- Error bars are missing. The caption states that each point corresponded to the average value of a single tooth or bone sample. However, it is not clear if you ran samples in duplicates or have multiple samples from a single tooth/bone. This is explained in the SI, so refer to this.

- I am aware that Sr data is not very variable, and that's the point of plot B, I guess. However, you could reduce the size of the axis (e.g., span between 0.708 and 0.710 only) so the data is not plotted too much on top of each other and one can see the individual data points.

- Not clear in which order the datapoints of the individual hominins are connected. According to eruption (mineralization) age? Clarify. Also note my comment about combining dental enamel or dentin of permanent dentition that forms pre- and post-weaning, especially with the incisors. Once you cleaned up this figure It might be feasible to show which teeth which presumable form pre- vs. post weaning?

- Your color scheme could be improved, see also my comment of the Figures in the SI. The green-yellow of Ind. 9 and 14 are difficult to distinguish and Ind. 13 generally hard to see. Maybe a black outline of the cycles would help. Also, the fox (red) and Ind. 5 (pink) are hard to distinguish.

- Move labels so they are not on top of datapoints/lines. Lines to labels look like a continuation of data-connecting lines, but if your color codes are clear, you will not even need labels and can therefore avoid these issues.

- It would be worthwhile to think about possible rearranging the panels by switching C and D so the plots with Zn have the same axis lined up. This way, the plots with $\delta^{15}\text{N}$ data is on the left side, albeit on different axis, which is fine, as if want TP on the y-axis. In A, you could also plot $\delta^{13}\text{C}$ on the y-axis and $\delta^{15}\text{N}$ on the x-axis to then have N in A and C on the same axis, but that not how a C vs. N plot is

5usually shown, so whatever you prefer.

- I do like that you call it “ $\delta^{15}\text{N}$ collagen” here (see my comment below). This seems however not to be done consistent throughout the ms. I suggest to use $\delta^{15}\text{N}$ collagen, $\delta^{13}\text{C}$ collagen, $\delta^{34}\text{S}$ collagen, $\delta^{66}\text{Zn}$ enamel, $^{87}\text{Sr}/^{86}\text{Sr}$ enamel etc. consistently in the text, SI, table and figures.

- Legend on the left:

o You do not have a “group” of carnivores, herbivores, or unassigned humans anywhere (and Ind. 13? Hard to see), remove from legend accordingly.

o Change colors of key of the species according to their diet (i.e., the Canidae will always be red, so no need to have a black symbol here).

o Seems the species key is sorted by alphabet, but I can’t detect a pattern in the key for the groups.

o You could make clear that Ind. and “unassigned” are the humans in the legend. Maybe even indicate which one is adult, which one infant.

- No need to specify $\delta^{15}\text{N}$ twice in the caption.

- Use Latin name throughout (e.g., for “fox”).

- The differences in the spot size for adult and infant are difficult to see, think of different way to make this clear.

Line 194: remove “in literature”.

Line 202 and elsewhere: You only have two carnivores (which potentially are not even true carnivores), this is problematic when calculating TLS. I understand that you already sampled all available carnivore teeth (well done!), nothing to be done here. However, you need to tone down your argumentation here a bit. There is a large variation in e.g., N and Zn data in the herbivores (and humans, for that matter), and you have no control to understand these patterns in small carnivore dataset.

Line 204f: I do not think you only have considered teeth after weaning here, but also included permanent dentition that typically mineralize pre-weaning. See my comment above.

Line 219: This is not a $\delta^{66}\text{Zn}$ interpretation, this is an Interpretation of $\delta^{66}\text{Zn}$ data.

Line 245ff: Here or at least in the SI you should state if you used the same (or some of the same) individuals (in sheep or humans) as Lee-Thorp (don’t forget the dash in the name!). How big is their dataset? Did they use dentine or bone collagen? You state correctly that they the focused on sheep, but they also have Bos, Equus, Gazelle, how do these values compare? The main results of Lee-Thorp is that the Sector 10 humans did consume a reasonable amount of animal resources, but no marine resources, which fits (partly to your conclusion). Elaborate!

Line 255ff: That cereals have generally higher $\delta^{15}\text{N}$ values compared to forage plant is a very important point for your study. Here I’m missing some info/discussion about this.

Line 264: Should it say vegan instead of vegetarian? Do you have any evidence of animal products like dairy or eggs that are considered part of a vegetarian diet.

Line 266: So there are cut marks on the sheep? I think this is new information for the reader which is quite crucial.

Line 270ff. This is another example where some crucial information is lacking in the data table and figures: Which sample is di2 and which one the long bone and the rib (both simply called “bone” in Table SI3, and not labeled at all in the figures)? Also, the info when these bones formed is in the text, but missing in the table (where is this information from? Please add reference!). In the table the “age of crown dev.” (development?) of the dm2 is at 10 months, you sampled the bottom of the tooth and estimate that this formed at the beginning of the first year. This is confusing, how does the “bottom” of a tooth form before the crown? Or did I misunderstand the abbreviation? Moreover, in the SI with the

6Scans of the tooth the collagen sampling is shown through the whole length of the tooth, not just the bottom.

Line 289: not clear what you refer to as a “broad-spectrum and dietary breath models”, elaborate.

Line 290: what do you consider as higher and lower ranking resources in the light of your study? Lower: everything but animal products? Only wild plants? Higher: meat only? Cereal? Explain! How does cooking (charred plant remains) play a role here?

Line 300: you mentioned the possible storage in baskets before (in the intro), I suggest to move the info about the Alfa grass up where you mention this observation first.

Line 302: Reference for seasonality of plant maturity?

Concluding remarks:

Line 319. “Younger Dryas (YD)”: You should only introduce an abbreviation if you use it a few times afterwards. The only time you mention this term again (Line 327), you don’t even use the abbreviation.

Supplementary Information:

Generally, the SI is full of minor and major mistakes and written extremely sloppy. Please carefully read this again and correct mistakes and formatting errors throughout, I can’t point them all out in here.

Generally, the SI would greatly benefit from some reorganization. Report results of each analyzes in the respective section,

Line 22: instead of using “distant from village” indicate direction (e.g., northeast).

Line 37ff. There is a large age gap between the two units. Any idea why (erosional surface? Hiatus?)?

Fig. S1.1: The maps in the lower right corner should be reworked. The first one is not really needed, as a map like this is in the main text (Fig. 1). The second, closer map does not help at all without any names, rivers etc. Maybe instead use the zoomed-in map with some labels, and then an even more zoomed-in map with the location of the cave within the village?

Line 105: I suggest to use “preferred” instead of “favorite”

Line 107: Should be “was primary consumed by this population” instead of “was primary consumption by this population”

Line 110: punctuation mark missing

Table S1.1: I suggest to sort these by abundance

127ff: I am missing information about possible diagenesis. How are your C/N ratios (you note these in Table S1.3, but do not mention this data at all. You need to show that you can eliminate the effect of possible overprint of the dietary signals.

Line 144: Explain what you mean with “different sources”

Line 154: refer to most important studies here (for bulk isotope analyzes)

Line 166: reference for this equation is missing here

Line 170: you stated the abbreviations for Phe and Val already above. Only do this once and then use abbreviations!

Line 183: decide if you use “terrestrial C3/C4 consumers” or “C4/C3 terrestrial consumers” and stick with it

Line 174/175: does this really refer to all animals, or only herbivores?

Line 211: By how much?

Line 227: ...is associated with low...

Line 235: the statement “This later is produced by the process of the radioactive decay of rubidium Rb” is repeated in a few lines down again.

Line 254f: “For humans, strontium is incorporated into their skeleton via diet”. Isn’t that also true for all

7other animals?

Line 267: should be: "...close to +20 ‰ which..."

Line 282: and elsewhere: Delete large numbers of random paragraph breaks

Line 295: Here you correctly state that the sampled carnivores might not have a pure carnivorous diet, since many canid species can feed on a variable amount of plant material. This is very important for your interpretations and should definitely be mentioned in the main text!

Line 297: you only talk about the carnivores here. What about the herbivores? Looks like they were all clearly identified?

Line 342: should be taxonomic identification

Table S3.1: what is ID 34566, Bone or tooth?

Line 367: Here you state correctly that the hypoplasia can be the result of weaning, but you sample material across this line and still use it as "adult" diet if I understand correctly. Elaborate

Line 406: correct spelling of your own institute is with capital first letters

Line 413f: why did you not sample systematically in all teeth?

Line 421: You mean the age of the individual at the time when the enamel was formed. Rephrase

Line 431: did you only serially sample equids?

Line 457: You mean measured in duplicates? Was the enamel also pre-treated in two separate batches so you have true duplicates?

Line 458: report typical standard deviation

Line 463: Did you use the same aliquot for Sr that you've used for Zn?

Line 466: correct to "...2 ml of 3 ‰..."

Line 484: add units (‰)

Line 485: name standards

Line 537: you send it where? Or do you mean you measured it? Are you using the same setup as Jaouen et al., 2019?

Figure S4.1.: Add units to axis. Error bars are 1σ ?

Figure S4.2.: Delete unnecessary digits

Line 601 and elsewhere: usually, three digits are sufficient to show significance. Use $p < 0.001$ here.

Figure S4.3: Again, think about human vs. animal. Maybe use non-human fauna instead?

Line 621: You have two individuals here that you refer to as carnivores, right? A M1 from a *Vulpes vulpes* and a M? from a *Canidae*. Therefore, a) the sample number ($n = 2$) is much too small to run these statistics, b) M1 most likely incorporates a nursing signal in the fox (who knows about the molar from the *Canidae*) and c) at least the fox and possibly the other canid is not a "true" carnivore but often supplements its diet with plant materials etc. So, you have to tone down your interpretations/conclusions

Line 624f: But even your permanent-tooth-dataset includes enamel that typically forms pre-weaning (see above)

Line 625: t missing in breastfeeding. Better, use the word nursing. Citation at wrong place (before the comma).

Line 627: Again, calculating a trophic spacing from a very small sample set without true carnivores with an adult diet is problematic to say the best...

Figure S4.5: This is a good place to indicate all teeth that (might) include a nursing effect. If I am not mistaken, you use deciduous teeth only for the juvenile boxplot, but these should also include permanent teeth that typically form prior to weaning. In sheep, also only M2 and M3 form post-weaning, I think. Hence, the "juvenile" sheep data should include the incisors you sample, and the adult one only

the M3 and M2 data. It looks like you included the incisors of the sheep also in the adult data. Also, fix y-axis label.

Line 653ff: Move N results to N section

Table S4.5: Confusing to have the N results in the Zn section.

Figure S4.6: Again, separate teeth into pre- and post-weaning (here and elsewhere, also for the non-human fauna)

Figure S4.8: Looks like at least in Ind 1 a weaning effect is visible with lower Zn values in the I, C and P4 and higher ones in M2s (apart from the one M2 with a low Zn value. This is not clear in Ind. 5 and the unassigned individuals; this needs to be discussed. You have a tooth-type vs. Zn values figure below (Fig. S5.1), would be helpful to have that Fig. up here. Also, make the same Fig. with N data vs. tooth type.

Line 683 and elsewhere (see also comment above): I suggest to use $\delta^{13}\text{C}_{\text{collagen}}$ (vs. $\delta^{13}\text{C}_{\text{Pro}}$ etc.) to make this clearer. If you do, make sure to change this everywhere, in this SI, but also in the main text and figures.

Line 686: again, make sure you really only have adult diet here.

Line 690: It's an important info here that one canid seems to occupy a carnivore trophic level. State in main text?

Line 704: Decide if you use subscript for C3 and C4 or not and make the same throughout the text.

Figure 765: The yellow data of Taforalt Humans is hard to see, change color scheme as this is the important data.

The lines are confusing. Looks like the line for "C3 humans" goes into the red field (C3 carnivore?) and the C3 herbivore Taforalt to the yellow field (Taforalt humans). Also, lines cross some of the labels, clean up. What is the orange group ("human") here? If they are not C3, C4 r marine consumers or from Taforalt?

For Figure S4.10 and the following figures: Again, yellow hard to see. Needs at least dark outlines. Remove figure header, you also don't have any in the other figures. Add vs. which standard to axis titles.

Line 751: I think your sample size is too small for this statement as you only have one bone from one individual, right? This tells you nothing about the range and variability of this taxon.

Line 785: for one equid but all hartebeest? Clarify. Also, here and elsewhere: the common names are spelled with small first letter

Line 794f: Is the TP when excluding the adult samples which formed pre-weaning?

Line 800 f: why do you reject this?

Line 801: "Hartebeest, hare, the Hartebeest and the Rhinoceros"?

Line 815: Again, what about permanent teeth that form during nursing?

Line 839: Did you analyze all humans? Is this an average value?

Line 872ff: Obviously, reading your ms until here I was missing this section! Parts of this definitely needs to be included in the main text. However, your argumentation here does not convince me at all! What about the incisors and canines. Or did I misunderstand that you included their results also in your statistics and interpretations?

Figure S5.1: Do these Rennes and Lapa do Santo studies also have $\delta^{15}\text{N}$ analyses? I would like to see the same plot for bulk collagen $\delta^{15}\text{N}$ and other analyzes you conducted on all tooth types.

Figure 5.2 (should be S5.2): This is not very helpful. A line for each individual sorted by tooth type as in the previous study would be more helpful.

Line 915: reference for equation?

Line 917: "still"? Compared to what?

Line 946: Do humans with low status usually die younger? The sampled individuals here all have died young, can you elaborate on that?

Line 973 and elsewhere: the unit (‰) belongs after the stdev

Line 2021: You state that "the shells possibly were used as ornaments or grave goods that accompanied the grave of the disease". That's what comes to mind first of course. But, why is that, so you have a reference or is this just a well-educated guess? As this is unimportant for your study, I would remove this sentence.

Line 1031: I agree that the sample is reflecting breastfeeding. But you have yet to convince me that early forming teeth of adults do not show the same.

Excel Datasheet (some of these corrections/comment should be considered for all tables):

Table SI1:

- typo in header (study)
- Gazelle is not a carnivore!
- The Canidae is an M1, right? At least that's what you state in Table SI3. Please crosscheck all for to have correct data here!
- Water intake after which study. Fill empty field of the one Hartebeest.
- Clearer if sorted by diet/taxa instead of SEVA ID.
- I guess the actual sample ID is #REF? Indicate which catalogue this refers to. Some of these IDs have a dot, others don't, is this correct? Also check in other tables.
- Again, here you have a few mammalian teeth that typically form during nursing. These should be indicated and removed from or highlighted in certain data compilations.

Table SI2:

- Again, indicate material that typically forms pre-weaning

Table S3:

- Refer to citations correctly, add year and doi.
- Add units for all columns with data, not just some (e.g., add ‰ for $\delta^{66}\text{Zn}$, not ppm in the column with the concentration.
- "What does the "dev" at Age crown dev. Stand for? Explain all abbreviations.

Table S4:

- Add units,
- Here you only have SEVA IDs, but it's called "sample name". Make consistent. You could add collection ID as well.

Table S5

- Add units.
- Move caption above table.

Table S7:

- Use same text size throughout
- The colors of each individual do not correspond to the color in Fig. 1. Choose one color scheme and stick without throughout the whole MS, including SI and Figures.

General

- Check consistency e.g., space between number and unit, the use of "cal. BP" vs. "cal BP", using the delta notation once introduced

- Search for double spaces and delete them
- Delete spaces before citation ID
- Table captions should be above the tables.
- Make sure you introduce abbreviations when you first use the term in the text and SI. After that, stick with the same abbreviation and don't introduce them again.
- Use a coma after e.g.
- Do not mix British and American English.
- Element names are spelled with small starting letter. If you use the abbreviation, it's starts with a capital letter (e.g., nitrogen and N). Correct throughout ms.
- Letters until 12 are spelled out if they are not followed by a unit
- Decide on capitalization of the headers and stick with it
- Use same units. E.g., for hour you sometimes use h, sometimes hr, sometimes hrs and sometimes spell it out, I think. Make consistent.

Reviewer #3 (Remarks to the Author):

REVIEWING REPORT

The manuscript "Earliest isotopic evidence of high reliance on plant food in the Late Pleistocene hunter-gatherer population (Taforalt, Morocco)" by Ms Moubtahij and co-authors, which has been submitted for publication in Nature Ecology & Evolution lies within the scope of this journal. This research deals with the issue on the origin of agricultural-based economy and the role of increased reliance on plant foods in the Late Pleistocene hunter-gatherer population of Taforalt (Morocco) with parallel to the Natufian experience. In order to reconstruct the dietary habits and the mobility pattern of the human groups from Taforalt, the authors conducted for the first time a strong multi-isotope approach which combines zinc ($\delta^{66}\text{Zn}$) and strontium ($^{87}\text{Sr}/^{86}\text{Sr}$) analysis on dental enamel, carbon ($\delta^{13}\text{C}$), and nitrogen ($\delta^{15}\text{N}$) isotope analysis on dentin and bone collagen, along with single amino acid analysis, which makes this manuscript highly important. The isotopic analyses were conducted on a high-resolution data containing human remains and associated fauna recovered in the recently excavated burials from sector 10 at Taforalt. The study results emphasize to us the adoption of a starchy diet along with a decreased consumption of meat compared to Upper Palaeolithic sites with available isotopic data in Europe and Asia. Furthermore, the trophic position (TP) values at Taforalt are similar to the TP values of Neolithic farmers from the Levant, which enhances the evidence for substantial plant consumption, thus challenging traditional ideas about pre-Neolithic hunter-gatherer societies.

Overall, the manuscript is well-written. Data collection is adequately and openly presented in sufficient detail with additional information structured into 7 chapters provided in the supplementary information. The literature cited is very informative and relevant to the topic of the current manuscript. All figures are appropriate and the statistical tests are displayed with accuracy. The argumentation is well stated as it is clearly indicated in the abstract. However, some assumptions may need clarification or require to be substantiated by references. Here are listed the main points along with the comments:

Lines 95-97 The definition given by the authors to the Iberomaurusian hunter-gatherers needs to be enhanced by appropriate references. Here are some comments/suggestions:

- (1) the production of blades and bladelets is not only peculiar to the Iberomaurusian knappers. Instead, a term such as bladelet-based stone technology is more appropriate.
- (2) there is no agreement between scholars as for the longevity of the Iberomaurusian (whether the end

11of the Iberomaurusian falls within Terminal Pleistocene or rather coincides with the onset of the Holocene)¹.

(3) In regard to the age 25,000 cal BP, the authors may refer to scholars who have published the lithic record of Tamar Hat basing on their personal examination^{2,3}.

Line 119: No burials were found at Tamar Hat. Only 6 deciduous teeth were recovered and they belong to individuals who were not buried in this site⁴. Furthermore, I suggest the term sheltered sites instead of caves, since Afalou is a rockshelter.

Line 128: Afalou bou Rhummel burials (layer IV) are older than those of Taforalt (and it is likely that layer V burials are of a similar age or slightly older⁵. If they are proven not to be, the authors need to provide the reference.

Lines 293-296: This research emphasizes the role of highly reliance on cariogenic wild plant food and teeth carries. Maybe it would be germane to also mention that dental evulsion (which has not addressed in this research) is not believed to be linked directly to oral pathology⁶. Furthermore, I recommend that the authors mention (maybe in the introduction) the archaeological records concomitant with the LGM showing evidence for plant processing. I refer in particular to Ifri el Baroud where small seeds of wild legumes may have probably been gathered for consumption purposes⁷, or Tamar Hat where wild seeds of pine cones with pestle-grinders were recovered⁴. It is also relevant to cite Ohalo II, a late Upper Paleolithic site in the Levant where the earliest known usage of plants ca. 23,000 years ago has been reported⁸ (Weiss et al. 2008). In all three cases, no substantial use of edible plants has been evidenced, but this will better support the relevance of the study results of the current manuscript.

Lines 300-302: According to the statement of this paragraph, year-round assumption at Taforalt relies on the availability of edible plant-harvesting and possibly storage. However, relevant signs of seasonality are evidenced from selective hunting of Barbary sheep and collection/consumption of edible molluscs⁹. Authors may provide further clarification to better enhance the increased sedentism assumption at Taforalt.

Lines 303-305: Yes, the population of Taforalt increased their reliance on plant resource, but there is no significant change in the availability of higher-ranking foods, if we assume that the local (and primary) game Barbary sheep is the highest-ranked resource, and then the main source of protein⁹. Besides, Taforalt burials were directly dated to 15,077 - 13,892 cal. BP (cited in this manuscript), which coincides with the rapid warming period GS1. In this case, why Late Iberomaurusian populations in Taforalt have shifted to a plant-based economy if there are no signs of population experiencing resource stress, or climatic deterioration? At another level, there are no records of flakes or blades with macroscopic gloss, and none of the microliths showed evidence of any mastic or remains of hafting materials⁹, although the prevalence of charred plant materials. I understand these issues are beyond the scope of this manuscript, but it would be relevant to briefly raised them in the text to enhance the argumentation.

Here are additional remarks:

The title: I would rather suggest to mention the term Later Stone Age (already used in the literature) instead of Late Pleistocene (c. 129,000 and c. 11,700 years ago), since the collected data belong to the Iberomaurusian which covers a very small part of the Late Pleistocene. The term Pre-Neolithic, already mentioned in this paper is another option.

Line 87: It would be useful to also cite the new paleogenomic study conducted on Epipalaeolithic-Middle Neolithic human remains from Morocco (Simões et al. 2023) which concluded to a mosaic of incoming groups admixing with local people¹⁰.

Line 99: References are needed to assert the idea of the absence of domestication during the Iberomaurusian¹¹⁻¹³.

Line 131: the sentence should be "...warming period following (and not during) the Last Glacial Maximum (LGM)". Also, Poti et al. 2019 are cited instead of Barton et al. 2019.

Line 152: remove the repeated word.

Line 200: I think a word is missing here.

Line 320: Authors need to add uncal BP to the age of Younger Dryas.

Supplementary Information 1 - Line 108: It is worth mentioning that Barbary sheep horn cores used as funerary objects accompanying human burials in sector 10 at Taforalt have also been reported at Afalou bou Rhummel^{14,15}. This is to raise the idea that some common and synchronous idiosyncratic practices occurred in different biogeographic entities during the Late Iberomaurusian.

Reviewer's decision

This is an interesting and original paper. As reported by the authors, there is no compelling evidence at Taforalt for the precocious domestication of either plants or animals. However, the current research demonstrated the abundance of plant resources in the environment of the Pre-Neolithic groups led to a greater concentration of foraging for lower-ranking resources. Further research into coastal sites is needed to distinguish between dietary habits rigorously determined by different local environments. I expect this paper will draw interest from researchers in different fields who are interested in the origin of farming. Therefore, I would highly recommend publishing the manuscript after the minor corrections which have been noted above.

References

1. Hogue, J. The origin and development of the Pleistocene LSA in Northwest Africa: A case study from Grotte des Pigeons (Taforalt), Morocco. Oxford: University of Oxford, PhD thesis (2014) [https://ora.ox.ac.uk/objects/uuid:3a95e8b0-5e0b-4e8b-baac-b2e44a327d8a/download_file?file_format=application/pdf&safe_filename=THESIS01&type_of_work=The sis](https://ora.ox.ac.uk/objects/uuid:3a95e8b0-5e0b-4e8b-baac-b2e44a327d8a/download_file?file_format=application/pdf&safe_filename=THESIS01&type_of_work=The%20thesis)
2. Close, A. E. The Iberomaurusian sequence at Tamar Hat. *Libyca* 29, 69–103 (1981).
3. Sari, L. Technological change in Iberomaurusian culture: The case of Tamar Hat, Rassel and Columnata lithic assemblages (Algeria). *Quaternary International* 320, 131–142 (2014).
4. Saxon, E. C., Close, A., Cluzel, C., Morse, V., Shackelton, N. J. Results of recent investigations at Tamar Hat. *Libyca* 22, 49–91 (1974).
5. Hachi, S. Fröhlich, F., Gendron-badou A., de Lumley H., Roubet C., Abdessadok S. Figurines du Paléolithique supérieur en matière minérale plastique cuite d'Afalou Bou Rhummel (Babors, Algérie) : premières analyses par spectroscopie d'absorption infra-rouge. *L'Anthropologie* 106, 57–97 (2002).
6. Humphrey, L. T. et al. Earliest evidence for caries and exploitation of starchy plant foods in Pleistocene hunter-gatherers from Morocco. *Proc. Nat. Acad. Sci.* 111, 954–959 (2014).
7. Poti, A. et al. Human occupation and environmental change in the western Maghreb during the Last Glacial Maximum (LGM) and the Late Glacial. New evidence from the Iberomaurusian site Ifri El Baroud (northeast Morocco). *Quat. Sci. Rev.* 220, 87–110 (2019).
8. Weiss, E., Kislev, M.E., Simchoni, O., Nadel, D. & Tschauner, H. "Plant-food preparation area on an Upper Paleolithic brush hut floor at Ohalo II, Israel". *Journal of Archaeological Science* 35 (8), 2400–2414 (2008). Bibcode:2008JARSc..35.2400W. doi:10.1016/j.jas.2008.03.012.
9. Barton, R. N. E. et al. Origins of the Iberomaurusian in NW Africa: New AMS radiocarbon dating of the Middle and Later Stone Age deposits at Taforalt Cave, Morocco. *Journal of Human Evolution* 65, 266–281 (2013).
10. Simões, L.G., Günther, T., Martínez-Sánchez, R.M., Vera-Rodríguez, J.C., Iriarte, E., Ricardo

13Rodríguez-Varela, Bokbot, Y., Valdiosera C. & Jakobsson, M. Northwest African Neolithic initiated by migrants from Iberia and Levant. *Nature* 618, 550–556 (2023).

11. Klein, R. G., & Scott, K. Re-analysis of faunal assemblage from the Haua Fteah and other late quaternary archaeological sites in Cyrenaican Libya. *Journal of Archaeological Science* 13, 515–542 (1986)

12. Merzoug, S. & Sari, L. Re-examination of the Zone I Material from Tamar Hat (Algeria): Zooarchaeological and Technofunctional Analyses. *Afr. Archaeol. Rev.* 25, 57–73 (2008).

13. Turner, E. Large mammalian faunal assemblages. in *Cemeteries and Sedentism in the Later Stone Age of NW Africa: Excavations at Grotte des Pigeons, Taforalt, Morocco 239–308* (Monographien des Römisch-Germanischen Zentralmuseums, 2019).

14. Arambourg, C., Boule, M., Vallois, H. & Verneau R. Les grottes paléolithiques des Beni Ségoual (Algérie). (*Archives de l'Institut de Paléontologie Humaine* (Édition Masson (1934).

15. Hachi, S. Résultats des fouilles récentes d'Afalou Bou Rhummel (Bédjaïa, Algérie). in *El Món Mediterrani Despres del Pleniglacial (18.000-12.000 BP)* (Ed. (ed. Fullola, J.-M. & Soler, N.) 77–92 (Girona : Museu d'Arqueologia de Catalunya, 1997).

Author Rebuttal to Initial comments

Reviewer 1

In their manuscript, Moubtahij et al. use a multiproxy geochemical approach to reconstruct the diet of a Late Pleistocene hunter-gatherer population in Morocco. They focus on the onset of agriculture and if the population relied on local foods, the proportion of plant foods in their diet, predating the advent of agriculture by several millennia. While I overall agree with their findings and these data could potentially be of great interest for a broad readership, the presented manuscript is lacking some important aspects in the results and discussion. Moreover, especially the SI has a large number of grammatical, structural errors, typos, structural issues and other rather careless mistakes.

Major comments:

In your sample set (human and non-human) are some permanent (“adult”) teeth that typically mineralize pre-weaning (e.g., canines, some incisors, see your Table SI3...). These materials (enamel and dentin) are therefore affected by the nursing effect, which is especially pronounced in $\delta^{15}N$. There is a rather small section dedicated to weaning towards the end of the SI, but as I understand this, these permanent teeth are still part of the “adult diet” reconstructions. The

authors should elaborate on this, should clearly mark tooth-type/nursing etc. in the figures and exclude this data from general reconstructions.

A: Thank you for your valuable feedback. Following your comment, we have taken several steps to improve the clarity of our data presentation and address the potential impact of nursing on teeth formed pre- and post-weaning. In Supplementary Information 3, Table S6.3, Table S6.4 and the figures, we have now separated the teeth that mineralize pre-weaning from those that mineralize post-weaning.

Additionally, in Supplementary Information 3, we have incorporated a paragraph that elaborates on our categorization of teeth and the potential influence of nursing (3.3.1.1), both for human samples and for certain species of fauna. We recognize that the available literature for fauna is limited, and therefore, our approach is based on the best available data.

In our interpretation of the adult diet, we have exclusively utilized values unaffected by nursing.

The text, especially the SI would benefit from some restructuring, reorganizing and improving the language. For example, line 162ff, mention first what you analyzed (humans, equids, hare...), which material (dentin, enamel, bone) and then which samples were analyzed for what (CT-scans of human teeth only (?)...). Moreover, the SI is very hard to read, the titles are misleading and there are a lot of errors. The authors should work on this carefully to meet publication standards. I indicated some issues in this review, but it was just too many to keep track.

A: We thank you for the comprehensive review. The text and especially the SI have been reorganized and the language has been improved thanks to the native speakers who are co-authoring the manuscript. We restructured the paragraph at line 162 (now line 173-184) to provide a clearer presentation of the analyzed elements, materials used, and details of each sample. We have further gone through the SI Material in its entirety addressing all comments made by the reviewer (see below) with great care and beyond that we have improved the presentation, language and structuring of the SI Material in general.

Some very important points in this study seem to be just mentioned as a side note despite the fact that many of the interpretations rely on this. Here I'm missing some info/discussion to elaborate important aspects. For example, the authors state that cereals have generally higher $\delta^{15}\text{N}$ values compared to forage plant, and use only one reference for this statement. Is this systematically true in different habitats (i.e., including Northern Africa)? How do we know this was true already for paleolithic cereal, when agriculture was potentially very young (i.e., is this an effect of fertilizer today? Did this isotopic enrichment increase over time with evolving cereal?). What

causes these differences in wild plants vs. cereal? Did the sampled herbivore have access to these cereals? In the same paragraph they casually mention cut marks on the sheep, again an important factor. Not clear in the text if there is evidence for domestication of these sheep or if they were probably wild. These are two examples where the manuscript would benefit from some clear information/discussion.

A: We appreciate the reviewer's comments and feedback. We would like to clarify our statements:

1) Regarding the first example, we indeed acknowledge the potential difference in $\delta^{15}\text{N}$ values between human-consumed grains and animal-consumed plants. This distinction highlights an important aspect of isotopic studies, a topic that has been discussed by Hedges and Reynard (Hedges and Reynard, 2007). The common assumption is that local herbivores would exhibit $\delta^{15}\text{N}$ values reflecting the local vegetation (since they were not domesticated), and subsequently, the dietary plant proteins of humans. However, it is essential to recognize that various factors can contribute to variations between the isotopic values of plants consumed by humans and those of herbivores:

- The charring effect can result in an increase of up to 2‰ in the $\delta^{15}\text{N}$ values (Bogaard et al., 2007; Fraser et al., 2013, 2011), which primarily affects plants consumed by humans and not those by herbivores.

- Manuring can indeed lead to higher $\delta^{15}\text{N}$ values in plants, however, we want to clarify that our text does not make any statement suggesting such enrichment. Instead, our emphasis is on the differentiation of $\delta^{15}\text{N}$ values in baseline plants. We also want to clarify that no evidence of early cultivation is detected at the site of Taforalt.

- We added a new statement to answer your comment lines 293-297.

2) In response to the comment about the cut marks on the Barbary sheep, we would like to emphasize that at no point in our text did we claim that the Barbary sheep was domesticated, nor is there any evidence to suggest that this species was domesticated in Taforalt or other Iberomaurusian sites. The presence of cut marks on Barbary sheep is undeniable, but it is important to note that similar cut marks were also identified on other species recovered from the Iberomaurusian levels at Taforalt.

To address this, we explicitly stated in our introduction and discussion that the zooarchaeological evidence indicates that the Taforalt population primarily relied on Barbary sheep but also consumed other ungulates (Lines 110-112 and 305-306). We appreciate the opportunity to clarify

this point and we ensure that our revised manuscript includes a more comprehensive discussion of these factors.

I am confused about the sulphur analyzes. They state in the SI that only three samples had enough collage for measurement, then replot “a sheep”, “an equid” and “the humans”. Is this only one human or many (all) of them? In Table SI3 I can only find one number for d34S (unassigned human #34561). I cannot find the S data anywhere else, there is also no plot etc., please explain and make clear what the results/interpretations are (i.e., sea spray effect). And show the data in a plot or clear table.

A: In response to the reviewer's feedback, we have integrated a dedicated section in the Supplementary Results (lines 927-951) presenting and discussing the few $\delta^{34}\text{S}$ data and our interpretation and featuring a plot illustrating the sulfur isotope values for the three samples with sufficient collagen. Notably, we emphasize the limited interpretative scope due to the small sample size, a concern explicitly addressed in the revised paragraph (Supplementary Information 4). We have corrected the spelling mistake ‘humans’ to ‘human’. We have also provided the sample ID for each of the three $\delta^{34}\text{S}$ analysed samples in the caption of the Figure 4.12.

Minor comments:

Abstract:

Information of the geological age of these hunter-gatherer population is missing. “Late Pleistocene” is stated in the title and abstract, but a detailed age is only given much later, in the intro (Line 131pp.).

A: Thank you very much for noting this. We have added that the human remains were dated to 15,000 - 13,000 cal BP in the abstract (line 38-39).

Sulphur was also measured, but not mentioned in the abstract.

A: Following your suggestion, we now mention sulphur isotope analyses in the abstract (line 41)

Line 38: many Nature journals ask authors to refrain from making priority or novelty claims. Check if you can use “first” here.

A: Thank you for your comment. In line with Nature's policy, which permits priority claims in fields like archaeology when verified through peer review, we believe it is appropriate to include the claim of being the first in our study. For now, we would thus prefer using the term but are of course willing to refrain from using it should the reviewers or the editor advise against it.

Introduction:

Figure 1: This is a large, almost empty map and pretty useless. It could profit from some more information which you mention in the text (e.g., indicating the region of the Natufian in the Near East, and/or showing the Upper Paleolithic sites in Europe or Asia you compare your AA data with, etc.).

A: We thank the reviewer for the valuable suggestions. Following your recommendation, we included the other sites mentioned in the text (Figure 1, Main text).

Lines 107, 116, 157, 165, 219 and elsewhere: the word “also” is often redundant. Check and delete wherever appropriate.

A: We have carefully reviewed the text and have made the necessary revisions eliminating the word ‘also’ where redundant.

Line 120: Refer to Fig. 1 when you mention the sites you show in the figure.

A: It has been done.

Line 145: you mention “specific individuals”, but should make clear that you not only analyzed humans, but also coexisting fauna that is part of the local food web.

A: We appreciate the reviewer’s comment. We have clarified that our analysis encompasses not only specific individuals but also coexisting fauna. (line 150-152)

Line 144 and elsewhere: The “see” in See Supp. Info. should start with a small s or should be removed.

A: Thank you, the “see” has been removed.

Line 145f: The manuscript would benefit from a short explanation what these baseline variations could be and note that you analyzed non-human faunal remains as well (see my comment on line 146, too).

A: We thank the reviewer for this suggestion. We have added a brief explanation of the baseline variations (lines 153-155), which can include natural variations in stable isotope ratios within the local environment. Additionally, we have explicitly mentioned that we analyzed both human individuals and faunal remains (line 172).

Line 158: this is the first time you mention Sulphur and is thrown in a bit as a surprise in this section. This should be in abstract etc as well.

A: Thank you for this comment. We have incorporated sulfur into the abstract to provide a comprehensive overview of our analytical approach (line 41).

Line 162: Make clear if you really scanned all samples. The bones, too? And did you scan the faunal tooth (and bone) remains as well, or just human teeth?

A: Our analysis involved the scanning of human teeth exclusively, and we have updated the manuscript to reflect this accurately. We thank the reviewer for helping us clarify this aspect of our methods (lines 171-176)

Line 164f: The manuscript could benefit from an overview table with number of individuals and which geochemical analyzes were performed on which element, e.g., a compiled table S3.1 and 3.2.

A: Following your recommendation we added an overview table in the Excel sheet (Table S6.11) showing the number of individuals and the different geochemical analyses performed for the different isotope systems. We refer to this additional table line 174.

168: Are there any indicators that these animals were used by the humans and part of the same local food web?

A: The animal species analyzed in our study are all from Sector 10 or Sector 8 of the site. Study on the remains shows that these species were deliberately used by the humans. They all show evidence and marks of butchery activities (Turner, 2020). Following your comment, we now clarify this in the main paper line 178.

Line 174f and elsewhere: Be careful with dividing your samples into “human” and “animal” as humans are also animals. Maybe use other wording like non-human fauna or something

A: Thank you for your valuable suggestion. We have revised the terminology. We now use the term “Fauna” to distinguish the “non-human” samples from the human samples.

Results and Discussion:

Figure 2: This figure needs some work to present the data in a way that's easy to understand for the reader.

- *Have you considered performing statistical analyzes of this data? I know your samples size are small in some cases, but if you group accordingly, (e.g., all (adult) humans vs. all herbivores), your sample size might be big enough for some statistical sound implications of your findings using e.g., multivariate isotopes comparisons corrected for small sample size using e.g., R SIBER after Jackson et al., 2011 (doi:10.1111/j.1365-2656.2011.01806).*

A: We thank the reviewer for their detailed comments on Figure 2. Following their suggestion, we used SIBER and realized that it was equivalent to produce 95% confidence ellipses and geometric shape similar to what stat_chull does, so this is what we did to produce new subfigures, the Figures 2C and 2D of the new Figure 2.

- *Error bars are missing. The caption states that each point corresponded to the average value of a single tooth or bone sample. However, it is not clear if you ran samples in duplicates or have multiple samples from a single tooth/bone. This is explained in the SI, so refer to this.*

A: We are sorry for not adding the error bars earlier. We now produce two figures using the same isotope systems. Figure 2A and 2B show every single sample with line connected samples belonging to the same individuals, and Figure 2C and 2D show the average values with the error bars. We further included the analytical errors in the Figure caption.

- *I am aware that Sr data is not very variable, and that's the point of plot B, I guess. However, you could reduce the size of the axis (e.g., span between 0.708 and 0.710 only) so the data is not plotted too much on top of each other and one can see the individual data points.*

A: Thank you for this suggestion. It has been corrected on all figures.

- *Not clear in which order the datapoints of the individual hominins are connected. According to eruption (mineralization) age? Clarify. Also note my comment about combining dental enamel or dentin of permanent dentition that forms pre- and post-weaning, especially with the incisors. Once you cleaned up this figure It might be feasible to show which teeth which presumable form pre- vs. post weaning?*

A: This is an excellent suggestion. The symbols used on the new Figures 2A and B indicate which sample is formed post or pre-weaning.

- *You color scheme could be improved, see also my comment of the Figures in the SI. The green- yellow of Ind. 9 and 14 are difficult to distinguish and Ind. 13 generally hard to see. Maybe a black outline of the cycles would help. Also, the fox (red) and Ind. 5 (pink) are hard to distinguish.*

A: We used a color scheme recommended for color blind people but we do agree with the comment of the reviewer and we modified the colors. We also feel that the use of the 95% ellipse now helps the distinction

- *Move labels so they are not on top of datapoints/lines. Lines to labels look like a continuation of data-connecting lines, but if your color codes are clear, you will not even need labels and can therefore avoid these issues.*

A: We moved the labels as recommended.

- *It would be worthwhile to think about possible rearranging the panels by switching C and D so the plots with Zn have the same axis lined up. This way, the plots with $\delta^{15}\text{N}$ data is on the left side, albeit on different axis, which is fine, as if want TP on the y-axis. In A, you could also plot $\delta^{13}\text{C}$ on the y-axis and $\delta^{15}\text{N}$ on the x-axis to then have N in A and C on the same axis, but that not how a C vs. N plot is usually shown, so whatever you prefer.*

A: Thank you for this comment. It has been done. In addition, Figure 2 has been subdivided into Figure 2 and Figure 3, for more readability and homogeneity.

- *I do like that you call it " $\delta^{15}\text{N}_{\text{collagen}}$ " here (see my comment below). This seems however not to be done consistent throughout the ms. I suggest to use $\delta^{15}\text{N}_{\text{collagen}}$, $\delta^{13}\text{C}_{\text{collagen}}$, $\delta^{34}\text{S}_{\text{collagen}}$, $\delta^{66}\text{Zn}_{\text{enamel}}$, $^{87}\text{Sr}^{86}\text{Sr}_{\text{enamel}}$ etc. consistently in the text, SI, table and figures.*

A: It has been done

- *Legend on the left:*

○ *You do not have a "group" of carnivores, herbivores, or unassigned humans anywhere (and Ind. 13? Hard to see), remove from legend accordingly.*

A: It has been done

○ *Change colors of key of the species according to their diet (i.e., the Canidae will always be red, so no need to have a black symbol here).*

A: It has been done

○ *Seems the species key is sorted by alphabet, but I can't detect a pattern in the key for the groups.*

A: It has been corrected.

○ *You could make clear that Ind. and "unassigned" are the humans in the legend. Maybe even indicate which one is adult, which one infant.*

A: We hope the new labels help answering this comment.

- *No need to specify $\delta^{15}\text{N}$ twice in the caption*

A: It has been corrected

- *Use Latin name throughout (e.g., for “fox”).*

A: It has been corrected

- *The differences in the spot size for adult and infant are difficult to see, think of different way to make this clear.*

A: We hope that the use of the polygons helps to address this comment.

- *Line 194: remove “in literature”.*

A: It has been removed

Line 202 and elsewhere: You only have two carnivores (which potentially are not even true carnivores), this is problematic when calculating TLS. I understand that you already sampled all available carnivore teeth (well done!), nothing to be done here. However, you need to tone down your argumentation here a bit. There is a large variation in e.g., N and Zn data in the herbivores (and humans, for that matter), and you have no control to understand these patterns in small carnivore dataset.

A: We thank the reviewer for this comment. according to Wilman et al 2014, *Vulpes vulpes* are feeding on 90% of animal foods, and the canids eating the smallest amount of meat present in the region in this time period (ref: <https://una-editions.fr/co-evolution-hommes-canides-en-afrique-du-nord-ouest-une-longue-histoire/>) still feed on 80% of meat. Most of the canids found at Tatoralt are *Canis aureus* that feed on 90% of animal foods. We therefore changed the word carnivore for meat-based, see lines 184, 228. In the caption, carnivore was replaced with Canids (see Figure 2 and Figure 3).

Still, most of the Zn comes from meat products, and fruits have very low Zn concentrations. Foxes feed on 90% animal foods and 10% fruits. Zinc isotopes of the dental enamel of a fox should be undistinguishable to that of a purely carnivorous animal.

The reviewer has a good point on the small dataset and we toned down our statements, see for example lines 260-262, line 228.

Line 204f: I do not think you only have considered teeth after weaning here, but also included permanent dentition that typically mineralize pre-weaning. See my comment above.

A: This has been corrected.

Line 219: This is not a $\delta^{66}\text{Zn}$ interpretation, this is an Interpretation of $\delta^{66}\text{Zn}$ data.

A: Thank you for your suggestion. It has been corrected.

Line 245ff: Here or at least in the SI you should state if you used the same (or some of the same) individuals (in sheep or humans) as Lee-Thorp (don't forget the dash in the name!). How big is their dataset? Did they use dentine or bone collagen? You state correctly that they the focused on sheep, but they also have Bos, Equus, Gazelle, how do these values compare? The main results of Lee-Thorp is that the Sector 10 humans did consume a reasonable amount

of animal resources, but no marine resources, which fits (partly to your conclusion). Elaborate!

A: Thank you for your valuable feedback and suggestions. We have taken your recommendations into account and made the necessary updates to provide information regarding the comparison with Lee-Thorp study (paragraph 4.3.3, Supplementary Information 4). We also referenced the supplementary information in the main paper to ensure that the readers can review the details.

Line 255: That cereals have generally higher $\delta^{15}\text{N}$ values compared to forage plants is a very important point for your study. Here I'm missing some info/discussion about this.

A: Thank you for your comment. The differences in isotopic ratios between the plants in the baseline diet of humans and the herbivores they consumed are indeed an important aspect of our research. We have addressed this concern by providing a more detailed explanation of the factors contributing to the observed variations in $\delta^{15}\text{N}$ values between grains and forage plants, as well as the potential implications for our study in the main paper (see lines 293-297).

Line 264: Should it say vegan instead of vegetarian? Do you have any evidence of animal products like dairy or eggs that are considered part of a vegetarian diet.

A: We thank you for this comment. While there is no evidence of milk or other dairy food consumption in the site, the presence of the ostrich eggshells in Iberomaurusian sites could indicate that they consumed eggs occasionally. However, you are correct that in this context, the word vegetarian is incorrect, and we changed it for “vegetalian” which only refers to the diet (whereas vegans also exclude animal products from other aspects of their life).

Line 266: So there are cut marks on the sheep? I think this is new information for the reader which is quite crucial.

A: We appreciate the reviewer’s comment. There are cut marks on many faunal species on the site but primarily on the Barbary sheep since it is the predominant species in the site. We have clarified that this supports the notion that animal proteins were also a part of the diet of this population at Taforalt (lines 246-249).

Line 270ff. This is another example where some crucial information is lacking in the data table and figures: Which sample is di2 and which one the long bone and the rib (both simply called “bone” in Table SI3, and not labeled at all in the figures)? Also, the info when these bones formed is in the text, but missing in the table (where is this information from? Please add reference!). In the table the “age of crown dev.” (development?) of the dm2 is at 10 months, you sampled the bottom of the tooth and estimate that this formed at the beginning of the first year. This is confusing, how does the “bottom” of a tooth form before the crown? Or did I misunderstand the abbreviation? Moreover, in the SI with the Scans of the tooth the collagen sampling is shown through the whole length of the tooth, not just the bottom.

A: We thank the reviewer for the valuable feedback regarding the clarity of information in our paper. The reviewer correctly pointed out the lack of clear identification of the formation of the dm2, long bone, and rib samples. To enhance clarity, we ensured that these samples were

clearly labeled in both the data table (Tables S6.3, S6.4), figure 4 and main text (line 312-315).

We would like to clarify the information regarding the 'age of tissue development' in Table S3.3 and S6.3 referred to in the manuscript line 174. In the table, we added two distinct columns related to the dm2 tooth: one for the formation time of the enamel sampled for Zn and Sr. The other columns are for the formation of the tissues sampled for collagen. For the dm2 as it was sampled as a whole for collagen we have estimated the age of between -0.34 and 1 year since the Individual died between 6 and 12 months.

Line 289: not clear what you refer to as a “broad-spectrum and dietary breath models”, elaborate.

A: We thank you for your comment regarding the clarity of the "broad-spectrum and dietary breadth models" in our manuscript. Following your comment, we have provided a comprehensive explanation of these models in our paper (lines 335-340)

Line 290: what do you consider as higher and lower ranking resources in the light of your study? Lower: everything but animal products? Only wild plants? Higher: meat only? Cereal? Explain! How does cooking (charred plant remains) play a role here?

A: Thank you for your comment. In this context, higher-ranking resources are exemplified by the large and medium ungulates, which include species such as the Barbary sheep and gazelles, that were the primary focus of exploitation at Taforalt. Lower-ranking resources encompass smaller game animals, such as small birds and lagomorphs, as well as wild plants. Specifically, at Taforalt, lower-ranking resources would include items like acorns, pine nuts, and some legumes.

Following your comment, we decided to deliberately refrain from using the terms 'high-ranked' and 'low-ranked' to avoid subjectivity in the classification of food resources. Instead, we have directly referred to the types of food resources.

Cooking, as indicated by the presence of charred plant remains, underscores the significance of preparation methods in making these lower-ranking resources more digestible and nutritionally valuable.

For Line 300: you mentioned the possible storage in baskets before (in the intro), I suggest to move the info about the Alfa grass up where you mention this observation first.

A: We have moved the information in the introduction as suggested.

Line 302: Reference for seasonality of plant maturity?

A: Thank you we added the reference (line 361) .

Concluding remarks:

Line 319. “Younger Dryas (YD)”: You should only introduce an abbreviation if you use it a few times afterwards. The only time you mention this term again (Line 327), you don't even use the abbreviation.

A: We thank the reviewer. Following your comment, we do not use the abbreviation anymore (lines 378 and 387).

Supplementary Information:

Generally, the SI is full of minor and major mistakes and written extremely sloppy. Please carefully read this again and correct mistakes and formatting errors throughout, I can't point them all out in here. Generally, the SI would greatly benefit from some reorganization. Report results of each analyzes in the respective section,

A: We apologize for the mistakes and formatting errors in the Supplementary Information. We have addressed all comments below in addition to other mistakes we noticed not mentioned by the reviewer. We have greatly improved the presentation, readability and language of the SI.

Line 22: instead of using "distant from village" indicate direction (e.g., northeast).

A: It has been done (line 37)

Line 37ff. There is a large age gap between the two units. Any idea why (erosional surface? Hiatus?)?

A: We appreciate your question about the age gap between the Yellow series (YS) and the Grey series (GS). We want to clarify that there is no significant age gap between these two units. The Yellow series commences at approximately 22,292 cal BP and continues uninterrupted until around 15,000 cal BP. The sedimentation change that marks the transition between these two series occurs around 15,190-14,830 cal BP.

We have taken your input into consideration and have modified the paragraph in the text to make it clearer to the reader (line 55-64).

Fig. S1.1: The maps in the lower right corner should be reworked. The first one is not really needed, as a map like this is in the main text (Fig. 1). The second, closer map does not help at all without any names, rivers etc. Maybe instead use the zoomed-in map with some labels, and then an even more zoomed-in map with the location of the cave within the village?

A: Thank you for your suggestion. We modified the maps accordingly (Figure S1.1).

Line 105: I suggest to use "preferred" instead of "favorite"

A: Thank you. We changed the word as suggested.

Line 107: Should be "was primary consumed by this population" instead of "was primary consumption by this population"

A: Thank you, the phrase has been corrected.

Line 110: punctuation mark missing

A: We added the punctuation mark.

Table S1.1: I suggest to sort these by abundance

A: We appreciate your feedback. We sorted the table by the abundance of the species.

127ff: I am missing information about possible diagenesis. How are your C/N ratios (you note these in Table S1.3, but do not mention this data at all. You need to show that you can eliminate the effect of possible overprint of the dietary signals.

A: Thank you for your comment regarding the potential influence of diagenesis and the need to address the possible overprinting of dietary signals. In response to this comment, we included a section in the manuscript that discusses the potential impact of diagenesis on the C/N ratios (lines 172 and 673).

Line 144: Explain what you mean with “different sources”

A: Thank you for your feedback regarding the need for clarification on what we meant by 'different sources'. We have taken your comment into consideration and have added an explanation in the text to make it clear (lines 197-204).

Line 154: refer to most important studies here (for bulk isotope analyzes)

A: We have cited now the most important studies for bulk isotope analyses in the section of bulk carbon and nitrogen isotopes.

Line 166: reference for this equation is missing here

A: Thank you we added the reference

Line 170: you stated the abbreviations for Phe and Val already above. Only do this once and then use abbreviations!

A: Thank you for pointing that out. We used the abbreviation as suggested.

Line 183: decide if you use “terrestrial C₃/C₄ consumers” or “C₄/C₃ terrestrial consumers” and stick with it

A: We decided to use the term “terrestrial C₃/C₄ consumers”. We have changed that throughout the text.

Line 174/175: does this really refer to all animals, or only herbivores?

A: Thank you for pointing out the potential confusion regarding the term 'animals.' To eliminate this ambiguity and ensure clarity, we have modified the text by replacing 'animals' with the term 'consumer.' This change accurately reflects that we are specifically referring to consumers in the context of our study, which includes herbivores and other organisms within the same ecological environment.

Line 211: By how much?

A: The requested modification has been made in the manuscript. We added the approximate difference in $\delta^{66}\text{Zn}$ values between herbivores and carnivores (0.30 to 0.60 ‰) with the appropriate literature citations.

Line 227: ...is associated with low...

A: Thank you, we have made the correction.

Line 235: the statement "This later is produced by the process of the radioactive decay of rubidium Rb" is repeated in a few lines down again.

A: Thank you for your feedback. We have revised the paragraph to eliminate the repetition of the statement.

Line 254f: "For humans, strontium is incorporated into their skeleton via diet". Isn't that also true for all other animals?

A: Thank you for pointing that out. Indeed, strontium is incorporated in humans and other animals. We have corrected it.

Line 267: should be: "...close to +20 ‰ which..."

A: We corrected the sentence.

Line 282: and elsewhere: Delete large numbers of random paragraph breaks

A: We appreciate your comment. We deleted the breaks.

Line 295: Here you correctly state that the sampled carnivores might not have a pure carnivorous diet, since many canid species can feed on a variable amount of plant material. This is very important for your interpretations and should definitely mentioned in the main text!

A: Thank you for your insightful comment regarding the importance of acknowledging that the sampled carnivores might not have a pure carnivorous diet. Therefore, we highlighted this aspect in our main text (Main text, line 226).

Line 297: you only talk about the carnivores here. What about the herbivores? Looks like they were all clearly identified?

A: Thank you for bringing attention to this omission. We have made the necessary modifications to the manuscript to ensure that the identification of herbivores is now included and discussed alongside that of carnivores (lines 370- 379).

Line 342: should be taxonomic identification

A: It has been done

Table S3.1: what is ID 34566, Bone or tooth?

A: Thank you for pointing that out. It is a tooth, we corrected it in the table.

Line 367: Here you state correctly that the hypoplasia can be the result of weaning, but you sample material across this line and still use it as “adult” diet if I understand correctly. Elaborate

A: Thank you for your insightful comment. We appreciate your feedback and have taken it into consideration. In response to your concern, we have elaborated on the potential causes of linear enamel hypoplasia, highlighting that it may not be solely a consequence of weaning. As indicated in our revised paragraph, we found the presence of linear enamel hypoplasia in teeth formed after the weaning period, such as the M2 and M3 molars (414-422). This finding suggests that factors beyond weaning, such as ongoing nutritional stress or traumatic events, could also contribute to the formation of these stress markers.

Additionally, this stress is also present in a di1 which is formed during the first year of life and is associated with a breastfeeding period.

Line 406: correct spelling of your own institute is with capital first letters

A: It has been corrected.

Line 413f: why did you not sample systematically in all teeth?

A: We appreciate the reviewer's question regarding the systematic sampling of teeth in our study. While we aimed for a comprehensive sampling approach, it's important to note that not all teeth were sampled systematically. The reason for this variation in sampling frequency was primarily due to the condition of the teeth themselves. In some cases, we encountered teeth with insufficient enamel (which we sampled for Sr too) along the height of the crown, often a result of wear. This has made it challenging to collect enamel samples without risking further damage to the teeth. We modified our paragraph to clarify this sampling strategy : paragraph 3.3.1.2.

Line 421: You mean the age of the individual at the time when the enamel was formed. Rephrase

A: We have rephrased the sentence as requested.

Line 431: did you only serial sample equids?

A: We also did serial sampling for some Barbary sheep, hartebeest, and the Rhinoceros. Following your comment, we have changed the sentence to include those species too section 3.3.2.

Line 457: You mean measured in duplicates? Was the enamel also pre-treated in two separate batches so you have true duplicates?

A: To address your query, we would like to clarify that we conducted duplicate measurements for each enamel sample. This means running the samples twice to ensure the reliability and precision of our isotopic ratio measurements. We have clarified this in the paragraph (line 516-517).

Line 458: report typical standard deviation

A: Thank you for your input, the typical standard deviation is reported now (lines 517-518).

Line 463: Did you use the same aliquot for Sr that you've used for Zn?

A: For our study, we sampled enamel for Zn and Sr separately and we used separate aliquots. Following your query, we have added a sentence to clarify this.

Line 466: correct to "...2 ml of 3 %..."

A: Thank you. It has been corrected.

Line 484: add units (%)

A: It has been corrected.

Line 485: name standards

A: We added the standard's name.

Line 537: you send it where? Or do you mean you measured it? Are you using the same setup as Jaouen et al., 2019?

A: We included the standard samples alongside our archaeological samples and sent the entire set to the University of Davis for amino acid analyses. This approach was consistent with the methodology used in Jaouen et al., 2019, where the same standard was analyzed with the experimental samples. Therefore, we followed a similar setup to Jaouen et al., 2019, by measuring the standards in the same analytical run as our samples to ensure comparability and consistency in the results.

Figure S4.1.: Add units to axis. Error bars are 1σ ?

A: We added the unit. The error bars are the standard deviation for the amino acids measurement (now S4.7).

Figure S4.2.: Delete unnecessary digits

A: We added the units and deleted the unnecessary digits (now Figure S4.8).

Line 601 and elsewhere: usually, three digits are sufficient to show significance. Use $p < 0.001$ here.

A: It has been modified.

Figure S4.3: Again, think about human vs. animal. Maybe use non-human fauna instead?

A: Following your comment, we decided to use the term fauna for non-human samples in our study.

*Line 621: You have two individuals here that you refer to as carnivores, right? A M1 from a *Vulpes Vulpes* and a M? from a *Canidae*. Therefore, a) the sample number ($n = 2$) is much too small to run these statistics, b) M1 most likely incorporates a nursing signal in the fox*

(who knows about the molar from the Canidae) and c) at least the fox and possible the other canid is not a “true” carnivore but often supplements its diet with plant materials etc. So, you have to tone down your interpretations/conclusions

A: We appreciate the reviewer's comment on the zinc isotopic compositions of M1 teeth in *Vulpes vulpes* and Canidae. We appreciate your feedback and would like to address your points accordingly (line 590):

- We acknowledge that our sample size is indeed small, as we are working with only two specimens. This limitation does affect the robustness of our statistical analyses, but as it has been mentioned, we did not have access to more individuals. We will make sure to highlight the small sample size and its implications in our paper.

- The formation of the M1 crown in the fox initiates during the prenatal phase and experiences calcification during the weaning period, which typically occurs around 41 days post-birth. This developmental timeline implies that the zinc value of the M1 (0.61‰) might be influenced by nursing. On the other hand, the M of the canidae shows lower Zn values indicating that it is unlikely to be affected by the nursing. The roots of the molar of both the fox and the Canidae are usually formed post-weaning thus reflecting adult diet. Zinc isotope ratios are therefore possibly impacted by mother milk consumption, which increases the $\delta^{66}\text{Zn}$. The fox might look more “herbivorous” and it actually is, due to these higher $\delta^{66}\text{Zn}$. We now make clear the influence of the mother milk consumption on the M1.

- As for the comment regarding the carnivorous nature of foxes (*Vulpes vulpes*) and Canidae. While we believe that these species are not pure carnivores, meat still makes up 90% of their diet, and fruits make up only 10% of their diet (Wilman et al., 2014). We therefore change the wording, replacing carnivore with meat-based diet. Moreover, we would like to mention that both zinc and nitrogen are preferentially absorbed from animal resources.

We incorporated these points into our discussion and made sure to tone down our argument regarding the diet of these specimens.

Line 624f: But even your permanent-tooth-dataset includes enamel that typically forms pre-weaning (see above)

A: We separated the teeth formed pre-weaning from those formed post-weaning. We now only used the values of enamel formed post-weaning for the interpretation of our data

Line 625: t missing in breastfeeding. Better, use the word nursing. Citation at wrong place (before the comma).

A: We changed it to nursing and corrected the citation place.

Line 627: Again, calculating a trophic spacing from a very small sample set without true carnivores with an adult diet is problematic to say the best...

A: Thank you for your comment. We have rephrased the paragraph to acknowledge the limited sample size of two carnivores with an adult diet. However, it is worth noting that the trophic spacing we calculated between herbivores and carnivores aligns with results from similar sites (line 598-602).

Figure S4.5: This is a good place to indicate all teeth that (might) include a nursing effect. If I am not mistaken, you use deciduous teeth only for the juvenile boxplot, but these should also include permanent teeth that typically form prior to weaning. In sheep, also only M2 and M3 form post-weaning, I think. Hence, the “juvenile” sheep data should include the incisors you sample, and the adult one only the M3 and M2 data. It looks like you included the incisors of the sheep also in the adult data. Also, fix y-axis label.

A: Thank you for your valuable feedback. We have revised the plot to better indicate teeth formed before weaning and those formed after weaning (now Figure S4.2).

Line 653ff: Move N results to N section

A: Thank you for the feedback. We have now separated the Zinc and the Nitrogen results into different sections (Supplementary Information 4).

Table S4.5: Confusing to have the N results in the Zn section.

A: We have moved the N results and plot to the Nitrogen results section.

Figure S4.6: Again, separate teeth into pre- and post-weaning (here and elsewhere, also for the non- human fauna)

A: We have separated the teeth into pre and post-weaning (now figure S4.3).

Figure S4.8: Looks like at least in Ind 1 a weaning effect is visible with lower Zn values in the I, C and P4 and higher ones in M2s (apart from the one M2 with a low Zn value. This is not clear in Ind. 5 and the unassigned individuals; this needs to be discussed. You have a tooth-type vs. Zn values figure below (Fig. S5.1), would be helpful to have that Fig. up here. Also, make the same Fig. with N data vs. tooth type.

A: Figure S4.8 is now Figure S4.4 and has been redone for more clarity We moved figure S5.1 to the Zinc section on results (Now Figure S5.1) and we created the requested figure of N data vs tooth type (Figure S5.2). We also now discuss the variations seen for each individual lines 634-642 and 993-992.

Line 683 and elsewhere (see also comment above): I suggest to use $\delta^{13}\text{C}_{\text{collagen}}$ (vs. $\delta^{13}\text{C}_{\text{Pro}}$ etc.) to make this clearer. If you do, make sure to change this everywhere, in this SI, but also in the main text and figures.

A: Thank you for the suggestion. We have decided to use $\delta^{13}\text{C}_{\text{collagen}}$ to make the text clearer.

Line 686: again, make sure you really only have adult diet here.

A: Following your comment, we now only included the teeth and bones formed post-weaning to estimate the adult diet

Line 690: It's an important info here that one canid seems to occupy a carnivore trophic level. State in main text?

A: We have stated this information in our main text. This concerns the fox.

Line 704: Decide if you use subscript for C₃ and C₄ or not and make the same throughout the text.

A: Thank you for your comment. We decided not to use the subscript.

Figure 765: The yellow data of Taforalt Humans is hard to see, change color scheme as this is the important data.

A: We have changed the color scheme so it would be easier to see (Figure S4.9 S4.10-S4.11-S4.12).

The lines are confusing. Looks like the line for “C3 humans” goes into the red field (C3 carnivore?) and the C3 herbivore Taforalt to the yellow field (Taforalt humans). Also, lines cross some of the labels, clean up. What is the orange group (“human”) here? If they are not C3, C4 r marine consumers or from Taforalt?

A: Thank you for your feedback. We have addressed your concerns by removing the confusing lines and deleting the "Human" group, as it did not belong to a specific diet category. The plot now features a cleaner design with a simplified legend that should be sufficient for interpretation (Figure S4.9 S4.10-S4.11-S4.12).

For Figure S4.10 and the following figures: Again, yellow hard to see. Needs at least dark outlines. Remove figure header, you also don't have any in the other figures. Add vs. which standard to axis titles.

A: We have made the requested changes. The data for Taforalt should be easy to see on the plot. We removed the figure header and added the standard to the axis titles (Figure S4.10).

Line 751: I think your sample size is too small for this statement as you only have one bone from one individual, right? This tells you nothing about the range and variability of this taxon.

A: Thank you for your valuable feedback. The sample size for this particular statement is indeed small, as it is based on a single bone from a single hare individual. We removed the statement.

Line 785: for one equid but all hartebeest? Clarify. Also, here and elsewhere: the common names are spelled with small first letter

A: We have clarified our paragraph (line 684-692). We used small first letter for the common names.

Line 794f: Is the TP when excluding the adult samples which formed pre-weaning?

A: The TP of the herbivores from Taforalt still ranges from 1.7-2.3 when excluding the samples formed pre-weaning. The teeth that could be affected by nursing still have a TP value of 2.1. this is the case of an equid sample and a barbary sheep sample

Line 800 f: why do you reject this?

A: We reject the possibility of suckling for the high values for those teeth formed pre-weaning because their TP is only 2.1. This value is lower than expected if nursing had a significant effect. We added the explanation in the paragraph (line 898)

Line 801: "Hartebeest, hare, the Hartebeest and the Rhinoceros"?

A: We apologize for the oversight in our manuscript where we inadvertently repeated the word "Hartebeest." This was indeed a typographical error. We made the necessary correction.

Line 815: Again, what about permanent teeth that form during nursing?

A: We corrected the sentence so that it shows the values recorded on teeth formed pre-weaning from those formed post-weaning (line: 908).

Line 839: Did you analyze all humans? Is this an average value?

A: For the CSIA, we analyzed the human that have yielded enough collagen for the analyses. We took the chance to clarify the number of samples and the average values in the paragraph 4.4.3.3.

Line 872ff: Obviously, reading your ms until here I was missing this section! Parts of this definitely needs to be included in the main text. However, your argumentation here does not convince me at all! What about the incisors and canines. Or did I misunderstand that you included their results also in your statistics and interpretations?

A: We now refer to this section in the ms, line 172, and changed all figures to have this effect clearly underlined. It is true that some pre-weaning data were mixed with post-weaning one in the statistics, and we therefore now only considered post weaning samples, and make it clear in the text.

Figure S5.1: Do these Rennes and Lapa do Santo studies also have $\delta^{15}N$ analyses? I would like to see the same plot for bulk collagen $d^{15}N$ and other analyzes you conducted on all tooth types.

A: There are no available $\delta^{15}N$ data for the sites of Rennes and Lapa do Santo. Regarding the plot, we made a similar plot with bulk collagen $\delta^{15}N$ vs the tooth type in each individual. Thank you for the suggestion (Figure S5.2)

Figure 5.2 (should be S5.2): This is not very helpful. A line for each individual sorted by tooth type as in the previous study would be more helpful.

A: Thank you for the recommendation we changed the box plot into a line for each individual sorted by tooth type (Figure S5.2)

Line 915: reference for equation?

A: We added the correct reference for the equation.

Line 917: "still"? Compared to what?

A: We are sorry for the confusion. We calculated the average TP level of all the adult humans using our zinc equation and we compared it to the TP equation for amino acids. We made sure to clarify this in the Supplementary information (line 1025).

Line 946: Do humans with low status usually die younger? The sampled individuals here all have died young, can you elaborate on that?

A: Following your comment, we added a paragraph elaborating on the age of death of this population and its relationship to their diet (section 4.6)

Line 973 and elsewhere: the unit (‰) belongs after the stdev

A: It has been corrected.

Line 2021: You state that “the shells possibly were used as ornaments or grave goods that accompanied the grave of the disease”. That’s what comes to mind first of course. But, why is that, so you have a reference or is this just a well-educated guess? As this is unimportant for your study, I would remove this sentence.

A: Thank you for your feedback. We have removed the sentence.

Line 1031: I agree that the sample is reflecting breastfeeding. But you have yet to convince me that early forming teeth of adults do not show the same.

A: The deciduous second molar forms earlier than than the M1, which root (which we sampled) formed after the age of 2.5. It explains why the pattern is different.

Excel Datasheet (some of these corrections/comment should be considered for all tables):
Table S11:

- *typo in header (study)*

A: Thank you in has been corrected

- *Gazelle is not a carnivore!*

A: It has been corrected, thank you

- *The Canidae is an M1, right? At least that’s what you state in Table S13. Please crosscheck all for to have correct data here!*

A: The fox is an M1 but the Canidae is an M.

- *Water intake after which study. Fill empty field of the one Hartebeest.*

A: We added the reference for the water intake in the table S11. The field for the hartebeest has been corrected.

- *Clearer if sorted by diet/taxa instead of SEVA ID.*

A: Thank you for the suggestion, we have sorted the table by taxa.

- *I guess the actual sample ID is #REF? Indicate which catalogue this refers to. Some of these IDs have a dot, others don’t, is this correct? Also check in other tables.*

A: The #REF is the ID given to the samples during excavation. We change the #REF to “ID excavation” to make it clear for the reader. We removed the dots from the IDs. It has been checked in the other tables.

- *Again, here you have a few mammalian teeth that typically form during nursing.*

These should be indicated and removed from or highlighted in certain data compilations.

A: We have added in Table SI4 the teeth that are affected by nursing for which isotopic analyses. We did not include the teeth that are formed pre-weaning in the data interpretation.

Table SI2:

- *Again, indicate material that typically forms pre-weaning*

A: We indicated the samples that are formed pre-weaning.

Table S3:

- *Refer to citations correctly, add year and doi.*

A: We added the citations in the last column with the year and doi.

- *Add units for all columns with data, not just some (e.g., add ‰ for $\delta^{66}\text{Zn}$, not ppm in the column with the concentration).*

A: Thank you, we added the units.

- *“What does the “dev” at Age crown dev. Stand for? Explain all abbreviations.*

A: Following your comment, we decided to change the name into “Age of initiation of the enamel” and “Age of completion on the sampled enamel”.

Table S4:

- *Add units,*

A: Done

- *Here you only have SEVA IDs, but it’s called “sample name”. Make consistent. You could add collection ID as well.*

A: We changed it to SEVA. Additionally, we added the ID of excavation as well in Table S6.3 and Table S6.4.

Table S5

- *Add units.*

- *Move caption above table.*

A: We added the units and moved the caption above the table.

Table S7:

- *Use same text size throughout*

A: It has been corrected.

- *The colors of each individual do not correspond to the color in Fig. 1. Choose one color scheme and stick without throughout the whole MS, including SI and Figures.*

The colors have been changed to match those in Figure. Thank you for the feedback.

General

- *Check consistency e.g., space between number and unit, the use of “cal. BP” vs. “cal BP”, using the delta notation once introduced*

A: It has been corrected. We switched to cal BP.

- *Search for double spaces and delete them*

A: It has been corrected

- *Delete spaces before citation ID*

A: It has been corrected

- *Table captions should be above the tables.*

A: It has been corrected

- *Make sure you introduce abbreviations when you first use the term in the text and SI. After that, stick with the same abbreviation and don't introduce them again.*

A: We corrected the abbreviations

- *Use a coma after e.g.*

A: It has been corrected

- *Do not mix British and American English.*

A: Thank you, we went through the manuscript to switch everything to British English

- *Element names are spelled with small starting letter. If you use the abbreviation, it's starts with a capital letter (e.g., nitrogen and N). Correct throughout ms.*

A: It has been corrected

- *Letters until 12 are spelled out if they are not followed by a unit*

A: We are not sure we understood this comment. Do you mean numbers?

- *Decide on capitalization of the headers and stick with it*

A: It has been corrected

- *Use same units. E.g., for hour you sometimes use h, sometimes hr, sometimes hrs and sometimes spell it out, I think. Make this consistent.*

A: It has been corrected

References cited in the response:

Bogaard, A., Heaton, T.H.E., Poulton, P., Merbach, I., 2007. The impact of manuring on nitrogen isotope ratios in cereals: archaeological implications for reconstruction of diet and crop management practices. *J. Archaeol. Sci.* 34, 335–343.

<https://doi.org/10.1016/j.jas.2006.04.009>

Fraser, R.A., Bogaard, A., Heaton, T., Charles, M., Jones, G., Christensen, B.T., Halstead, P., Merbach, I., Poulton, P.R., Sparkes, D., Styring, A.K., 2011. Manuring and stable nitrogen isotope ratios in cereals and pulses: towards a new archaeobotanical approach to the inference of land use and dietary practices. *J. Archaeol. Sci.* 38, 2790–2804.

<https://doi.org/10.1016/j.jas.2011.06.024>

Fraser, R.A., Bogaard, A., Schäfer, M., Arbogast, R., Heaton, T.H.E., 2013. Integrating botanical, faunal and human stable carbon and nitrogen isotope values to reconstruct land

use and palaeodiet at LBK Vaihingen an der Enz, Baden-Württemberg. *World Archaeol.* 45, 492–517. <https://doi.org/10.1080/00438243.2013.820649>

Hedges, R.E., Reynard, L.M., 2007. Nitrogen isotopes and the trophic level of humans in archaeology. *J. Archaeol. Sci.* 34, 1240–1251.

Turner, E., 2020. Large mammalian faunal assemblages, in: Barton, R.N.E., Bouzouggar, A., Collcutt, S.N., HUMPHREY, L. (Eds.), *Cemeteries and Sedentism in the Later Stone Age of NW Africa: Excavations at Grotte Des Pigeons, Taforalt, Morocco*, Monographien Des Römisch-Germanischen Zentralmuseums. Monographien des Römisch-Germanischen Zentralmuseums, pp. 239–308. <https://doi.org/10.11588/propylaeum.734>

Wilman, H., Belmaker, J., Simpson, J., de la Rosa, C., Rivadeneira, M.M., Jetz, W., 2014. EltonTraits 1.0: Species-level foraging attributes of the world's birds and mammals: Ecological Archives E095-178. *Ecology* 95, 2027–2027.

Reviewer 2

The manuscript "*Earliest isotopic evidence of high reliance on plant food in the Late Pleistocene hunter-gatherer population (Taforalt, Morocco)*" by Ms Moubtahij and co-authors, which has been submitted for publication in Nature Ecology & Evolution lies within the scope of this journal. This research deals with the issue on the origin of agricultural-based economy and the role of increased reliance on plant foods in the Late Pleistocene hunter-gatherer population of Taforalt (Morocco) with parallel to the Natufian experience. In order to reconstruct the dietary habits and the mobility pattern of the human groups from Taforalt, the authors conducted for the first time a strong multi-isotope approach which combines zinc ($\delta^{66}\text{Zn}$) and strontium ($^{87}\text{Sr}/^{86}\text{Sr}$) analysis on dental enamel, carbon ($\delta^{13}\text{C}$), and nitrogen ($\delta^{15}\text{N}$) isotope analysis on dentin and bone collagen, along with single amino acid analysis, which makes this manuscript highly important. The isotopic analyses were conducted on a high-resolution data containing human remains and associated fauna recovered in the recently excavated burials from sector 10 at Taforalt. The study results emphasize to us the adoption of a starchy diet along with a decreased consumption of meat compared to Upper Palaeolithic sites with available isotopic data in Europe and Asia. Furthermore, the trophic position (TP) values at Taforalt are similar to the TP values of Neolithic farmers from the Levant, which enhances the evidence for substantial plant consumption, thus challenging traditional ideas about pre-Neolithic hunter-gatherer societies.

Overall, the manuscript is well-written. Data collection is adequately and openly presented in sufficient detail with additional information structured into 7 chapters provided in the supplementary information. The literature cited is very informative and relevant to the topic of the current manuscript. All figures are appropriate and the statistical tests are displayed with accuracy. The argumentation is well stated as it is clearly indicated in the abstract. However, some assumptions may need clarification or require to be substantiated by references. Here are listed the main points along with the comments:

Lines 95-97 The definition given by the authors to the Iberomaurusian hunter-gatherers needs to be enhanced by appropriate references. Here are some comments/suggestions: the production of blades and bladelets is not only peculiar to the Iberomaurusian knappers. Instead, a term such as bladelet-based stone technology is more appropriate.

A: Thanks for your feedback! We've incorporated your suggestion and now refer to the stone technology as "bladelet-based stone technology". We appreciate your input and believe this change enhances the clarity and precision of our terminology (line 94).

(1) there is no agreement between scholars as for the longevity of the Iberomaurusian (whether the end of the Iberomaurusian falls within Terminal Pleistocene or rather coincides with the onset of the Holocene)¹.

A: We appreciate the reviewer's comment. In response, we have revised the sentence and incorporated the appropriate reference (line 96).

(2) In regard to the age 25,000 cal BP, the authors may refer to scholars who have published the lithic record of Tamar Hat basing on their personal examination ^{2,3}.

A: We thank the reviewer for the suggestions. In response, we have included the relevant references (line 96)

Line 119: No burials were found at Tamar Hat. Only 6 deciduous teeth were recovered and they belong to individuals who were not buried in this site⁴. Furthermore, I suggest the term sheltered sites instead of caves, since Afalou is a rockshelter.

A: Thank you for your valuable feedback. We have addressed your comment by removing Tamar Hat as a site with a cemetery. Furthermore, we've incorporated your suggestion and adjusted the terminology to now refer to 'sheltered sites' instead of 'caves' (line 126).

Line 128: Afalou bou Rhummel burials (layer IV) are older than those of Taforalt (and it is likely that layer V burials are of a similar age or slightly older⁵. If they are proven not to be, the authors need to provide the reference.

A: We now refer to the cemetery of Taforalt as one of the oldest cemeteries in North Africa. We believe this adjustment accurately reflects the available evidence.

Lines 293-296: This research emphasizes the role of highly reliance on cariogenic wild plant food and teeth carries. Maybe it would be germane to also mention that dental evulsion (which has not addressed in this research) is not believed to be linked directly to oral pathology⁶. Furthermore, I recommend that the authors mention (maybe in the introduction) the archaeological records concomitant with the LGM showing evidence for plant processing. I refer in particular to Ifri el Baroud where small seeds of wild legumes may have probably been gathered for consumption purposes⁷, or Tamar Hat where wild seeds of pine cones with pestle-grinders were recovered⁴. It is also relevant to cite Ohalo II, a late Upper Paleolithic site in the Levant where the earliest known usage of plants ca. 23,000 years ago has been reported⁸ (Weiss et al. 2008). In all three cases, no substantial use of edible plants has been evidenced, but this will better support the relevance of the study results of the current manuscript.

A: Thank you for your comments.

- We appreciate your suggestion to mention dental evulsion. We now mention this in line 349-350.

- Additionally, we have incorporated archaeological records concomitant with the Last Glacial Maximum (LGM) in the introduction, citing Ifri el Baroud, Tamar Hat (line 108), and Ohalo II (line 72). We believe that these references provide valuable context to support the relevance of our study results, as suggested.

Lines 300-302: According to the statement of this paragraph, year-round assumption at Taforalt relies on the availability of edible plant-harvesting and possibly storage. However, relevant signs of seasonality are evidenced from selective hunting of Barbary

sheep and collection/consumption of edible molluscs⁹. Authors may provide further clarification to better enhance the increased sedentism assumption at Taforalt.

A: We have revised our argumentation for the year-round assumption at Taforalt, presenting it as a possibility rather than a certainty given the limited data on this matter. Thank you for your constructive feedback (line 366-376).

Lines 303-305: Yes, the population of Taforalt increased their reliance on plant resource, but there is no significant change in the availability of higher-ranking foods, if we assume that the local (and primary) game Barbary sheep is the highest-ranked resource, and then the main source of protein⁹. Besides, Taforalt burials were directly dated to 15,077 - 13,892 cal. BP (cited in this manuscript), which coincides with the rapid warming period GSI. In this case, why Late Iberomaurusian populations in Taforalt have shifted to a plant-based economy if there are no signs of population experiencing resource stress, or climatic deterioration? At another level, there are no records of flakes or blades with macroscopic gloss, and none of the microliths showed evidence of any mastic or remains of hafting materials⁹, although the prevalence of charred plant materials. I understand these issues are beyond the scope of this manuscript, but it would be relevant to briefly raised them in the text to enhance the argumentation.

A: We appreciate the reviewer's insightful comment regarding the availability if higher-ranked foods. We acknowledge that there is no decline in the higher-ranking foods (Barbary sheep). We now addressed this aspect (line 370). Thank you for your valuable input.

Here are additional remarks:

The title: I would rather suggest to mention the term Later Stone Age (already used in the literature) instead of Late Pleistocene (c. 129,000 and c. 11,700 years ago), since the collected data belong to the Iberomaurusian which covers a very small part of the Late Pleistocene. The term Pre-Neolithic, already mentioned in this paper is another option.

A: Thank you for the suggestion. We changed the 'Late Pleistocene' to 'Later Stone Age' in the title.

Line 87: It would be useful to also cite the new paleogenomic study conducted on Epipalaeolithic-Middle Neolithic human remains from Morocco (Simões et al. 2023) which concluded to a mosaic of incoming groups admixing with local people¹⁰.

A: We thank the reviewer for the suggestion. We incorporated this reference into our paragraph (line 87)

Line 99: References are needed to assert the idea of the absence of domestication during the Iberomaurusian¹¹⁻¹³.

A: We now included the appropriate reference (line 100).

Line 131: the sentence should be “...warming period following (and not during) the Last Glacial Maximum (LGM)”. Also, Poti et al. 2019 are cited instead of Barton et al. 2019.

A: We thank the reviewer for pointing out the errors in our citation and sentence structure. We corrected the citation and the sentence (line 141).

Line 152: remove the repeated word.

A: We removed the repeated word

Line 200: I think a word is missing here.

A: We revised the missing words

Line 320: Authors need to add uncal BP to the age of Younger Dryas.

A: We added the uncal BP to the age of Younger Dryas (line 391).

Supplementary Information 1 - Line 108: It is worth mentioning that Barbary sheep horn cores used as funerary objects accompanying human burials in sector 10 at Taforalt have also been reported at Afalou bou Rhummel^{14,15}. This is to raise the idea that some common and synchronous idiosyncratic practices occurred in different biogeographic entities during the Late Iberomaurusian.

A: Thank you for this suggestion, we have included it in the paragraph

Reviewer's decision

This is an interesting and original paper. As reported by the authors, there is no compelling evidence at Taforalt for the precocious domestication of either plants or animals. However, the current research demonstrated the abundance of plant resources in the environment of the Pre-Neolithic groups led to a greater concentration of foraging for lower-ranking resources. Further research into coastal sites is needed to distinguish between dietary habits rigorously determined by different local environments. I expect this paper will draw interest from researchers in different fields who are interested in the origin of farming. Therefore, I would highly recommend publishing the manuscript after the minor corrections which have been noted above.

References

- 1. Hogue, J. The origin and development of the Pleistocene LSA in Northwest Africa: A case study from Grotte des Pigeons (Taforalt), Morocco. Oxford: University of Oxford, PhD thesis (2014) https://ora.ox.ac.uk/objects/uuid:3a95e8b0-5e0b-4e8b-baac-b2e44a327d8a/download_file?file_format=application%2Fpdf&safe_filename=THE_SIS01&type_of_work=Thesis*
- 2. Close, A. E. The Iberomaurusian sequence at Tamar Hat. Libyca 29, 69–103 (1981).*
- 3. Sari, L. Technological change in Iberomaurusian culture: The case of Tamar Hat, Rassel and Columnata lithic assemblages (Algeria). Quaternary International 320, 131–*

142 (2014).

4. Saxon, E. C., Close, A., Cluzel, C., Morse, V., Shackelton, N. J. *Results of recent investigations at Tamar Hat. Libyca* 22, 49–91 (1974).
5. Hachi, S. Fröhlich, F., Gendron-badou A., de Lumley H., Roubet C., Abdessadok S. *Figurines du Paléolithique supérieur en matière minérale plastique cuite d'Afalou Bou Rhummel Bou Rhummel (Babors, Algérie) : premières analyses par spectroscopie d'absorption infra-rouge. L'Anthropologie* 106, 57–97 (2002).
6. Humphrey, L. T. et al. *Earliest evidence for caries and exploitation of starchy plant foods in Pleistocene hunter-gatherers from Morocco. Proc. Nat. Acad. Sci.* 111, 954–959 (2014).
7. Potì, A. et al. *Human occupation and environmental change in the western Maghreb during the Last Glacial Maximum (LGM) and the Late Glacial. New evidence from the Iberomaurusian site Ifri El Baroud (northeast Morocco). Quat. Sci. Rev.* 220, 87–110 (2019).
8. Weiss, E., Kislev, M.E., Simchoni, O., Nadel, D. & Tschauner, H. "Plant-food preparation area on an Upper Paleolithic brush hut floor at Ohalo II, Israel". *Journal of Archaeological Science* 35 (8), 2400–2414 (2008).
Bibcode:2008JARSc..35.2400W. doi:10.1016/j.jas.2008.03.012.
9. Barton, R. N. E. et al. *Origins of the Iberomaurusian in NW Africa: New AMS radiocarbon dating of the Middle and Later Stone Age deposits at Taforalt Cave, Morocco. Journal of Human Evolution* 65, 266–281 (2013).
10. Simões, L.G., Günther, T., Martínez-Sánchez, R.M., Vera-Rodríguez, J.C., Iriarte, E., Ricardo Rodríguez-Varela, Bokbot, Y., Valdiosera C. & Jakobsson, M. *Northwest African Neolithic initiated by migrants from Iberia and Levant. Nature* 618, 550–556 (2023).
11. Klein, R. G., & Scott, K. *Re-analysis of faunal assemblage from the Haua Fteah and other late quaternary archaeological sites in Cyrenaican Libya. Journal of Archaeological Science* 13, 515–542 (1986)
12. Merzoug, S. & Sari, L. *Re-examination of the Zone I Material from Tamar Hat (Algeria): Zooarchaeological and Technofunctional Analyses. Afr. Archaeol. Rev.* 25, 57–73 (2008).
13. Turner, E. *Large mammalian faunal assemblages. in Cemeteries and Sedentism in the Later Stone Age of NW Africa: Excavations at Grotte des Pigeons, Taforalt, Morocco 239–308 (Monographien des Römisch-Germanischen Zentralmuseums, 2019).*
14. Arambourg, C., Boule, M., Vallois, H. & Verneau R. *Les grottes paléolithiques des Beni Ségoual (Algérie). (Archives de l'Institut de Paléontologie Humaine (Édition Masson (1934).*
15. Hachi, S. *Résultats des fouilles récentes d'Afalou Bou Rhummel (Bédjaïa, Algérie). in El Món Mediterrani Despres del Pleniglacial (18.000-12.000 BP) (Ed. (ed. Fullola, J.-M. & Soler, N.) 77–92 (Girona : Museu d'Arqueologia de Catalunya, 1997).*

A: The references recommended by the reviewer have been added

Decision Letter, first revision:

9th January 2024

Dear Dr. Moubtahij,

Thank you for submitting your revised manuscript "Earliest isotopic evidence of high reliance on plant food in the Later Stone Age hunter-gatherer population (Taforalt, Morocco)" (NATECOLEVOL-23061316A). It has now been seen again by the original reviewers and their comments are below. The reviewers find that the paper has improved in revision, and therefore we'll be happy in principle to publish it in Nature Ecology & Evolution, pending minor revisions to comply with our editorial and formatting guidelines.

[REDACTED]

Reviewer #1 (Remarks to the Author):

The authors addressed all my comments carefully and explained their changes detailed in the rebuttal. Most of my suggestions were implemented. Especially, the timing of tooth formation regarding nursing and hence milk consumption is now clear, this was the most important correction to be made. In my opinion, the manuscript now meets publications standards. I have no further suggestions for improvement.

Reviewer #2 (Remarks to the Author):

The revised manuscript entitled "Earliest isotopic evidence of high reliance on plant food in the Later Stone Age hunter-gatherer population (Taforalt, Morocco)" is now easier to follow based on feedback from the reviewers. I am satisfied with the author's responses to comments and issues raised in the initial review. Indeed, the authors have addressed point-by point the technical and editorial issues raised by the reviewers. I recommend that the revised paper be accepted.

Our ref: NATECOLEVOL-23061316A

18th January 2024

Dear Dr. Moubtahij,

Thank you for your patience as we've prepared the guidelines for final submission of your Nature Ecology & Evolution manuscript, "Earliest isotopic evidence of high reliance on plant food in the Later Stone Age hunter-gatherer population (Taforalt, Morocco)" (NATECOLEVOL-23061316A). Please carefully follow the step-by-step instructions provided in the attached file, and add a response in each row of the table to indicate the changes that you have made. Please also check and comment on any additional marked-up edits we have proposed within the text. Ensuring that each point is addressed will help to ensure that your revised manuscript can be swiftly handed over to our production team.

****We would like to start working on your revised paper, with all of the requested files and forms, as soon as possible (preferably within two weeks). Please get in contact with us immediately if you anticipate it taking more than two weeks to submit these revised files.****

In recognition of the time and expertise our reviewers provide to Nature Ecology & Evolution's editorial process, we would like to formally acknowledge their contribution to the external peer review of your manuscript entitled "Earliest isotopic evidence of high reliance on plant food in the Later Stone Age hunter-gatherer population (Taforalt, Morocco)". For those reviewers who give their assent, we will be publishing their names alongside the published article.

Nature Ecology & Evolution offers a Transparent Peer Review option for new original research manuscripts submitted after December 1st, 2019. As part of this initiative, we encourage our authors to support increased transparency into the peer review process by agreeing to have the reviewer comments, author rebuttal letters, and editorial decision letters published as a Supplementary item. When you submit your final files please clearly state in your cover letter whether or not you would like to participate in this initiative. Please note that failure to state your preference will result in delays in accepting your manuscript for publication.

Cover suggestions

We welcome submissions of artwork for consideration for our cover. For more information, please see our guide for cover artwork.

Nature Ecology & Evolution has now transitioned to a unified Rights Collection system which will allow our Author Services team to quickly and easily collect the rights and permissions required to publish your work. Approximately 10 days after your paper is formally accepted, you will receive an email in providing you with a link to complete the grant of rights. If your paper is eligible for Open Access, our

Author Services team will also be in touch regarding any additional information that may be required to arrange payment for your article.

Please note that *Nature Ecology & Evolution* is a Transformative Journal (TJ). Authors may publish their research with us through the traditional subscription access route or make their paper immediately open access through payment of an article-processing charge (APC). Authors will not be required to make a final decision about access to their article until it has been accepted. Find out more about Transformative Journals

Authors may need to take specific actions to achieve compliance with funder and institutional open access mandates. If your research is supported by a funder that requires immediate open access (e.g. according to Plan S principles) then you should select the gold OA route, and we will direct you to the compliant route where possible. For authors selecting the subscription publication route, the journal's standard licensing terms will need to be accepted, including [a href="https://www.nature.com/nature-portfolio/editorial-policies/self-archiving-and-license-to-publish"](https://www.nature.com/nature-portfolio/editorial-policies/self-archiving-and-license-to-publish). Those licensing terms will supersede any other terms that the author or any third party may assert apply to any version of the manuscript.

[REDACTED]

[REDACTED]

Reviewer #1:

Remarks to the Author:

The authors addressed all my comments carefully and explained their changes detailed in the rebuttal. Most of my suggestions were implemented. Especially, the timing of tooth formation regarding nursing and hence milk consumption is now clear, this was the most important correction to be made. In my opinion, the manuscript now meets publications standards. I have no further suggestions for improvement.

Reviewer #2:

Remarks to the Author:

The revised manuscript entitled "Earliest isotopic evidence of high reliance on plant food in the Later Stone Age hunter-gatherer population (Taforalt, Morocco)" is now easier to follow based on feedback from the reviewers. I am satisfied with the author's responses to comments and issues raised in the initial review. Indeed, the authors have addressed point-by-point the technical and editorial issues raised by the reviewers. I recommend that the revised paper be accepted.

Author Rebuttal, first revision:

Reviewer 1

In their manuscript, Moubtahij et al. use a multiproxy geochemical approach to reconstruct the diet of a Late Pleistocene hunter-gatherer population in Morocco. They focus on the onset of agriculture and if the population relied on local foods, the proportion of plant foods in their diet, predating the advent of agriculture by several millennia. While I overall agree with their findings and these data could potentially be of great interest for a broad readership, the presented manuscript is lacking some important aspects in the results and discussion. Moreover, especially the SI has a large number of grammatical, structural errors, typos, structural issues and other rather careless mistakes.

Major comments:

In your sample set (human and non-human) are some permanent (“adult”) teeth that typically mineralize pre-weaning (e.g., canines, some incisors, see your Table SI3...). These materials (enamel and dentin) are therefore affected by the nursing effect, which is especially pronounced in $\delta^{15}N$. There is a rather small section dedicated to weaning towards the end of the SI, but as I understand this, these permanent teeth are still part of the “adult diet” reconstructions. The authors should elaborate on this, should clearly mark tooth-type/nursing etc. in the figures and exclude this data from general reconstructions.

A: Thank you for your valuable feedback. Following your comment, we have taken several steps to improve the clarity of our data presentation and address the potential impact of nursing on teeth formed pre- and post-weaning. In Supplementary Information 3, Table S6.3, Table S6.4 and the figures, we have now separated the teeth that mineralize pre-weaning from those that mineralize post-weaning.

Additionally, in Supplementary Information 3, we have incorporated a paragraph that elaborates on our categorization of teeth and the potential influence of nursing (3.3.1.1), both for human samples and for certain species of fauna. We recognize that the available literature for fauna is limited, and therefore, our approach is based on the best available data.

In our interpretation of the adult diet, we have exclusively utilized values unaffected by nursing.

The text, especially the SI would benefit from some restructuring, reorganizing and improving the language. For example, line 162ff, mention first what you analyzed (humans, equids, hare...), which material (dentin, enamel, bone) and then which samples were analyzed for what (CT-scans of human teeth only (?)...). Moreover, the SI is very hard to read, the titles are misleading and there are a lot of errors. The authors should work on this carefully to meet publication standards. I indicated some issues in this review, but it was just too many to keep track.

A: We thank you for the comprehensive review. The text and especially the SI have been reorganized and the language has been improved thanks to the native speakers who are co-authoring the manuscript. We restructured the paragraph at line 162 (now line 173-184) to provide a clearer presentation of the analyzed elements, materials used, and details of each

nature portfolio

sample. We have further gone through the SI Material in its entirety addressing all comments made by the reviewer (see below) with great care and beyond that we have improved the presentation, language and structuring of the SI Material in general.

Some very important points in this study seem to be just mentioned as a side note despite the fact that many of the interpretations rely on this. Here I'm missing some info/discussion to elaborate important aspects. For example, the authors state that cereals have generally higher $\delta^{15}\text{N}$ values compared to forage plant, and use only one reference for this statement. Is this systematically true in different habitats (i.e., including Northern Africa)? How do we know this was true already for paleolithic cereal, when agriculture was potentially very young (i.e., is this an effect of fertilizer today? Did this isotopic enrichment increase over time with evolving cereal?). What causes these differences in wild plants vs. cereal? Did the sampled herbivore have access to these cereals? In the same paragraph they casually mention cut marks on the sheep, again an important factor. Not clear in the text if there is evidence for domestication of these sheep or if they were probably wild. These are two examples where the manuscript would benefit from some clear information/discussion.

A: We appreciate the reviewer's comments and feedback. We would like to clarify our statements:

3) Regarding the first example, we indeed acknowledge the potential difference in $\delta^{15}\text{N}$ values between human-consumed grains and animal-consumed plants. This distinction highlights an important aspect of isotopic studies, a topic that has been discussed by Hedges and Reynard (Hedges and Reynard, 2007). The common assumption is that local herbivores would exhibit $\delta^{15}\text{N}$ values reflecting the local vegetation (since they were not domesticated), and subsequently, the dietary plant proteins of humans. However, it is essential to recognize that various factors can contribute to variations between the isotopic values of plants consumed by humans and those of herbivores:

- The charring effect can result in an increase of up to 2‰ in the $\delta^{15}\text{N}$ values (Bogaard et al., 2007; Fraser et al., 2013, 2011), which primarily affects plants consumed by humans and not those by herbivores.
- Manuring can indeed lead to higher $\delta^{15}\text{N}$ values in plants, however, we want to clarify that our text does not make any statement suggesting such enrichment. Instead, our emphasis is on the differentiation of $\delta^{15}\text{N}$ values in baseline plants. We also want to clarify that no evidence of early cultivation is detected at the site of Taforalt.
- We added a new statement to answer your comment lines 293-297.

4) In response to the comment about the cut marks on the Barbary sheep, we would like to emphasize that at no point in our text did we claim that the Barbary sheep was domesticated, nor is there any evidence to suggest that this species was domesticated in Taforalt or other Iberomaurusian sites. The presence of cut marks on Barbary sheep is undeniable, but it is important to note that similar cut marks were also identified on other species recovered from the Iberomaurusian levels at Taforalt.

To address this, we explicitly stated in our introduction and discussion that the zooarchaeological evidence indicates that the Taforalt population primarily relied on Barbary sheep but also consumed other ungulates (Lines 110-112 and 305-306). We appreciate the opportunity to clarify this point and we ensure that our revised manuscript includes a more comprehensive discussion of these factors.

I am confused about the sulphur analyzes. They state in the SI that only three samples had enough collage for measurement, then replot “a sheep”, “an equid” and “the humans”. Is this only one human or many (all) of them? In Table SI3 I can only find one number for d34S (unassigned human #34561). I cannot find the S data anywhere else, there is also no plot etc., please explain and make clear what the results/interpretations are (i.e., sea spray effect). And show the data in a plot or clear table.

A: In response to the reviewer's feedback, we have integrated a dedicated section in the Supplementary Results (lines 927-951) presenting and discussing the few $\delta^{34}\text{S}$ data and our interpretation and featuring a plot illustrating the sulfur isotope values for the three samples with sufficient collagen. Notably, we emphasize the limited interpretative scope due to the small sample size, a concern explicitly addressed in the revised paragraph (Supplementary Information 4). We have corrected the spelling mistake ‘humans’ to ‘human’. We have also provided the sample ID for each of the three $\delta^{34}\text{S}$ analysed samples in the caption of the Figure 4.12.

Minor comments:

Abstract:

Information of the geological age of these hunter-gatherer population is missing. “Late Pleistocene” is stated in the title and abstract, but a detailed age is only given much later, in the intro (Line 131pp.).

A: Thank you very much for noting this. We have added that the human remains were dated to 15,000 - 13,000 cal BP in the abstract (line 38-39).

Sulphur was also measured, but not mentioned in the abstract.

A: Following your suggestion, we now mention sulphur isotope analyses in the abstract (line 41)

Line 38: many Nature journals ask authors to refrain from making priority or novelty claims. Check if you can use “first” here.

A: Thank you for your comment. In line with Nature's policy, which permits priority claims in fields like archaeology when verified through peer review, we believe it is appropriate to include the claim of being the first in our study. For now, we would thus prefer using the term but are of course willing to refrain from using it should the reviewers or the editor advise against it.

nature portfolio

Introduction:

Figure 1: This is a large, almost empty map and pretty useless. It could profit from some more information which you mention in the text (e.g., indicating the region of the Natufian in the Near East, and/or showing the Upper Paleolithic sites in Europe or Asia you compare your AA data with, etc.).

A: We thank the reviewer for the valuable suggestions. Following your recommendation, we included the other sites mentioned in the text (Figure 1, Main text).

Lines 107, 116, 157, 165, 219 and elsewhere: the word “also” is often redundant. Check and delete wherever appropriate.

A: We have carefully reviewed the text and have made the necessary revisions eliminating the word ‘also’ where redundant.

Line 120: Refer to Fig. 1 when you mention the sites you show in the figure.

A: It has been done.

Line 145: you mention “specific individuals”, but should make clear that you not only analyzed humans, but also coexisting fauna that is part of the local food web.

A: We appreciate the reviewer’s comment. We have clarified that our analysis encompasses not only specific individuals but also coexisting fauna. (line 150-152)

Line 144 and elsewhere: The “see” in See Supp. Info. should start with a small s or should be removed.

A: Thank you, the “see” has been removed.

Line 145f: The manuscript would benefit from a short explanation what these baseline variations could be and note that you analyzed non-human faunal remains as well (see my comment on line 146, too).

A: We thank the reviewer for this suggestion. We have added a brief explanation of the baseline variations (lines 153-155), which can include natural variations in stable isotope ratios within the local environment. Additionally, we have explicitly mentioned that we analyzed both human individuals and faunal remains (line 172).

Line 158: this is the first time you mention Sulphur and is thrown in a bit as a surprise in this section. This should be in abstract etc as well.

A: Thank you for this comment. We have incorporated sulfur into the abstract to provide a comprehensive overview of our analytical approach (line 41).

Line 162: Make clear if you really scanned all samples. The bones, too? And did you scan the faunal tooth (and bone) remains as well, or just human teeth?

nature portfolio

A: Our analysis involved the scanning of human teeth exclusively, and we have updated the manuscript to reflect this accurately. We thank the reviewer for helping us clarify this aspect of our methods (lines 171-176)

Line 164f: The manuscript could benefit from an overview table with number of individuals and which geochemical analyzes were performed on which element, e.g., a compiled table S3.1 and 3.2.

A: Following your recommendation we added an overview table in the Excel sheet (Table S6.11) showing the number of individuals and the different geochemical analyses performed for the different isotope systems. We refer to this additional table line 174.

168: Are there any indicators that these animals were used by the humans and part of the same local food web?

A: The animal species analyzed in our study are all from Sector 10 or Sector 8 of the site. Study on the remains shows that these species were deliberately used by the humans. They all show evidence and marks of butchery activities (Turner, 2020). Following your comment, we now clarify this in the main paper line 178.

Line 174f and elsewhere: Be careful with dividing your samples into “human” and “animal” as humans are also animals. Maybe use other wording like non-human fauna or something

A: Thank you for your valuable suggestion. We have revised the terminology. We now use the term “Fauna” to distinguish the “non-human” samples from the human samples.

Results and Discussion:

Figure 2: This figure needs some work to present the data in a way that's easy to understand for the reader.

- *Have you considered performing statistical analyzes of this data? I know your samples size are small in some cases, but if you group accordingly, (e.g., all (adult) humans vs. all herbivores), your sample size might be big enough for some statistical sound implications of your findings using e.g., multivariate isotopes comparisons corrected for small sample size using e.g., R SIBER after Jackson et al., 2011 (doi:10.1111/j.1365-2656.2011.01806).*

A: We thank the reviewer for their detailed comments on Figure 2. Following their suggestion, we used ellipses similar to SIBER and realized that it was equivalent to produce 95% confidence ellipses and geometric shape similar to what stat_chull does, so this is what we did to produce new subfigures, the Figures 2C and 2D of the new Figure 2.

- *Error bars are missing. The caption states that each point corresponded to the average value of a single tooth or bone sample. However, it is not clear if you ran samples in duplicates or have multiple samples from a single tooth/bone. This is explained in the SI, so refer to this.*

A: We are sorry for not adding the error bars earlier. We now produce two figures using the same isotope systems. Figure 2A and 2B show every single sample with line connected samples belonging to the same individuals, and Figure 2C and 2D show the average values with the error bars. We further included the analytical errors in the Figure caption.

- *I am aware that Sr data is not very variable, and that's the point of plot B, I guess. However, you could reduce the size of the axis (e.g., span between 0.708 and 0.710 only) so the data is not plotted too much on top of each other and one can see the individual data points.*

A: Thank you for this suggestion. It has been corrected on all figures.

- *Not clear in which order the datapoints of the individual hominins are connected. According to eruption (mineralization) age? Clarify. Also note my comment about combining dental enamel or dentin of permanent dentition that forms pre- and post-weaning, especially with the incisors. Once you cleaned up this figure It might be feasible to show which teeth which presumable form pre- vs. post weaning?*

A: This is an excellent suggestion. The symbols used on the new Figures 2A and B indicate which sample is formed post or pre-weaning.

- *You color scheme could be improved, see also my comment of the Figures in the SI. The green- yellow of Ind. 9 and 14 are difficult to distinguish and Ind. 13 generally hard to see. Maybe a black outline of the cycles would help. Also, the fox (red) and Ind. 5 (pink) are hard to distinguish.*

A: We used a color scheme recommended for color blind people but we do agree with the comment of the reviewer and we modified the colors. We also feel that the use of the 95% ellipse now helps the distinction

- *Move labels so they are not on top of datapoints/lines. Lines to labels look like a continuation of data-connecting lines, but if your color codes are clear, you will not even need labels and can therefore avoid these issues.*

A: We moved the labels as recommended.

- *It would be worthwhile to think about possible rearranging the panels by switching C and D so the plots with Zn have the same axis lined up. This way, the plots with $\delta^{15}\text{N}$ data is on the left side, albeit on different axis, which is fine, as if want TP on the y-axis. In A, you could also plot $\delta^{13}\text{C}$ on the y-axis and $\delta^{15}\text{N}$ on the x-axis to then have N in A and C on the same axis, but that not how a C vs. N plot is usually shown, so whatever you prefer.*

A: Thank you for this comment. It has been done. In addition, Figure 2 has been subdivided into Figure 2 and Figure 3, for more readability and homogeneity.

- *I do like that you call it " $\delta^{15}\text{N}_{\text{collagen}}$ " here (see my comment below). This seems however not to be done consistent throughout the ms. I suggest to use $\delta^{15}\text{N}_{\text{collagen}}$, $\delta^{13}\text{C}_{\text{collagen}}$, $\delta^{34}\text{S}_{\text{collagen}}$, $\delta^{66}\text{Zn}_{\text{enamel}}$, $^{87}\text{Sr}^{86}\text{Sr}_{\text{enamel}}$ etc. consistently in the text, SI, table and figures.*

A: It has been done

- *Legend on the left:*

○ *You do not have a "group" of carnivores, herbivores, or unassigned humans anywhere (and Ind. 13? Hard to see), remove from legend accordingly.*

A: It has been done

○ *Change colors of key of the species according to their diet (i.e., the Canidae will always be red, so no need to have a black symbol here).*

A: It has been done

○ *Seems the species key is sorted by alphabet, but I can't detect a pattern in the key for the groups.*

A: It has been corrected.

○ *You could make clear that Ind. and "unassigned" are the humans in the legend. Maybe even indicate which one is adult, which one infant.*

A: We hope the new labels help answering this comment.

- *No need to specify $\delta^{15}\text{N}$ twice in the caption*

A: It has been corrected

- *Use Latin name throughout (e.g., for “fox”).*

A: It has been corrected

- *The differences in the spot size for adult and infant are difficult to see, think of different way to make this clear.*

A: We hope that the use of the polygons helps to address this comment.

- *Line 194: remove “in literature”.*

A: It has been removed

Line 202 and elsewhere: You only have two carnivores (which potentially are not even true carnivores), this is problematic when calculating TLS. I understand that you already sampled all available carnivore teeth (well done!), nothing to be done here. However, you need to tone down your argumentation here a bit. There is a large variation in e.g., N and Zn data in the herbivores (and humans, for that matter), and you have no control to understand these patterns in small carnivore dataset.

A: We thank the reviewer for this comment. according to Wilman et al 2014, *Vulpes vulpes* are feeding on 90% of animal foods, and the canids eating the smallest amount of meat present in the region in this time period (ref: <https://una-editions.fr/co-evolution-hommes-canides-en-afrique-du-nord-ouest-une-longue-histoire/>) still feed on 80% of meat. Most of the canids found at Tatoralt are *Canis aureus* that feed on 90% of animal foods. We therefore changed the word carnivore for meat-based, see lines 184, 228. In the caption, carnivore was replaced with Canids (see Figure 2 and Figure 3).

Still, most of the Zn comes from meat products, and fruits have very low Zn concentrations. Foxes feed on 90% animal foods and 10% fruits. Zinc isotopes of the dental enamel of a fox should be undistinguishable to that of a purely carnivorous animal.

The reviewer has a good point on the small dataset and we toned down our statements, see for example lines 260-262, line 228.

Line 204f: I do not think you only have considered teeth after weaning here, but also included permanent dentition that typically mineralize pre-weaning. See my comment above.

A: This has been corrected.

Line 219: This is not a $\delta^{66}\text{Zn}$ interpretation, this is an Interpretation of $\delta^{66}\text{Zn}$ data.

A: Thank you for your suggestion. It has been corrected.

Line 245ff: Here or at least in the SI you should state if you used the same (or some of the same) individuals (in sheep or humans) as Lee-Thorp (don't forget the dash in the name!). How big is their dataset? Did they use dentine or bone collagen? You state correctly that they the focused on sheep, but they also have Bos, Equus, Gazelle, how do these values compare? The main results of Lee-Thorp is that the Sector 10 humans did consume a reasonable amount

of animal resources, but no marine resources, which fits (partly to your conclusion). Elaborate!

A: Thank you for your valuable feedback and suggestions. We have taken your recommendations into account and made the necessary updates to provide information regarding the comparison with Lee-Thorp study (paragraph 4.3.3, Supplementary Information 4). We also referenced the supplementary information in the main paper to ensure that the readers can review the details.

Line 255: That cereals have generally higher $\delta^{15}\text{N}$ values compared to forage plants is a very important point for your study. Here I'm missing some info/discussion about this.

A: Thank you for your comment. The differences in isotopic ratios between the plants in the baseline diet of humans and the herbivores they consumed are indeed an important aspect of our research. We have addressed this concern by providing a more detailed explanation of the factors contributing to the observed variations in $\delta^{15}\text{N}$ values between grains and forage plants, as well as the potential implications for our study in the main paper (see lines 293-297).

Line 264: Should it say vegan instead of vegetarian? Do you have any evidence of animal products like dairy or eggs that are considered part of a vegetarian diet.

A: We thank you for this comment. While there is no evidence of milk or other dairy food consumption in the site, the presence of the ostrich eggshells in Iberomarusian sites could indicate that they consumed eggs occasionally. However, you are correct that in this context, the word vegetarian is incorrect, and we changed it for “vegetalian” which only refers to the diet (whereas vegans also exclude animal products from other aspects of their life).

Line 266: So there are cut marks on the sheep? I think this is new information for the reader which is quite crucial.

A: We appreciate the reviewer’s comment. There are cut marks on many faunal species on the site but primarily on the Barbary sheep since it is the predominant species in the site. We have clarified that this supports the notion that animal proteins were also a part of the diet of this population at Taforalt (lines 246-249).

Line 270ff. This is another example where some crucial information is lacking in the data table and figures: Which sample is di2 and which one the long bone and the rib (both simply called “bone” in Table SI3, and not labeled at all in the figures)? Also, the info when these bones formed is in the text, but missing in the table (where is this information from? Please add reference!). In the table the “age of crown dev.” (development?) of the dm2 is at 10 months, you sampled the bottom of the tooth and estimate that this formed at the beginning of the first year. This is confusing, how does the “bottom” of a tooth form before the crown? Or did I misunderstand the abbreviation? Moreover, in the SI with the Scans of the tooth the collagen sampling is shown through the whole length of the tooth, not just the bottom.

A: We thank the reviewer for the valuable feedback regarding the clarity of information in our paper. The reviewer correctly pointed out the lack of clear identification of the formation of the dm2, long bone, and rib samples. To enhance clarity, we ensured that these samples were

clearly labeled in both the data table (Tables S6.3, S6.4), figure 4 and main text (line 312-315).

We would like to clarify the information regarding the 'age of tissue development' in Table S3.3 and S6.3 referred to in the manuscript line 174. In the table, we added two distinct columns related to the dm2 tooth: one for the formation time of the enamel sampled for Zn and Sr. The other columns are for the formation of the tissues sampled for collagen. For the dm2 as it was sampled as a whole for collagen we have estimated the age of between -0.34 and 1 year since the Individual died between 6 and 12 months.

Line 289: not clear what you refer to as a “broad-spectrum and dietary breath models”, elaborate.

A: We thank you for your comment regarding the clarity of the "broad-spectrum and dietary breadth models" in our manuscript. Following your comment, we have provided a comprehensive explanation of these models in our paper (lines 335-340)

Line 290: what do you consider as higher and lower ranking resources in the light of your study? Lower: everything but animal products? Only wild plants? Higher: meat only? Cereal? Explain! How does cooking (charred plant remains) play a role here?

A: Thank you for your comment. In this context, higher-ranking resources are exemplified by the large and medium ungulates, which include species such as the Barbary sheep and gazelles, that were the primary focus of exploitation at Taforalt. Lower-ranking resources encompass smaller game animals, such as small birds and lagomorphs, as well as wild plants. Specifically, at Taforalt, lower-ranking resources would include items like acorns, pine nuts, and some legumes.

Following your comment, we decided to deliberately refrain from using the terms 'high-ranked' and 'low-ranked' to avoid subjectivity in the classification of food resources. Instead, we have directly referred to the types of food resources.

Cooking, as indicated by the presence of charred plant remains, underscores the significance of preparation methods in making these lower-ranking resources more digestible and nutritionally valuable.

For Line 300: you mentioned the possible storage in baskets before (in the intro), I suggest to move the info about the Alfa grass up where you mention this observation first.

A: We have moved the information in the introduction as suggested.

Line 302: Reference for seasonality of plant maturity?

A: Thank you we added the reference (line 361) .

Concluding remarks:

Line 319. “Younger Dryas (YD)”: You should only introduce an abbreviation if you use it a few times afterwards. The only time you mention this term again (Line 327), you don't even use the abbreviation.

A: We thank the reviewer. Following your comment, we do not use the abbreviation anymore (lines 378 and 387).

Supplementary Information:

Generally, the SI is full of minor and major mistakes and written extremely sloppy. Please carefully read this again and correct mistakes and formatting errors throughout, I can't point them all out in here. Generally, the SI would greatly benefit from some reorganization. Report results of each analyzes in the respective section,

A: We apologize for the mistakes and formatting errors in the Supplementary Information. We have addressed all comments below in addition to other mistakes we noticed not mentioned by the reviewer. We have greatly improved the presentation, readability and language of the SI.

Line 22: instead of using "distant from village" indicate direction (e.g., northeast).

A: It has been done (line 37)

Line 37ff. There is a large age gap between the two units. Any idea why (erosional surface? Hiatus?)?

A: We appreciate your question about the age gap between the Yellow series (YS) and the Grey series (GS). We want to clarify that there is no significant age gap between these two units. The Yellow series commences at approximately 22,292 cal BP and continues uninterrupted until around 15,000 cal BP. The sedimentation change that marks the transition between these two series occurs around 15,190-14,830 cal BP.

We have taken your input into consideration and have modified the paragraph in the text to make it clearer to the reader (line 55-64).

Fig. S1.1: The maps in the lower right corner should be reworked. The first one is not really needed, as a map like this is in the main text (Fig. 1). The second, closer map does not help at all without any names, rivers etc. Maybe instead use the zoomed-in map with some labels, and then an even more zoomed-in map with the location of the cave within the village?

A: Thank you for your suggestion. We modified the maps accordingly (Figure S1.1).

Line 105: I suggest to use "preferred" instead of "favorite"

A: Thank you. We changed the word as suggested.

Line 107: Should be "was primary consumed by this population" instead of "was primary consumption by this population"

A: Thank you, the phrase has been corrected.

Line 110: punctuation mark missing

A: We added the punctuation mark.

Table S1.1: I suggest to sort these by abundance

A: We appreciate your feedback. We sorted the table by the abundance of the species.

127ff: I am missing information about possible diagenesis. How are your C/N ratios (you note these in Table S1.3, but do not mention this data at all. You need to show that you can eliminate the effect of possible overprint of the dietary signals.

A: Thank you for your comment regarding the potential influence of diagenesis and the need to address the possible overprinting of dietary signals. In response to this comment, we included a section in the manuscript that discusses the potential impact of diagenesis on the C/N ratios (lines 172 and 673).

Line 144: Explain what you mean with “different sources”

A: Thank you for your feedback regarding the need for clarification on what we meant by 'different sources'. We have taken your comment into consideration and have added an explanation in the text to make it clear (lines 197-204).

Line 154: refer to most important studies here (for bulk isotope analyzes)

A: We have cited now the most important studies for bulk isotope analyses in the section of bulk carbon and nitrogen isotopes.

Line 166: reference for this equation is missing here

A: Thank you we added the reference

Line 170: you stated the abbreviations for Phe and Val already above. Only do this once and then use abbreviations!

A: Thank you for pointing that out. We used the abbreviation as suggested.

Line 183: decide if you use “terrestrial C₃/C₄ consumers” or “C₄/C₃ terrestrial consumers” and stick with it

A: We decided to use the term “terrestrial C₃/C₄ consumers”. We have changed that throughout the text.

Line 174/175: does this really refer to all animals, or only herbivores?

A: Thank you for pointing out the potential confusion regarding the term 'animals.' To eliminate this ambiguity and ensure clarity, we have modified the text by replacing 'animals' with the term 'consumer.' This change accurately reflects that we are specifically referring to consumers in the context of our study, which includes herbivores and other organisms within the same ecological environment.

Line 211: By how much?

A: The requested modification has been made in the manuscript. We added the approximate difference in $\delta^{66}\text{Zn}$ values between herbivores and carnivores (0.30 to 0.60 ‰) with the appropriate literature citations.

Line 227: ...is associated with low...

A: Thank you, we have made the correction.

Line 235: the statement "This later is produced by the process of the radioactive decay of rubidium Rb" is repeated in a few lines down again.

A: Thank you for your feedback. We have revised the paragraph to eliminate the repetition of the statement.

Line 254f: "For humans, strontium is incorporated into their skeleton via diet". Isn't that also true for all other animals?

A: Thank you for pointing that out. Indeed, strontium is incorporated in humans and other animals. We have corrected it.

Line 267: should be: "...close to +20 ‰ which..."

A: We corrected the sentence.

Line 282: and elsewhere: Delete large numbers of random paragraph breaks

A: We appreciate your comment. We deleted the breaks.

Line 295: Here you correctly state that the sampled carnivores might not have a pure carnivorous diet, since many canid species can feed on a variable amount of plant material. This is very important for your interpretations and should definitely mentioned in the main text!

A: Thank you for your insightful comment regarding the importance of acknowledging that the sampled carnivores might not have a pure carnivorous diet. Therefore, we highlighted this aspect in our main text (Main text, line 226).

Line 297: you only talk about the carnivores here. What about the herbivores? Looks like they were all clearly identified?

A: Thank you for bringing attention to this omission. We have made the necessary modifications to the manuscript to ensure that the identification of herbivores is now included and discussed alongside that of carnivores (lines 370- 379).

Line 342: should be taxonomic identification

A: It has been done

Table S3.1: what is ID 34566, Bone or tooth?

A: Thank you for pointing that out. It is a tooth, we corrected it in the table.

Line 367: Here you state correctly that the hypoplasia can be the result of weaning, but you sample material across this line and still use it as “adult” diet if I understand correctly. Elaborate

A: Thank you for your insightful comment. We appreciate your feedback and have taken it into consideration. In response to your concern, we have elaborated on the potential causes of linear enamel hypoplasia, highlighting that it may not be solely a consequence of weaning. As indicated in our revised paragraph, we found the presence of linear enamel hypoplasia in teeth formed after the weaning period, such as the M2 and M3 molars (414-422). This finding suggests that factors beyond weaning, such as ongoing nutritional stress or traumatic events, could also contribute to the formation of these stress markers.

Additionally, this stress is also present in a di1 which is formed during the first year of life and is associated with a breastfeeding period.

Line 406: correct spelling of your own institute is with capital first letters

A: It has been corrected.

Line 413f: why did you not sample systematically in all teeth?

A: We appreciate the reviewer's question regarding the systematic sampling of teeth in our study. While we aimed for a comprehensive sampling approach, it's important to note that not all teeth were sampled systematically. The reason for this variation in sampling frequency was primarily due to the condition of the teeth themselves. In some cases, we encountered teeth with insufficient enamel (which we sampled for Sr too) along the height of the crown, often a result of wear. This has made it challenging to collect enamel samples without risking further damage to the teeth. We modified our paragraph to clarify this sampling strategy : paragraph 3.3.1.2.

Line 421: You mean the age of the individual at the time when the enamel was formed. Rephrase

A: We have rephrased the sentence as requested.

Line 431: did you only serial sample equids?

A: We also did serial sampling for some Barbary sheep, hartebeest, and the Rhinoceros. Following your comment, we have changed the sentence to include those species too section 3.3.2.

Line 457: You mean measured in duplicates? Was the enamel also pre-treated in two separate batches so you have true duplicates?

A: To address your query, we would like to clarify that we conducted duplicate measurements for each enamel sample. This means running the samples twice to ensure the reliability and precision of our isotopic ratio measurements. We have clarified this in the paragraph (line 516-517).

Line 458: report typical standard deviation

A: Thank you for your input, the typical standard deviation is reported now (lines 517-518).

Line 463: Did you use the same aliquot for Sr that you've used for Zn?

A: For our study, we sampled enamel for Zn and Sr separately and we used separate aliquots. Following your query, we have added a sentence to clarify this.

Line 466: correct to "...2 ml of 3 %..."

A: Thank you. It has been corrected.

Line 484: add units (%)

A: It has been corrected.

Line 485: name standards

A: We added the standard's name.

Line 537: you send it where? Or do you mean you measured it? Are you using the same setup as Jaouen et al., 2019?

A: We included the standard samples alongside our archaeological samples and sent the entire set to the University of Davis for amino acid analyses. This approach was consistent with the methodology used in Jaouen et al., 2019, where the same standard was analyzed with the experimental samples. Therefore, we followed a similar setup to Jaouen et al., 2019, by measuring the standards in the same analytical run as our samples to ensure comparability and consistency in the results.

Figure S4.1.: Add units to axis. Error bars are 1σ ?

A: We added the unit. The error bars are the standard deviation for the amino acids measurement (now S4.7).

Figure S4.2.: Delete unnecessary digits

A: We added the units and deleted the unnecessary digits (now Figure S4.8).

Line 601 and elsewhere: usually, three digits are sufficient to show significance. Use $p < 0.001$ here.

A: It has been modified.

Figure S4.3: Again, think about human vs. animal. Maybe use non-human fauna instead?

A: Following your comment, we decided to use the term fauna for non-human samples in our study.

*Line 621: You have two individuals here that you refer to as carnivores, right? A M1 from a *Vulpes Vulpes* and a M? from a *Canidae*. Therefore, a) the sample number ($n = 2$) is much too small to run these statistics, b) M1 most likely incorporates a nursing signal in the fox*

(who knows about the molar from the Canidae) and c) at least the fox and possible the other canid is not a “true” carnivore but often supplements its diet with plant materials etc. So, you have to tone down your interpretations/conclusions

A: We appreciate the reviewer's comment on the zinc isotopic compositions of M1 teeth in *Vulpes vulpes* and Canidae. We appreciate your feedback and would like to address your points accordingly (line 590):

- We acknowledge that our sample size is indeed small, as we are working with only two specimens. This limitation does affect the robustness of our statistical analyses, but as it has been mentioned, we did not have access to more individuals. We will make sure to highlight the small sample size and its implications in our paper.

- The formation of the M1 crown in the fox initiates during the prenatal phase and experiences calcification during the weaning period, which typically occurs around 41 days post-birth. This developmental timeline implies that the zinc value of the M1 (0.61‰) might be influenced by nursing. On the other hand, the M of the canidae shows lower Zn values indicating that it is unlikely to be affected by the nursing. The roots of the molar of both the fox and the Canidae are usually formed post-weaning thus reflecting adult diet. Zinc isotope ratios are therefore possibly impacted by mother milk consumption, which increases the $\delta^{66}\text{Zn}$. The fox might look more “herbivorous” and it actually is, due to these higher $\delta^{66}\text{Zn}$. We now make clear the influence of the mother milk consumption on the M1.

- As for the comment regarding the carnivorous nature of foxes (*Vulpes vulpes*) and Canidae. While we believe that these species are not pure carnivores, meat still makes up 90% of their diet, and fruits make up only 10% of their diet (Wilman et al., 2014). We therefore change the wording, replacing carnivore with meat-based diet. Moreover, we would like to mention that both zinc and nitrogen are preferentially absorbed from animal resources.

We incorporated these points into our discussion and made sure to tone down our argument regarding the diet of these specimens.

Line 624f: But even your permanent-tooth-dataset includes enamel that typically forms pre-weaning (see above)

A: We separated the teeth formed pre-weaning from those formed post-weaning. We now only used the values of enamel formed post-weaning for the interpretation of our data

Line 625: t missing in breastfeeding. Better, use the word nursing. Citation at wrong place (before the comma).

A: We changed it to nursing and corrected the citation place.

Line 627: Again, calculating a trophic spacing from a very small sample set without true carnivores with an adult diet is problematic to say the best...

A: Thank you for your comment. We have rephrased the paragraph to acknowledge the limited sample size of two carnivores with an adult diet. However, it is worth noting that the trophic spacing we calculated between herbivores and carnivores aligns with results from similar sites (line 598-602).

Figure S4.5: This is a good place to indicate all teeth that (might) include a nursing effect. If I am not mistaken, you use deciduous teeth only for the juvenile boxplot, but these should also include permanent teeth that typically form prior to weaning. In sheep, also only M2 and M3 form post-weaning, I think. Hence, the “juvenile” sheep data should include the incisors you sample, and the adult one only the M3 and M2 data. It looks like you included the incisors of the sheep also in the adult data. Also, fix y-axis label.

A: Thank you for your valuable feedback. We have revised the plot to better indicate teeth formed before weaning and those formed after weaning (now Figure S4.2).

Line 653ff: Move N results to N section

A: Thank you for the feedback. We have now separated the Zinc and the Nitrogen results into different sections (Supplementary Information 4).

Table S4.5: Confusing to have the N results in the Zn section.

A: We have moved the N results and plot to the Nitrogen results section.

Figure S4.6: Again, separate teeth into pre- and post-weaning (here and elsewhere, also for the non- human fauna)

A: We have separated the teeth into pre and post-weaning (now figure S4.3).

Figure S4.8: Looks like at least in Ind 1 a weaning effect is visible with lower Zn values in the I, C and P4 and higher ones in M2s (apart from the one M2 with a low Zn value. This is not clear in Ind. 5 and the unassigned individuals; this needs to be discussed. You have a tooth-type vs. Zn values figure below (Fig. S5.1), would be helpful to have that Fig. up here. Also, make the same Fig. with N data vs. tooth type.

A: Figure S4.8 is now Figure S4.4 and has been redone for more clarity We moved figure S5.1 to the Zinc section on results (Now Figure S5.1) and we created the requested figure of N data vs tooth type (Figure S5.2). We also now discuss the variations seen for each individual lines 634-642 and 993-992.

Line 683 and elsewhere (see also comment above): I suggest to use $\delta^{13}\text{C}_{\text{collagen}}$ (vs. $\delta^{13}\text{C}_{\text{Pro}}$ etc.) to make this clearer. If you do, make sure to change this everywhere, in this SI, but also in the main text and figures.

A: Thank you for the suggestion. We have decided to use $\delta^{13}\text{C}_{\text{collagen}}$ to make the text clearer.

Line 686: again, make sure you really only have adult diet here.

A: Following your comment, we now only included the teeth and bones formed post-weaning to estimate the adult diet

Line 690: It's an important info here that one canid seems to occupy a carnivore trophic level. State in main text?

A: We have stated this information in our main text. This concerns the fox.

Line 704: Decide if you use subscript for C₃ and C₄ or not and make the same throughout the text.

A: Thank you for your comment. We decided not to use the subscript.

Figure 765: The yellow data of Taforalt Humans is hard to see, change color scheme as this is the important data.

A: We have changed the color scheme so it would be easier to see (Figure S4.9 S4.10-S4.11-S4.12).

The lines are confusing. Looks like the line for “C3 humans” goes into the red field (C3 carnivore?) and the C3 herbivore Taforalt to the yellow field (Taforalt humans). Also, lines cross some of the labels, clean up. What is the orange group (“human”) here? If they are not C3, C4 r marine consumers or from Taforalt?

A: Thank you for your feedback. We have addressed your concerns by removing the confusing lines and deleting the "Human" group, as it did not belong to a specific diet category. The plot now features a cleaner design with a simplified legend that should be sufficient for interpretation (Figure S4.9 S4.10-S4.11-S4.12).

For Figure S4.10 and the following figures: Again, yellow hard to see. Needs at least dark outlines. Remove figure header, you also don't have any in the other figures. Add vs. which standard to axis titles.

A: We have made the requested changes. The data for Taforalt should be easy to see on the plot. We removed the figure header and added the standard to the axis titles (Figure S4.10).

Line 751: I think your sample size is too small for this statement as you only have one bone from one individual, right? This tells you nothing about the range and variability of this taxon.

A: Thank you for your valuable feedback. The sample size for this particular statement is indeed small, as it is based on a single bone from a single hare individual. We removed the statement.

Line 785: for one equid but all hartebeest? Clarify. Also, here and elsewhere: the common names are spelled with small first letter

A: We have clarified our paragraph (line 684-692). We used small first letter for the common names.

Line 794f: Is the TP when excluding the adult samples which formed pre-weaning?

A: The TP of the herbivores from Taforalt still ranges from 1.7-2.3 when excluding the samples formed pre-weaning. The teeth that could be affected by nursing still have a TP value of 2.1. this is the case of an equid sample and a barbary sheep sample

Line 800 f: why do you reject this?

A: We reject the possibility of suckling for the high values for those teeth formed pre-weaning because their TP is only 2.1. This value is lower than expected if nursing had a significant effect. We added the explanation in the paragraph (line 898)

Line 801: "Hartebeest, hare, the Hartebeest and the Rhinoceros"?

A: We apologize for the oversight in our manuscript where we inadvertently repeated the word "Hartebeest." This was indeed a typographical error. We made the necessary correction.

Line 815: Again, what about permanent teeth that form during nursing?

A: We corrected the sentence so that it shows the values recorded on teeth formed pre-weaning from those formed post-weaning (line: 908).

Line 839: Did you analyze all humans? Is this an average value?

A: For the CSIA, we analyzed the human that have yielded enough collagen for the analyses. We took the chance to clarify the number of samples and the average values in the paragraph 4.4.3.3.

Line 872ff: Obviously, reading your ms until here I was missing this section! Parts of this definitely needs to be included in the main text. However, your argumentation here does not convince me at all! What about the incisors and canines. Or did I misunderstand that you included their results also in your statistics and interpretations?

A: We now refer to this section in the ms, line 172, and changed all figures to have this effect clearly underlined. It is true that some pre-weaning data were mixed with post-weaning one in the statistics, and we therefore now only considered post weaning samples, and make it clear in the text.

Figure S5.1: Do these Rennes and Lapa do Santo studies also have $\delta^{15}\text{N}$ analyses? I would like to see the same plot for bulk collagen $\delta^{15}\text{N}$ and other analyzes you conducted on all tooth types.

A: There are no available $\delta^{15}\text{N}$ data for the sites of Rennes and Lapa do Santo. Regarding the plot, we made a similar plot with bulk collagen $\delta^{15}\text{N}$ vs the tooth type in each individual. Thank you for the suggestion (Figure S5.2)

Figure 5.2 (should be S5.2): This is not very helpful. A line for each individual sorted by tooth type as in the previous study would be more helpful.

A: Thank you for the recommendation we changed the box plot into a line for each individual sorted by tooth type (Figure S5.2)

Line 915: reference for equation?

A: We added the correct reference for the equation.

Line 917: "still"? Compared to what?

A: We are sorry for the confusion. We calculated the average TP level of all the adult humans using our zinc equation and we compared it to the TP equation for amino acids. We made sure to clarify this in the Supplementary information (line 1025).

Line 946: Do humans with low status usually die younger? The sampled individuals here all have died young, can you elaborate on that?

A: Following your comment, we added a paragraph elaborating on the age of death of this population and its relationship to their diet (section 4.6)

Line 973 and elsewhere: the unit (‰) belongs after the stdev

A: It has been corrected.

Line 2021: You state that “the shells possibly were used as ornaments or grave goods that accompanied the grave of the disease”. That’s what comes to mind first of course. But, why is that, so you have a reference or is this just a well-educated guess? As this is unimportant for your study, I would remove this sentence.

A: Thank you for your feedback. We have removed the sentence.

Line 1031: I agree that the sample is reflecting breastfeeding. But you have yet to convince me that early forming teeth of adults do not show the same.

A: The deciduous second molar forms earlier than than the M1, which root (which we sampled) formed after the age of 2.5. It explains why the pattern is different.

Excel Datasheet (some of these corrections/comment should be considered for all tables):
Table S11:

- *typo in header (study)*

A: Thank you in has been corrected

- *Gazelle is not a carnivore!*

A: It has been corrected, thank you

- *The Canidae is an M1, right? At least that’s what you state in Table S13. Please crosscheck all for to have correct data here!*

A: The fox is an M1 but the Canidae is an M.

- *Water intake after which study. Fill empty field of the one Hartebeest.*

A: We added the reference for the water intake in the table S11. The field for the hartebeest has been corrected.

- *Clearer if sorted by diet/taxa instead of SEVA ID.*

A: Thank you for the suggestion, we have sorted the table by taxa.

- *I guess the actual sample ID is #REF? Indicate which catalogue this refers to. Some of these IDs have a dot, others don’t, is this correct? Also check in other tables.*

A: The #REF is the ID given to the samples during excavation. We change the #REF to “ID excavation” to make it clear for the reader. We removed the dots from the IDs. It has been checked in the other tables.

- *Again, here you have a few mammalian teeth that typically form during nursing.*

These should be indicated and removed from or highlighted in certain data compilations.

A: We have added in Table SI4 the teeth that are affected by nursing for which isotopic analyses. We did not include the teeth that are formed pre-weaning in the data interpretation.

Table SI2:

- *Again, indicate material that typically forms pre-weaning*

A: We indicated the samples that are formed pre-weaning.

Table S3:

- *Refer to citations correctly, add year and doi.*

A: We added the citations in the last column with the year and doi.

- *Add units for all columns with data, not just some (e.g., add ‰ for $\delta^{66}\text{Zn}$, not ppm in the column with the concentration).*

A: Thank you, we added the units.

- *“What does the “dev” at Age crown dev. Stand for? Explain all abbreviations.*

A: Following your comment, we decided to change the name into “Age of initiation of the enamel” and “Age of completion on the sampled enamel”.

Table S4:

- *Add units,*

A: Done

- *Here you only have SEVA IDs, but it’s called “sample name”. Make consistent. You could add collection ID as well.*

A: We changed it to SEVA. Additionally, we added the ID of excavation as well in Table S6.3 and Table S6.4.

Table S5

- *Add units.*

- *Move caption above table.*

A: We added the units and moved the caption above the table.

Table S7:

- *Use same text size throughout*

A: It has been corrected.

- *The colors of each individual do not correspond to the color in Fig. 1. Choose one color scheme and stick without throughout the whole MS, including SI and Figures.*

The colors have been changed to match those in Figure. Thank you for the feedback.

General

- *Check consistency e.g., space between number and unit, the use of “cal. BP” vs. “cal BP”, using the delta notation once introduced*

A: It has been corrected. We switched to cal BP.

- *Search for double spaces and delete them*

A: It has been corrected

- *Delete spaces before citation ID*

A: It has been corrected

- *Table captions should be above the tables.*

A: It has been corrected

- *Make sure you introduce abbreviations when you first use the term in the text and SI. After that, stick with the same abbreviation and don't introduce them again.*

A: We corrected the abbreviations

- *Use a coma after e.g.*

A: It has been corrected

- *Do not mix British and American English.*

A: Thank you, we went through the manuscript to switch everything to British English

- *Element names are spelled with small starting letter. If you use the abbreviation, it's starts with a capital letter (e.g., nitrogen and N). Correct throughout ms.*

A: It has been corrected

- *Letters until 12 are spelled out if they are not followed by a unit*

A: We are not sure we understood this comment. Do you mean numbers?

- *Decide on capitalization of the headers and stick with it*

A: It has been corrected

- *Use same units. E.g., for hour you sometimes use h, sometimes hr, sometimes hrs and sometimes spell it out, I think. Make this consistent.*

A: It has been corrected

References cited in the response:

Bogaard, A., Heaton, T.H.E., Poulton, P., Merbach, I., 2007. The impact of manuring on nitrogen isotope ratios in cereals: archaeological implications for reconstruction of diet and crop management practices. *J. Archaeol. Sci.* 34, 335–343.

<https://doi.org/10.1016/j.jas.2006.04.009>

Fraser, R.A., Bogaard, A., Heaton, T., Charles, M., Jones, G., Christensen, B.T., Halstead, P., Merbach, I., Poulton, P.R., Sparkes, D., Styring, A.K., 2011. Manuring and stable nitrogen isotope ratios in cereals and pulses: towards a new archaeobotanical approach to the inference of land use and dietary practices. *J. Archaeol. Sci.* 38, 2790–2804.

<https://doi.org/10.1016/j.jas.2011.06.024>

Fraser, R.A., Bogaard, A., Schäfer, M., Arbogast, R., Heaton, T.H.E., 2013. Integrating botanical, faunal and human stable carbon and nitrogen isotope values to reconstruct land

use and palaeodiet at LBK Vaihingen an der Enz, Baden-Württemberg. *World Archaeol.* 45, 492–517. <https://doi.org/10.1080/00438243.2013.820649>

Hedges, R.E., Reynard, L.M., 2007. Nitrogen isotopes and the trophic level of humans in archaeology. *J. Archaeol. Sci.* 34, 1240–1251.

Turner, E., 2020. Large mammalian faunal assemblages, in: Barton, R.N.E., Bouzouggar, A., Collcutt, S.N., HUMPHREY, L. (Eds.), *Cemeteries and Sedentism in the Later Stone Age of NW Africa: Excavations at Grotte Des Pigeons, Taforalt, Morocco*, Monographien Des Römisch-Germanischen Zentralmuseums. Monographien des Römisch-Germanischen Zentralmuseums, pp. 239–308. <https://doi.org/10.11588/propylaeum.734>

Wilman, H., Belmaker, J., Simpson, J., de la Rosa, C., Rivadeneira, M.M., Jetz, W., 2014. EltonTraits 1.0: Species-level foraging attributes of the world's birds and mammals: Ecological Archives E095-178. *Ecology* 95, 2027–2027.

Reviewer 2

The manuscript "*Earliest isotopic evidence of high reliance on plant food in the Late Pleistocene hunter-gatherer population (Taforalt, Morocco)*" by Ms Moubtahij and co-authors, which has been submitted for publication in Nature Ecology & Evolution lies within the scope of this journal. This research deals with the issue on the origin of agricultural-based economy and the role of increased reliance on plant foods in the Late Pleistocene hunter-gatherer population of Taforalt (Morocco) with parallel to the Natufian experience. In order to reconstruct the dietary habits and the mobility pattern of the human groups from Taforalt, the authors conducted for the first time a strong multi-isotope approach which combines zinc ($\delta^{66}\text{Zn}$) and strontium ($^{87}\text{Sr}/^{86}\text{Sr}$) analysis on dental enamel, carbon ($\delta^{13}\text{C}$), and nitrogen ($\delta^{15}\text{N}$) isotope analysis on dentin and bone collagen, along with single amino acid analysis, which makes this manuscript highly important. The isotopic analyses were conducted on a high-resolution data containing human remains and associated fauna recovered in the recently excavated burials from sector 10 at Taforalt. The study results emphasize to us the adoption of a starchy diet along with a decreased consumption of meat compared to Upper Palaeolithic sites with available isotopic data in Europe and Asia. Furthermore, the trophic position (TP) values at Taforalt are similar to the TP values of Neolithic farmers from the Levant, which enhances the evidence for substantial plant consumption, thus challenging traditional ideas about pre-Neolithic hunter-gatherer societies.

Overall, the manuscript is well-written. Data collection is adequately and openly presented in sufficient detail with additional information structured into 7 chapters provided in the supplementary information. The literature cited is very informative and relevant to the topic of the current manuscript. All figures are appropriate and the statistical tests are displayed with accuracy. The argumentation is well stated as it is clearly indicated in the abstract. However, some assumptions may need clarification or require to be substantiated by references. Here are listed the main points along with the comments:

Lines 95-97 The definition given by the authors to the Iberomaurusian hunter-gatherers needs to be enhanced by appropriate references. Here are some comments/suggestions: the production of blades and bladelets is not only peculiar to the Iberomaurusian knappers. Instead, a term such as bladelet-based stone technology is more appropriate.

A: Thanks for your feedback! We've incorporated your suggestion and now refer to the stone technology as "bladelet-based stone technology". We appreciate your input and believe this change enhances the clarity and precision of our terminology (line 94).

(3) there is no agreement between scholars as for the longevity of the Iberomaurusian (whether the end of the Iberomaurusian falls within Terminal Pleistocene or rather coincides with the onset of the Holocene)¹.

A: We appreciate the reviewer's comment. In response, we have revised the sentence and incorporated the appropriate reference (line 96).

(4) In regard to the age 25,000 cal BP, the authors may refer to scholars who have published the lithic record of Tamar Hat basing on their personal examination^{2,3}.

A: We thank the reviewer for the suggestions. In response, we have included the relevant references (line 96)

Line 119: No burials were found at Tamar Hat. Only 6 deciduous teeth were recovered and they belong to individuals who were not buried in this site⁴. Furthermore, I suggest the term sheltered sites instead of caves, since Afalou is a rockshelter.

A: Thank you for your valuable feedback. We have addressed your comment by removing Tamar Hat as a site with a cemetery. Furthermore, we've incorporated your suggestion and adjusted the terminology to now refer to 'sheltered sites' instead of 'caves' (line 126).

Line 128: Afalou bou Rhummel burials (layer IV) are older than those of Taforalt (and it is likely that layer V burials are of a similar age or slightly older⁵. If they are proven not to be, the authors need to provide the reference.

A: We now refer to the cemetery of Taforalt as one of the oldest cemeteries in North Africa. We believe this adjustment accurately reflects the available evidence.

Lines 293-296: This research emphasizes the role of highly reliance on cariogenic wild plant food and teeth carries. Maybe it would be germane to also mention that dental evulsion (which has not addressed in this research) is not believed to be linked directly to oral pathology⁶. Furthermore, I recommend that the authors mention (maybe in the introduction) the archaeological records concomitant with the LGM showing evidence for plant processing. I refer in particular to Ifri el Baroud where small seeds of wild legumes may have probably been gathered for consumption purposes⁷, or Tamar Hat where wild seeds of pine cones with pestle-grinders were recovered⁴. It is also relevant to cite Ohalo II, a late Upper Paleolithic site in the Levant where the earliest known usage of plants ca. 23,000 years ago has been reported⁸ (Weiss et al. 2008). In all three cases, no substantial use of edible plants has been evidenced, but this will better support the relevance of the study results of the current manuscript.

A: Thank you for your comments.

- We appreciate your suggestion to mention dental evulsion. We now mention this in line 349-350.

- Additionally, we have incorporated archaeological records concomitant with the Last Glacial Maximum (LGM) in the introduction, citing Ifri el Baroud, Tamar Hat (line 108), and Ohalo II (line 72). We believe that these references provide valuable context to support the relevance of our study results, as suggested.

Lines 300-302: According to the statement of this paragraph, year-round assumption at Taforalt relies on the availability of edible plant-harvesting and possibly storage. However, relevant signs of seasonality are evidenced from selective hunting of Barbary

sheep and collection/consumption of edible molluscs⁹. Authors may provide further clarification to better enhance the increased sedentism assumption at Taforalt.

A: We have revised our argumentation for the year-round assumption at Taforalt, presenting it as a possibility rather than a certainty given the limited data on this matter. Thank you for your constructive feedback (line 366-376).

Lines 303-305: Yes, the population of Taforalt increased their reliance on plant resource, but there is no significant change in the availability of higher-ranking foods, if we assume that the local (and primary) game Barbary sheep is the highest-ranked resource, and then the main source of protein⁹. Besides, Taforalt burials were directly dated to 15,077 - 13,892 cal. BP (cited in this manuscript), which coincides with the rapid warming period GSI. In this case, why Late Iberomaurusian populations in Taforalt have shifted to a plant-based economy if there are no signs of population experiencing resource stress, or climatic deterioration? At another level, there are no records of flakes or blades with macroscopic gloss, and none of the microliths showed evidence of any mastic or remains of hafting materials⁹, although the prevalence of charred plant materials. I understand these issues are beyond the scope of this manuscript, but it would be relevant to briefly raised them in the text to enhance the argumentation.

A: We appreciate the reviewer's insightful comment regarding the availability if higher-ranked foods. We acknowledge that there is no decline in the higher-ranking foods (Barbary sheep). We now addressed this aspect (line 370). Thank you for your valuable input.

Here are additional remarks:

The title: I would rather suggest to mention the term Later Stone Age (already used in the literature) instead of Late Pleistocene (c. 129,000 and c. 11,700 years ago), since the collected data belong to the Iberomaurusian which covers a very small part of the Late Pleistocene. The term Pre-Neolithic, already mentioned in this paper is another option.

A: Thank you for the suggestion. We changed the 'Late Pleistocene' to 'Later Stone Age' in the title.

Line 87: It would be useful to also cite the new paleogenomic study conducted on Epipalaeolithic-Middle Neolithic human remains from Morocco (Simões et al. 2023) which concluded to a mosaic of incoming groups admixing with local people¹⁰.

A: We thank the reviewer for the suggestion. We incorporated this reference into our paragraph (line 87)

Line 99: References are needed to assert the idea of the absence of domestication during the Iberomaurusian¹¹⁻¹³.

A: We now included the appropriate reference (line 100).

Line 131: the sentence should be “...warming period following (and not during) the Last Glacial Maximum (LGM)”. Also, Poti et al. 2019 are cited instead of Barton et al. 2019.

A: We thank the reviewer for pointing out the errors in our citation and sentence structure. We corrected the citation and the sentence (line 141).

Line 152: remove the repeated word.

A: We removed the repeated word

Line 200: I think a word is missing here.

A: We revised the missing words

Line 320: Authors need to add uncal BP to the age of Younger Dryas.

A: We added the uncal BP to the age of Younger Dryas (line 391).

Supplementary Information 1 - Line 108: It is worth mentioning that Barbary sheep horn cores used as funerary objects accompanying human burials in sector 10 at Taforalt have also been reported at Afalou bou Rhummel^{14,15}. This is to raise the idea that some common and synchronous idiosyncratic practices occurred in different biogeographic entities during the Late Iberomaurusian.

A: Thank you for this suggestion, we have included it in the paragraph

Reviewer's decision

This is an interesting and original paper. As reported by the authors, there is no compelling evidence at Taforalt for the precocious domestication of either plants or animals. However, the current research demonstrated the abundance of plant resources in the environment of the Pre-Neolithic groups led to a greater concentration of foraging for lower-ranking resources. Further research into coastal sites is needed to distinguish between dietary habits rigorously determined by different local environments. I expect this paper will draw interest from researchers in different fields who are interested in the origin of farming. Therefore, I would highly recommend publishing the manuscript after the minor corrections which have been noted above.

References

16. Hogue, J. *The origin and development of the Pleistocene LSA in Northwest Africa: A case study from Grotte des Pigeons (Taforalt), Morocco*. Oxford: University of Oxford, PhD thesis (2014) https://ora.ox.ac.uk/objects/uuid:3a95e8b0-5e0b-4e8b-baac-b2e44a327d8a/download_file?file_format=application%2Fpdf&safe_filename=THE_SIS01&type_of_work=Thesis

17. Close, A. E. *The Iberomaurusian sequence at Tamar Hat*. *Libyca* 29, 69–103 (1981).

18. Sari, L. *Technological change in Iberomaurusian culture: The case of Tamar Hat, Rassel and Columnata lithic assemblages (Algeria)*. *Quaternary International* 320, 131–

142 (2014).

19. Saxon, E. C., Close, A., Cluzel, C., Morse, V., Shackelton, N. J. Results of recent investigations at Tamar Hat. *Libyca* 22, 49–91 (1974).
20. Hachi, S. Fröhlich, F., Gendron-badou A., de Lumley H., Roubet C., Abdessadok S. *Figurines du Paléolithique supérieur en matière minérale plastique cuite d'Afalou Bou Rhummel Bou Rhummel (Babors, Algérie) : premières analyses par spectroscopie d'absorption infra-rouge. L'Anthropologie* 106, 57–97 (2002).
21. Humphrey, L. T. et al. Earliest evidence for caries and exploitation of starchy plant foods in Pleistocene hunter-gatherers from Morocco. *Proc. Nat. Acad. Sci.* 111, 954–959 (2014).
22. Potì, A. et al. Human occupation and environmental change in the western Maghreb during the Last Glacial Maximum (LGM) and the Late Glacial. New evidence from the Iberomaurusian site Ifri El Baroud (northeast Morocco). *Quat. Sci. Rev.* 220, 87–110 (2019).
23. Weiss, E., Kislev, M.E., Simchoni, O., Nadel, D. & Tschauer, H. "Plant-food preparation area on an Upper Paleolithic brush hut floor at Ohalo II, Israel". *Journal of Archaeological Science* 35 (8), 2400–2414 (2008).
Bibcode:2008JARSc..35.2400W. doi:10.1016/j.jas.2008.03.012.
24. Barton, R. N. E. et al. Origins of the Iberomaurusian in NW Africa: New AMS radiocarbon dating of the Middle and Later Stone Age deposits at Tatoralt Cave, Morocco. *Journal of Human Evolution* 65, 266–281 (2013).
25. Simões, L.G., Günther, T., Martínez-Sánchez, R.M., Vera-Rodríguez, J.C., Iriarte, E., Ricardo Rodríguez-Varela, Bokbot, Y., Valdiosera C. & Jakobsson, M. Northwest African Neolithic initiated by migrants from Iberia and Levant. *Nature* 618, 550–556 (2023).
26. Klein, R. G., & Scott, K. Re-analysis of faunal assemblage from the Haua Fteah and other late quaternary archaeological sites in Cyrenaican Libya. *Journal of Archaeological Science* 13, 515–542 (1986)
27. Merzoug, S. & Sari, L. Re-examination of the Zone I Material from Tamar Hat (Algeria): Zooarchaeological and Technofunctional Analyses. *Afr. Archaeol. Rev.* 25, 57–73 (2008).
28. Turner, E. Large mammalian faunal assemblages. in *Cemeteries and Sedentism in the Later Stone Age of NW Africa: Excavations at Grotte des Pigeons, Tatoralt, Morocco* 239–308 (Monographien des Römisch-Germanischen Zentralmuseums, 2019).
29. Arambourg, C., Boule, M., Vallois, H. & Verneau R. *Les grottes paléolithiques des Beni Ségoual (Algérie). (Archives de l'Institut de Paléontologie Humaine (Édition Masson (1934).*
30. Hachi, S. Résultats des fouilles récentes d'Afalou Bou Rhummel (Bédjaïa, Algérie). in *El Món Mediterrani Despres del Pleniglacial (18.000-12.000 BP)* (Ed. (ed. Fullola, J.-M. & Soler, N.) 77–92 (Girona : Museu d'Arqueologia de Catalunya, 1997).

A: The references recommended by the reviewer have been added

Final Decision Letter:

1st March 2024

Dear Ms Moubtahij,

We are pleased to inform you that your Article entitled "Isotopic evidence of high reliance on plant food among Later Stone Age hunter-gatherers at Taforalt, Morocco", has now been accepted for publication in *Nature Ecology & Evolution*.

Over the next few weeks, your paper will be copyedited to ensure that it conforms to *Nature Ecology and Evolution* style. Once your paper is typeset, you will receive an email with a link to choose the appropriate publishing options for your paper and our Author Services team will be in touch regarding any additional information that may be required.

Due to the importance of these deadlines, we ask you please us know now whether you will be difficult to contact over the next month. If this is the case, we ask you provide us with the contact information (email, phone and fax) of someone who will be able to check the proofs on your behalf, and who will be available to address any last-minute problems. Once your paper has been scheduled for online publication, the Nature press office will be in touch to confirm the details.

Acceptance of your manuscript is conditional on all authors' agreement with our publication policies (see www.nature.com/authors/policies/index.html). In particular your manuscript must not be published elsewhere and there must be no announcement of the work to any media outlet until the publication date (the day on which it is uploaded onto our web site).

Please note that *Nature Ecology & Evolution* is a Transformative Journal (TJ). Authors may publish their research with us through the traditional subscription access route or make their paper immediately open access through payment of an article-processing charge (APC). Authors will not be required to make a final decision about access to their article until it has been accepted. Find out more about Transformative Journals

Authors may need to take specific actions to achieve compliance with funder and institutional open access mandates. If your research is supported by a funder that requires immediate open access (e.g. according to Plan S principles) then you should select the gold OA route, and we will direct you to the compliant route where possible. For authors selecting the subscription publication route, the journal's standard licensing terms will need to be accepted, including <https://www.nature.com/nature-portfolio/editorial-policies/self-archiving-and-license-to-publish>. Those licensing terms will supersede any other terms that the author or any third party may assert apply to any version of the manuscript.

nature portfolio

We welcome the submission of potential cover material (including a short caption of around 40 words) related to your manuscript; suggestions should be sent to Nature Ecology & Evolution as electronic files (the image should be 300 dpi at 210 x 297 mm in either TIFF or JPEG format). Please note that such pictures should be selected more for their aesthetic appeal than for their scientific content, and that colour images work better than black and white or grayscale images. Please do not try to design a cover with the Nature Ecology & Evolution logo etc., and please do not submit composites of images related to your work. I am sure you will understand that we cannot make any promise as to whether any of your suggestions might be selected for the cover of the journal.

You can generate the link yourself when you receive your article DOI by entering it here: <http://authors.springernature.com/share>.

[REDACTED]

P.S. Click on the following link if you would like to recommend Nature Ecology & Evolution to your librarian <http://www.nature.com/subscriptions/recommend.html#forms>

** Visit the Springer Nature Editorial and Publishing website at www.springernature.com/editorial-and-publishing-jobs for more information about our career opportunities. If you have any questions please click here.**